# Self-Supervised Diffusion MRI Denoising via Iterative and Stable Refinement

**Chenxu Wu**[1,2], **Qingpeng Kong**[1,2], **Zihang Jiang**[1,2,*]**& S. Kevin Zhou**[1,2,3,4,*]

[1]School of Biomedical Engineering, Division of Life Sciences and Medicine, USTC
[2]MIRACLE Center, Suzhou Institute for Advance Research, USTC
[3]State Key Laboratory of Precision and Intelligent Chemistry, USTC
[4]Key Laboratory of Intelligent Information Processing of CAS, Institute of Computing Technology, CAS
`{wuchenxu,qpkong27}@mail.ustc.edu.cn,{jzh0103,s.kevin.zhou}@gmail.com`

## Abstract

Magnetic Resonance Imaging (MRI), including diffusion MRI (dMRI), serves as a "microscope" for anatomical structures and routinely mitigates the influence of low signal-to-noise ratio scans by compromising temporal or spatial resolution. However, these compromises fail to meet clinical demands for both efficiency and precision. Consequently, denoising is a vital preprocessing step, particularly for dMRI, where clean data is unavailable. In this paper, we introduce Di-Fusion, a fully self-supervised denoising method that leverages the latter diffusion steps and an adaptive sampling process. Unlike previous approaches, our single-stage framework achieves efficient and stable training without extra noise model training and offers adaptive and controllable results in the sampling process. Our thorough experiments on real and simulated data demonstrate that Di-Fusion achieves state-of-the-art performance in microstructure modeling, tractography tracking, and other downstream tasks. Code is available at https://github.com/FouierL/Di-Fusion.

## 1 Introduction

Characterizing real-world noise using data distributions is difficult (Huang et al., 2021), particularly in non-invasive imaging modalities such as Magnetic Resonance Imaging (MRI), where the noise predominantly originates from numerous factors including thermal fluctuations (Fadnavis et al., 2020a). MRI, including its subtype Diffusion-weighted Magnetic Resonance Imaging (dMRI) (Basser et al., 1994), serves as a vital tool for observing inferred structures (Le Bihan, 2003; Le Bihan et al., 2006; Schilling et al., 2019) and necessitates a high Signal-to-Noise Ratio (SNR) for better clinical decision making. While it is possible to improve the SNR by increasing the acquisition time or reducing the image resolution, either way hinders the clinical application of MRI. Therefore, much research has focused on processing techniques like denoising for dMRI to improve its SNR and reduce acquisition time, which holds a great significance for clinical efficiency and accuracy.

The dMRI typically consists of 4D data ($X \in \mathbb{R}^{w \times h \times d \times l}$), including 3D spatial coordinates ($w,h$ and $d$) and 1D diffusion vectors ($l$), in which diffusion is measured along different gradient directions (Westin et al., 2016). Different clinical applications require varying numbers of diffusion vectors and acquisition strategies, leading to diverse noise sources and distributions, which complicates noise modeling and denoising implementation. For supervised methods (Gibbons et al., 2019; Kaye et al., 2020), not only is it non-practical to obtain paired data with high SNR and low SNR, but the diversity of dMRI also leads to distributional shifts among different datasets, resulting in a fundamental drop in their performances (Darestani et al., 2021). Different from these approaches, our method offers a self-supervised solution for dMRI denoising through a single-stage construction and an efficient adaptive sampling process. Without the need for paired training data or clean data, our method is capable of removing the noise from dMRI with a denoising diffusion model. To mitigate the drift problem, a **Fusion** process is proposed to align the forward process. Moreover, as real-world noise is difficult to characterize, a "**Di-**" process is introduced to represent the noise distribution in a more

---

*Corresponding authors

effective manner. Consequently, our method **Di-Fusion** is able to achieve better denoising results while preserving the desired anatomical structures.

The main contributions of our work are three-fold: **(i)** We propose Di-Fusion, a stable and self-supervised dMRI denoising method leveraging the latter diffusion steps (Section 3.2). Di-Fusion integrates the statistical self-supervised denoising techniques (Batson & Royer, 2019) into the diffusion models through the Fusion process and "Di-" process (Section 3.1). **(ii)** Di-Fusion enables iterative refinement through an adaptive sampling process (Section 3.3). **(iii)** With thorough comparisons on real and simulated data, Di-Fusion demonstrates state-of-the-art denoising performance in microstructure modeling, tractography, and other downstream tasks (Section 4).

## 2 BACKGROUND AND RELATED WORKS

### 2.1 STATISTICAL SELF-SUPERVISED IMAGE DENOISING

Built upon the assumption that additive noise is pixel-wise independent, Noise2Noise (Lehtinen et al., 2018) learns the process of image restoration solely by observing corrupted measurements:

$$\arg\min_{\theta} \left\{ \mathbb{E}\|f_{\theta}\left(x'\right) - x\|^2 \right\} \approx \arg\min_{\theta} \left\{ \mathbb{E}\|f_{\theta}\left(x'\right) - y\|^2 + \mathbb{E}\|x - y\|^2 \right\}, \tag{1}$$

where $x$ and $x'$ are independent corrupted measurements of the clean ground truth $y$ and $f_{\theta}$ is a denoising function which is parameterized by $\theta$. Due to the assumption of independent noise, $\mathbb{E}\|x - y\|^2$ is usually a constant. Furthermore, Noise2Self (Batson & Royer, 2019) proposes the $\mathcal{J}$-invariant theory, using only the same corrupted measurement to perform denoising. Following this theory, Noise2Void (Krull et al., 2019), Laine *et al.* (Laine et al., 2019) and Noise2Same (Xie et al., 2020) focus on how to construct unorganized collections of corrupted images by masked-based blind spot networks. Noisier2Noise (Moran et al., 2020) and Noisy-As-Clean (Xu et al., 2020) add additional noise to the original noisy image to generate training image pairs. Nevertheless, these methods exhibit a significant drop in performance when confronted with real-world noisy images, particularly when the explicit noise model is unknown (Huang et al., 2021; Mansour & Heckel, 2023).

### 2.2 DIFFUSION MODELS

Denoising Diffusion Probabilistic Model (DDPM) (Ho et al., 2020; Sohl-Dickstein et al., 2015) emerges as a powerful generative model, which is composed of a parameterized Markov chain with $T$ diffusion steps to fit a given data distribution. The forward process $q\left(x_t|x_{t-1}\right)$ serves to perturb the data by gradually adding Gaussian noise based on a pre-defined noise schedule $\beta_{1,\dots,T}$ (Following (Ho et al., 2020), $\sigma_t^2 := \beta_t$, $\alpha_t := 1 - \beta_t$ and $\bar{\alpha}_t := \prod_{s=1}^{t} \alpha_s$ are sets of predetermined constants in this paper) until the data distribution approaches a standard Gaussian distribution:

$$q\left(x_{1:T}|x_0\right) := \prod_{t=1}^{T} q\left(x_t|x_{t-1}\right), \quad q\left(x_t|x_{t-1}\right) := \mathcal{N}\left(x_t; \sqrt{1-\beta_t}x_{t-1}, \beta_t\mathbf{I}\right). \tag{2}$$

The reverse process starts from a Gaussian distribution $z \sim \mathcal{N}\left(\mathbf{0}, \mathbf{I}\right)$ and uses a parameterized Gaussian transformation kernel $F_{\theta}$ to learn the step-by-step restoration of the original data distribution:

$$p_F\left(x_{0:T}\right) := p\left(x_T\right) \prod_{t=1}^{T} p_F\left(x_{t-1}|x_t\right), \quad p_F\left(x_{t-1}|x_t\right) := \mathcal{N}\left(x_{t-1}; F_{\theta}(x_t, t), \sigma_t^2\mathbf{I}\right). \tag{3}$$

Recently, there has been a large interest in exploring ways to enhance the extensibility and sampling efficiency of DDPM. For enhancing extensibility, (Song & Ermon, 2019) uses gradient of the log density as a force to pull a random sample across the data space towards regions with a high data density characterized by $p\left(x\right)$ (Croitoru et al., 2023) by adopting Langevin dynamics algorithm (Hyvärinen & Dayan, 2005). (Song et al., 2020b) further extend the score function as solutions to reverse-time Stochastic Differential Equation (SDE) and extends DDPM to continuous states. Cold diffusion (Bansal et al., 2024) investigates the necessity of Gaussian noise or any form of randomness for diffusion models to work effectively in practical scenarios. (Zhou et al., 2024) introduces a family of processes that interpolate between two paired distributions given as endpoints. For accelerating sampling speed, (Song et al., 2020a) and (Watson et al., 2021) generalize DDPM by introducing a class of non-Markovian diffusion processes that achieves the same sampling objective.

Previous works have demonstrated that diffusion models can be effectively applied to image restoration tasks (Kawar et al., 2022; Xia et al., 2023; Özdenizci & Legenstein, 2023; Chung et al., 2022b; Saharia et al., 2022a; Fei et al., 2023). Conditioned on a low-resolution input image, (Saharia et al., 2022b) performs image super-resolution via repeated refinement. (Chung et al., 2022a) , (Song et al., 2021), (Song et al., 2024) and (Gao et al., 2023) extend diffusion models to inverse problems in medical imaging. However, these models require clean data (e.g., normal-dose CT) to capture their prior data distribution, which makes direct application of these methods to dMRI data impractical because no clean data is available in dMRI itself. Our method does not require extra noise model training or clean ground truth $y$ and can be applied to the aforementioned scenarios.

## 2.3 RELATED WORKS

The initial denoising methods employed for dMRI are adaptations of techniques developed for natural images, like non-local means (NL-means (Coupé et al., 2008) and its variants (Chen et al., 2016; Coupé et al., 2012)). Under the assumption that small spatial structures exhibit relative consistency across varied dMRI measurements, Local Principal Component Analysis (LPCA) (Manjón et al., 2013) and its Marchenko-Pastur extension (MPPCA) (Veraart et al., 2016) project dMRI to a local low-rank approximation. Training the Noise2Noise (Lehtinen et al., 2018) model directly using the same slices from different volumes can result in excessively smooth outcomes (Shown in the experiments of (Xiang et al., 2023)). So, utilizing the entire volumes, Patch2Self (Fadnavis et al., 2020a) trains a full-rank locally linear denoiser to perform volume-wise denoising. Patch2Self2 (Fadnavis et al., 2024) further enhances the computational efficiency of Patch2Self. Recently, Corruption2Self (Tu et al., 2025) extends denoising score matching to accommodate noisy observations and provides a framework for denoising MRI. A state-of-the-art self-supervised method DDM2 (Xiang et al., 2023) is proposed for denoising dMRI, which incorporates statistical image denoising into the diffusion model in a three-stage framework. However, the results obtained by DDM2 are prone to over-denoising as its performances in downstream tasks are not satisfactory (Section 4.2).

## 3 METHODS

4D dMRI consists of independent noisy samples acquired at different gradient directions. Considering $x = X_{*,*,i,j}$ ($i$: slice index, $j$: volume index) as the target slice to denoise, $x' = X_{*,*,i,j-1}$ and $x$ are independent corrupted measurements of the clean ground truth $y$. In this section, we demonstrate how to decompose the single-step mapping from $x'$ to $x$ into $T$ steps using a parameterized Markov chain (We denote $\mathcal{F}_\theta$ as the parameterized transformation kernel in our method). **We provide a complete definition of the entire Di-Fusion in Appendix B**.

There are five questions to be answered in our method. **Q1:** Since $x'$ and $x$ are still different, how can we obtain the forward process to construct the multi-step mapping between two endpoints? **Q2:** How can we represent the noise distribution without extra noise model training? **Q3:** How can training be conducted with only noisy data? **Q4:** Why does Di-Fusion only leverage the latter diffusion steps? **Q5:** How does the reverse process enable iterative refinement?

### 3.1 MODIFICATIONS OF FORWARD PROCESS

**Q1** We use $\mathcal{F}_\theta$ to map from $x'$ to $x$, considering $x'$ as $x_T$ and $x$ as $x_0$, $\mathcal{F}_\theta$ should take $x_t$ and $t$ as input and output $x_{out}$ close to $x$:

$$x + \epsilon_t = x_{out} = \mathcal{F}_\theta\left(x_t, t\right), \quad \|x - x_{out}\|^2 < \varepsilon, \tag{4}$$

where $\varepsilon$ represents a small positive value, $\epsilon_t$ is a perturbation term that depends on $t$, and $\epsilon_t$ decays as $t \to 0$. Performing the reverse process of DDPM, we find that $x_{t-1}$ should be a linear interpolation between $x_{out}$ and $x_t$ plus a noise instead of $\bar{x}_{t-1}$, the major difference is introduced by $x_{out}$ (See Appendix C.1 for detailed derivations):

$$x_{t-1} = \underbrace{\frac{\sqrt{\bar{\alpha}_{t-1}}\beta_t}{1 - \bar{\alpha}_t}(x + \epsilon_t)}_{\text{major difference}} + \frac{\sqrt{\alpha_t}\left(1 - \bar{\alpha}_{t-1}\right)}{1 - \bar{\alpha}_t}x_t + \sigma_t z \neq \bar{x}_{t-1} = \sqrt{\bar{\alpha}_{t-1}}x' + \sqrt{1 - \bar{\alpha}_{t-1}}z, \tag{5}$$

where $z \sim \mathcal{N}\left(\mathbf{0}, \mathbf{I}\right)$, $\{\bar{x}_t\}_1^T$ are obtained by directly performing the forward process in DDPM and $\{x_t\}_1^T$ are obtained from the reverse process of DDPM. Since the component $\epsilon_t \to 0$ as $t \to 0$, a

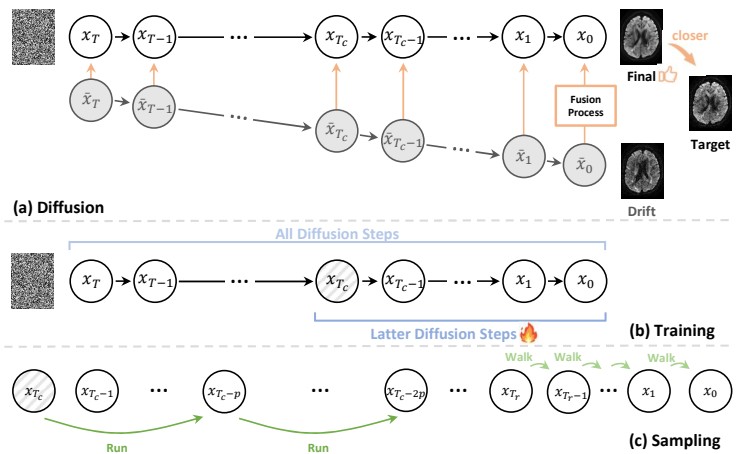

Figure 1: (a) Fusion process (Section 3.1) aligns $\{\bar{x}_t\}_1^T$ to $\{x_t\}_1^T$ and avoids drift ("Drift" means drifted results, "Final" means the denoised version of "Target"); (b) Training the latter diffusion steps (Section 3.2) imposes restrictions on the generation ability of diffusion models and decreases uncertainty; (c) *Run-Walk* accelerated sampling (Section 3.3) accelerates the entire sampling process.

larger proportion of $x_{t-1}$ aligns closer to $x$, rather than merely being a noisy version of $x'$. If we still feed $x_{t-1}$ and $t-1$ into $\mathcal{F}_\theta$, it will cause output deviations, which accumulate in the sampling chain and ultimately lead to the drift problem (Fig. 1 (a)).

**Fusion process (Q1)**  Since $\mathcal{F}_\theta$ learns the mapping from $x'$ to $x$, $\{x_t\}_1^T$ should be combinations of $x$ and $x'$, augmented with a sampled noise $z \sim \mathcal{N}(0, I)$. These combinations can be approximated by utilizing the reverse process in DDPM to compute the linear interpolation between $x'$ to $x$:

$$x_t^* = \lambda_1^t x + \lambda_2^t x', \tag{6}$$

$$q\left(x_t | x_t^*\right) := \mathcal{N}\left(x_t; \sqrt{\bar{\alpha}_t} x_t^*, (1 - \bar{\alpha}_t) I\right), \tag{7}$$

where we rewrite $\lambda_1^t = \frac{\sqrt{\bar{\alpha}_{t-1}}\beta_t}{1-\bar{\alpha}_t}$ and $\lambda_2^t = \frac{\sqrt{\alpha_t}(1-\bar{\alpha}_{t-1})}{1-\bar{\alpha}_t}$ for simplification. As $t$ decreases, $x_t^*$ becomes closer to $x$ since $\lambda_1^t$ has a higher value. By substituting $x_t^*$ for $x'$ in Eq. (5), the Fusion process can be achieved, which obtain $x_t^*$ with different $t$ as shown in Fig. S19. Intuitively, the Fusion process gradually introduces the target denoising slice $x$ to the model, guiding the model to optimize in a fixed direction, thereby mitigating the drift. We thereby address **Q1** by defining the forward process $q\left(x_t | x_t^*\right)$.

**Q2**  Approximating noise as $z$ is definitely a feasible approach. However, the noise distribution in the real world often exhibits complex statistical properties, and thus cannot be easily captured mathematically (Section 2.1). Similar challenges also exist in dMRI.

**"Di-" process (Q2)**  To better characterize real-world noise, we represent the noise distribution involving the input noisy data. Since $x$ and $x'$ are independent corrupted measurements of the redundant part $y$ and have independent noise, directly calculating $x - x'$ leaves some linear combinations of noise ($x = y + n_1$, $x' = y + n_2$, $x - x' = n_1 - n_2$, here $n_1$ and $n_2$ represent the noise in $x$ and $x'$, respectively), we perform a zero-mean operation on these linear combinations of noise to comply with the zero-mean constraint of $z$:

$$\xi_{x-x'} = mess\left((x - x') - \mu_{x-x'}\right), \quad \mu_{x-x'} = \frac{\sum_{m=1}^{w}\sum_{n=1}^{h}(x_{mn} - x'_{mn})}{w \cdot h}, \tag{8}$$

where $mess(\cdot)$ means spatial shuffling operation originated from DDM2 (Xiang et al., 2023), $\mu_{x-x'}$ is the mean of $x - x'$. $\xi_{x-x'}$ theoretically preserves the variance information of the noise (See Appendix C.2 for proof) and will serve as the noise distribution employed in both $q\left(x_t | x_t^*\right)$ and $p_\mathcal{F}\left(x_{t-1} | x_t\right)$. In this case, the forward process and reverse process no longer follow a Gaussian distribution, but they can be represented as Eq. (18) and Eq. (10), respectively. In Fig. S20, we demonstrate through experiments that $\xi_{x-x'}$ has different statistical properties from $z$. In Fig. S27, we show the impact of $\xi_{x-x'}$ and $z$ on the reverse process.

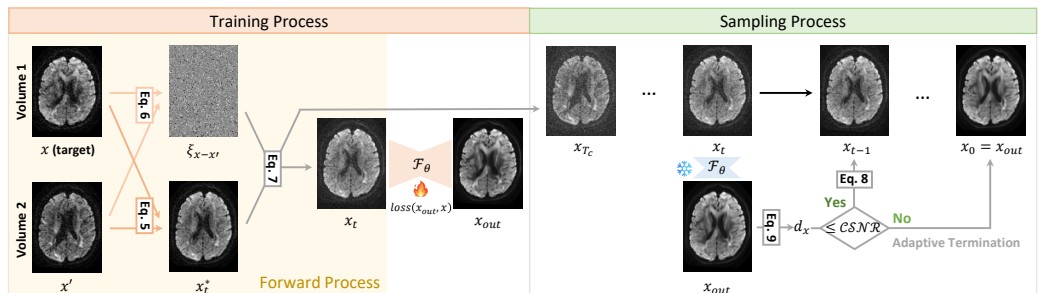

Figure 2: Overview of our single-stage Di-Fusion. The training process does not involve any extra model training apart from $\mathcal{F}_\theta$, and the sampling process offers adaptive and controllable results.

## 3.2 TRAINING PROCESS

**$\mathcal{J}$-Invariance optimization (Q3)**  When training $\mathcal{F}_\theta$, we first consider $x$ and $x'$ as $\mathcal{J} = \{x, x'\}$. Assuming that the noise distributions of $x$ and $x'$ are mutually independent, the model with $x'$ as input and $x$ as the optimization target satisfies the property of input-output independence. According to the *Proposition 1* declared in Noise2Self (Batson & Royer, 2019), the loss between $\mathcal{F}_\theta(x')$ and $x$ will in expectation equal to the loss between $\mathcal{F}_\theta(x')$ and clean ground truth $y$, plus a constant $\mathbb{E}\|x - y\|^2$ (Eq. (1)). Therefore, minimizing $\mathbb{E}\|\mathcal{F}_\theta(x') - x\|^2$ is equivalent to minimizing $\mathbb{E}\|\mathcal{F}_\theta(x') - y\|^2$ with respect to the clean ground truth $y$ and our simplified training objective is:

$$L_{\text{simple}}(\theta) := \mathbb{E}_{t, x_t^*, \xi_{x-x'}} \left[ \left\| x - \mathcal{F}_\theta(\sqrt{\bar{\alpha}_t} x_t^* + \sqrt{1 - \bar{\alpha}_t} \xi_{x-x'}, t) \right\|^2 \right] \tag{9}$$

**Intuition of training the latter diffusion steps (Q4)**  In DDPM, it is shown that when conditioned on the same latent, the samples share high-level attributes (when conditioned on say $x_{250}$, the samples are close to each other) (Ho et al., 2020). It is because of the thorough training in the former diffusion steps ($x_T \to x_{T_c}$) that DDPM possesses diverse generative capabilities. Since we perform an image denoising task with such a strong prior (from one noisy volume to another noisy volume), training only the latter diffusion steps is possible to reduce the diverse generative capabilities of $\mathcal{F}_\theta$. More precisely, only the last $T_c$ steps in the Markov chain ($x_{T_c} \to x_0$) are trained. In this way, a generative training task is simplified into a conditional generation task ($x_{T_c} \to x_0, T_c \leq T$), with more $x_0$ information provided in $\{x_t\}_1^{T_c}$ (Fig. 1 (b)).

There are two main reasons for adopting this strategy. Firstly, training the latter diffusion steps weakens the generation capacity of the diffusion model, reducing its diversity. This, in turn, lowers the uncertainty in denoising results for our task. Secondly, with the same training time, obtaining a more stable $\mathcal{F}_\theta$ is possible. By training only the latter diffusion steps, each step receives more training iterations, resulting in improved stability for the model performance. Algorithm 1 outlines the training process, and Fig. 2 (left) provides an overview of the entire training process.

---

**Algorithm 1** Training process

---

Initialize $\mathcal{F}_\theta$ randomly; input 4D data: $X \in \mathbb{R}^{w \times h \times d \times l}$
**repeat**
    $t \sim \text{Uniform}(\{1, \cdots, T_c\})$                      ▷ training the latter diffusion steps in Section 3.2
    $x = X_{*,*,i,j}, x' = X_{*,*,i,j-1}$                   ▷ $i$: slice index, $j$: volume index
    $\xi_{x-x'} = mess\left((x - x') - \mu_{x-x'}\right)$                     ▷ Eq. (8)
    $x_t^* = \lambda_1^t x + \lambda_2^t x'$                                    ▷ Eq. (6)
    take gradient descent step on: $\nabla_\theta \left\| x - \mathcal{F}_\theta\left(\sqrt{\bar{\alpha}_t} x_t^* + \sqrt{1 - \bar{\alpha}_t} \xi_{x-x'}, t\right) \right\|^2$     ▷ Eq. (9)
    resample $i$ and $j$
**until** converged

---

## 3.3 SAMPLING PROCESS

We make two specific modifications on $p_\mathcal{F}(x_{t-1}|x_t)$ to achieve an adaptive sampling process and directly begin the sampling process at $x_{T_c}$. See Fig. 2 (right) for an overview of the entire sampling process and Algorithm 2 for a detailed description of the complete sampling process.

**_Run-Walk_ accelerated sampling**    After substituting the standard normal distribution in Eq. (3) with $\xi_{x-x'}$, a typical reverse process $p_{\mathcal{F}}(x_{t-1}|x_t)$ could be formulated as:

$$p_{\mathcal{F}}(x_{t-1}|x_t) \rightarrow x_{t-1} = \lambda_1^t \mathcal{F}_\theta(x_t, t) + \lambda_2^t x_t + (\sigma_t \cdot \eta)\xi_{x-x'}, \tag{10}$$

where $\eta$ is a constant. DDIM (Song et al., 2020a) notes a special case when $\sigma_t = 0$ for all $t$ [1]; the forward process is deterministic given $x_{t-1}$ and $x_t^*$ except for $t = 1$; in the sampling process, the coefficient before the noise $\xi_{x-x'}$ becomes zero, resulting in an implicit probabilistic model (Mohamed & Lakshminarayanan, 2016). However, we do not follow the uniform step strategy of DDIM in the sampling process; instead, we use _Run-Walk_ accelerated sampling. Consider a DDPM sampling process from $x_{T_c}$ to $x_0$, when $t$ is large ($t > T_r, 1 \leq T_r \leq T_c$), the speed during each reverse process is slow; thus, acceleration can be applied (_Run_). Conversely, when $t$ is small ($t < T_r$), the speed is fast, and deceleration is required (_Walk_). In equation form, the difference between $x_{t-1}$ and $x_t$ can be represented as (See Appendix C.3 for additional derivations):

$$x_{t-1} - x_t = \underbrace{\lambda_1^t(x - x_t)}_{\text{speed}} + \underbrace{\lambda_1^t \epsilon_t}_{\text{perturbation}}. \tag{11}$$

When $t$ is large (e.g. $t > T_r$), $\lambda_1^t$ approaches zero and the speed ($\lambda_1^t(x - x_t)$) towards $x_0$ is relatively slow. This is when we perform accelerated sampling. When reaching the latter sampling process, $\lambda_1^t$ progressively increases and the speed towards $x_0$ is quite fast. This is when we stop accelerating. When $T_r = 1$, _Run-Walk_ accelerated sampling degenerates into DDIM sampling. When $T_r = T_c$, _Run-Walk_ accelerated sampling degenerates into DDPM sampling.

Now let us consider the forward process as defined not on all $\{x_t\}_1^{T_c}$, but on a subset $\{x_{\tau_1}, \ldots, x_{\tau_S}\}$, where $\tau$ is an increasing sub-sequence of $[1, \ldots, T_c]$ of length $S$. In particular, we define the sequential forward process over $x_{\tau_1}, \ldots, x_{\tau_S}$ ($x_{\tau_k} = \sqrt{\bar{\alpha}_{\tau_k}}(\lambda_1^{\tau_k} x + \lambda_2^{\tau_k} x') + \sqrt{1 - \bar{\alpha}_{\tau_k}}\xi_{x-x'}$, $1 \leq k \leq S$). The sampling process now samples according to reversed($\tau$) (In practice, $\tau = \{1, 2, \cdots, T_r - 1, T_r, T_r + p, \cdots, T_c - p, T_c\}$, where $p$ is an integer representing the acceleration factor). This can be more intuitively understood in Fig. 1 (c).

---

**Algorithm 2** Sampling process

Load pre-trained $\mathcal{F}_\theta$; input: $X \in \mathbb{R}^{w \times h \times d \times l}$, $i$, $j$ and $\mathcal{CSNR}$
$x = X_{*,*,i,j}, x' = X_{*,*,i,j-1}$             ▷ $i$: slice index, $j$: volume index
$\xi_{x-x'} = mess\left((x - x') - \mu_{x-x'}\right)$            ▷ Eq. (8)
$x_{T_c} = \sqrt{\bar{\alpha}_{T_c}}(\lambda_1^{T_c} x + \lambda_2^{T_c} x') + \sqrt{1 - \bar{\alpha}_{T_c}}\xi_{x-x'}$      ▷ Eq. (7)
$b_x = \frac{\sum_{m=1}^w \sum_{n=1}^h 1}{2 \cdot \sum_{m=1}^w \sum_{n=1}^h \mathbb{I}_{(x_{mn} > \rho_1)}} + \frac{\sum_{m=1}^w \sum_{n=1}^h 1}{2 \cdot \sum_{m=1}^w \sum_{n=1}^h \mathbb{I}_{(x_{mn} > \rho_2)}}$    ▷ Eq. (12)
**for** $\tau_k$ = reversed $\{1, 2, \cdots, T_r - 1, T_r, T_r + p, \cdots, T_c - p, T_c\}$ **do**
     $\xi_{x-x'} = mess(\xi_{x-x'})$               ▷ Shuffle $\xi_{x-x'}$ again
     $x_{out} = \mathcal{F}_\theta(x_{\tau_k}, \tau_k)$             ▷ Eq. (4)
     $d_x = \|x - x_{out}\|^2 \times b_x$            ▷ Eq. (13)
     **if** $d_x > \mathcal{CSNR}$ **then**
         $x_0 = x_{out}$; **break**           ▷ In Section 3.3
     **else**
         $x_{\tau_{k-1}} = \lambda_1^{\tau_k} x_{out} + \lambda_2^{\tau_k} x_{\tau_k} + (\sigma_{\tau_k} \cdot \eta)\xi_{x-x'}$    ▷ Eq. (10)
     **end if**
**end for**
**return** $x_0$

---

**Towards iterative and controllable refinement (Q5)**    During our experiments, we observe that the intermediate outputs, $x_{out}$, obtained during the sampling process demonstrate a substantial success in denoising. Therefore, we explore the feasibility of adaptive termination to stop sampling prematurely. More specifically, the degree of denoising in $x_{out}$ can be characterized by its distance from $x$. Nevertheless, directly computing this distance $\|x - x_{out}\|^2$ presents a problem. When the slice index $i$ is located at the edges, the resulting distance tends to be smaller due to the reduced

---

[1] We do this by multiplying $\sigma_t \xi_{x-x'}$ with $\eta$, where $\eta = 0$ if no special instructions are provided.

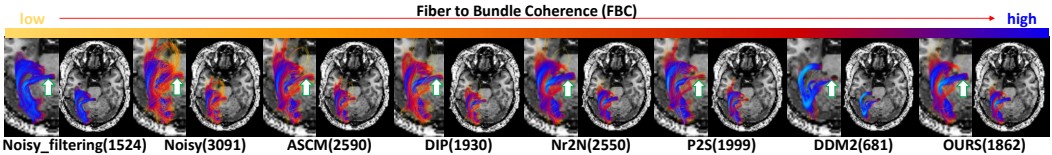

Figure 3: Density map of FBC projected on the streamlines of the OR bundles. The numbers in parentheses represent the number of streamlines. Di-Fusion generates the minimal number of streamlines while maintaining high FBCs (consider "Noisy_filtering" as references for high FBCs).

amount of brain tissue in these edge slices. Hence, it would be preferable to calculate a coefficient $b_x$ that accounts for the ratio of brain tissue to the entire image. Here, we adopt a simple definition:

$$b_x = \frac{\sum_{m=1}^{w}\sum_{n=1}^{h} 1}{2 \cdot \sum_{m=1}^{w}\sum_{n=1}^{h} \mathbb{I}_{(x_{mn} > \rho_1)}} + \frac{\sum_{m=1}^{w}\sum_{n=1}^{h} 1}{2 \cdot \sum_{m=1}^{w}\sum_{n=1}^{h} \mathbb{I}_{(x_{mn} > \rho_2)}}, \tag{12}$$

where $\rho_1$ and $\rho_2$ are constants depending on the data normalization methods employed[2] and $\mathbb{I}(\cdot)$ is an indicator function. $b_x$ can be used to correct $d_x$:

$$d_x = \|x - x_{out}\|^2 \times b_x. \tag{13}$$

Since $d_x$ has been corrected, we can pre-define a universal value $\mathcal{CSNR}$ to perform the iterative and controllable refinement on each slice. During $p_{\mathcal{F}}(x_{t-1}|x_t)$, we first get $x_{out} = \mathcal{F}_\theta(x_t, t)$ and compute $d_x$ (Eq. (13)). Then if $d_x$ is greater than $\mathcal{CSNR}$, $x_0 = x_{out}$ and the refinement iteration breaks. In contrast, the refinement iteration continues if $d_x$ is smaller than $\mathcal{CSNR}$. In extreme cases, when $\mathcal{CSNR} = 0$, the reverse process will immediately terminate and output $x_0$. When $\mathcal{CSNR} = 1$, the complete reversed($\tau$) will be executed until completion.

## 4 EXPERIMENTS

### 4.1 DATASETS AND COMPETING METHODS

**Datasets** To thoroughly evaluate Di-Fusion, we perform experiments on three publicly available brain dMRI datasets acquired using different, commonly-used acquisition schemes: *(i)* High-Angular Resolution Diffusion Imaging (Stanford HARDI, $X \in \mathbb{R}^{106 \times 81 \times 76 \times 150}$ (Rokem, 2016)); *(ii)* Multi-Shell (Sherbrooke 3-Shell dataset, $X \in \mathbb{R}^{128 \times 128 \times 64 \times 193}$ (Garyfallidis et al., 2014)); *(iii)* Single-Shell (Parkinson's Progression Markers Initiative (PPMI) dataset, $X \in \mathbb{R}^{116 \times 116 \times 72 \times 64}$ (Marek et al., 2011)). Simulated experiments are carried out on the fastMRI datasets (Tibrewala et al., 2023; Zbontar et al., 2018). We simulate noisy data with five different noise intensities.

**Competing methods** We compare Di-Fusion with five competing methods in the main paper (all experimental details are provided in Appendix D.1): *(i)* Adaptive Soft Coefficient Matching (ASCM), an improved extension of non-local means denoising (Coupé et al., 2012). *(ii)* Deep Image Prior (DIP), a self-supervised denoising method (Ulyanov et al., 2018). *(iii)* Noisier2Noise (Nr2N), a statistic-based denoising method (Moran et al., 2020). *(iv)* Patch2Self (P2S), a multi-volume denoising method (Fadnavis et al., 2020a). *(v)* DDM2, state-of-the-art denoising method (Xiang et al., 2023). More comparisons with other denoising methods, including MPPCA (Veraart et al., 2016), Noise2Score (Kim & Ye, 2021), Recorrupted2Recorrupted (Pang et al., 2021), and Patch2Self2 (Fadnavis et al., 2024), can be found in Appendix E.

### 4.2 IMPACTS ON DOWNSTREAM CLINICAL TASKS

**Effect on tractography** The noise in dMRI can impact tractography results, potentially causing the generation of spurious streamlines by the tracking algorithm (Fadnavis et al., 2020a; Garyfallidis et al., 2014; Schilling et al., 2019). We explore the effect of denoising on probabilistic tracking (Girard et al., 2014) by employing the Fiber Bundle Coherency (FBC) metric (Portegies et al., 2015) and reconstruct the optic radiation (OR) bundles (See Appendix D.2 for details). Since low FBCs indicate which fibers are poorly aligned with their neighbors, we further clean the tractography results of noisy data (captioned by "Noisy_filtering") using a stopping criterion (Meesters et al., 2016). In Fig. 3, we

---

[2]In our experiments, $\rho_1 = -0.93$ and $\rho_2 = -0.95$, changing their values has little impact on the results.

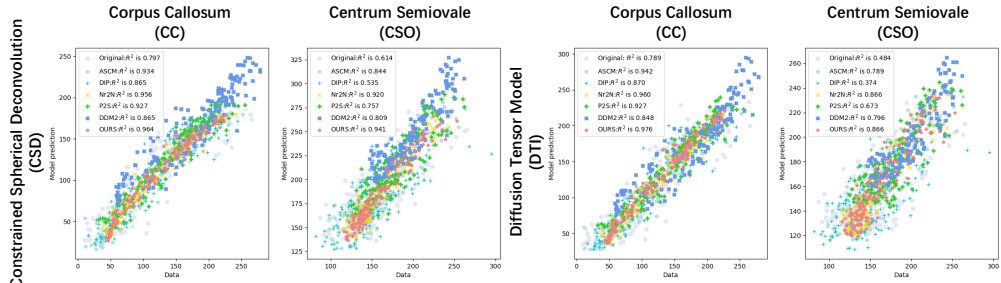

Figure 4: Scatter plots of the microstructure model predictions against input data. The top-left of each plot shows the quantitative $R^2$ metric computed from each model fit on the corresponding data. Our data points are more concentrated (higher $R^2$).

show the effect on the tractography of OR. Although DDM2 yields the fewest streamlines, noticeably, it misses the high FBCs indicated by the white arrow in Fig. 3. *Di-Fusion generates the minimal number of streamlines while maintaining high FBCs*, which indicates that our method maximizes the denoising performance while preserving fiber bundle information.

**Effect on microstructure model fitting**   Denoising methods can be compared based on their accuracy in fitting the diffusion signal (Ades-Aron et al., 2018). We apply two commonly used microstructure fitting models, namely diffusion tensor model (DTI) (Basser et al., 1994) and Constrained Spherical Deconvolution (CSD) (Tournier et al., 2007), on noisy and denoised data (Appendix D.2 for details). We show the quantitative $R^2$ metric of microstructure predictions against the original data for Corpus Callosum (CC) and Centrum SemiOvale (CSO) in Table S2 and the corresponding scatter plots are in Fig. 4. *As measured by $R^2$, Di-Fusion achieves the best results across all four different settings*. This means that Di-Fusion aids in the characterization of the microstructure.

**Effect on diffusion signal estimates**   We further examine how the denoising quality translates to creating quantitative and clinically-relevant DTI (Basser et al., 1994) diffusion signal estimates (Details are in Appendix D.2). In Fig. S9, we show fractional anisotropy, axial diffusivity, mean diffusivity, and radial diffusivity comparisons. *Our method effectively suppresses noise and reconstructs fiber tracts*.

### 4.3   QUANTITATIVE AND QUALITATIVE RESULTS ON *in-vivo* DATA

**Quantitative results on SNR/CNR metrics**   Considering the infeasibility of using metrics that need clean ground truth and their limited correlation with clinical utility (Mason et al., 2019), computing metrics in downstream clinical regions of interest is more reasonable (Adamson et al., 2021). To quantify the denoising performance, we employ Signal-to-Noise Ratio (SNR) and Contrast-to-Noise Ratio (CNR) metrics (Details are in Appendix D.3). The quantitative denoising results are reported as mean and standard deviation scores for the complete 4D volumes in Fig S12. *Di-Fusion indicates better performance in terms of SNR/CNR metrics*.

**Qualitative results**   In Fig. 5, we visualize the denoising results and residuals on axial slices for Stanford HARDI (Fig. S13, S14 and Fig. S15 for more qualitative results). From the residuals of Fig. 5, the area indicated by the red arrow does not appear in "OURS", indicating that Di-Fusion does not remove any anatomical structure during denoising. All pictures are best viewed when zoomed in.

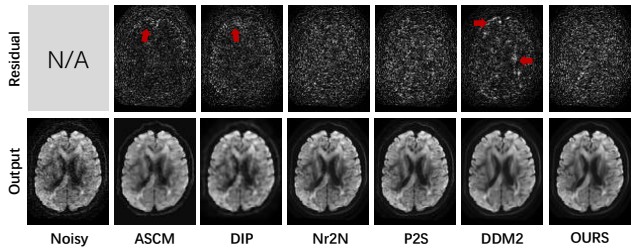

Figure 5: Qualitative results. "OURS" results are obtained when $\mathcal{CSNR} = 0.040$. The area indicated by the red arrow does not appear in "OURS", indicating that Di-Fusion does not remove structural information during denoising.

### 4.4 QUANTITATIVE RESULTS ON SIMULATED DATA

We show the PSNR and SSIM metrics [3] in Table 1 (Implementation details are in Appendix D.4, see Fig. S16 for qualitative results). For Nr2N, DDM2, and Di-Fusion, three rounds of experiments are conducted to provide error bars. As P2S utilizes linear regressors, each round's results are consistently similar. Hence, P2S's error bars are not provided. When the noise intensity is high, our method performs the best and shows stable performance. Di-Fusion *holds a tremendous potential for generalization and applicability for its stable performance and better performance under high noise intensity*. Moreover, these results provide evidence that *Di-Fusion could be extended to self-supervised MRI denoising without relying on any clean data*.

Table 1: Quantitative results on simulated data (Noisy means simulated data). PSNR (dB) and SSIM (%) are reported. Numbers are presented as mean value with standard deviation. Di-Fusion exhibits more stable performance with a smaller standard deviation.

|  | Simulation 1 | | Simulation 2 | | Simulation 3 | | Simulation 4 | | Simulation 5 | |
|---|---|---|---|---|---|---|---|---|---|---|
| Method | SSIM | PSNR | SSIM | PSNR | SSIM | PSNR | SSIM | PSNR | SSIM | PSNR |
| Noisy | 11.38 | 13.72 | 20.65 | 16.09 | 37.41 | 20.38 | 52.02 | 23.76 | 64.62 | 26.63 |
| P2S | 24.53 | 11.20 | 43.94 | 17.63 | 65.34 | 24.91 | 78.61 | **30.26** | 86.13 | **33.87** |
| Nr2N | $23.72_{0.93}$ | $16.78_{0.22}$ | $50.83_{2.27}$ | $22.13_{0.94}$ | $71.77_{6.94}$ | $26.80_{1.69}$ | $70.32_{7.99}$ | $26.40_{3.17}$ | $87.31_{4.70}$ | $32.82_{0.72}$ |
| DDM2 | $26.02_{2.63}$ | $17.92_{1.06}$ | $49.57_{15.6}$ | $21.76_{1.64}$ | $59.64_{8.47}$ | $23.25_{4.05}$ | $77.96_{2.16}$ | $29.14_{1.01}$ | $81.43_{2.23}$ | $31.35_{2.23}$ |
| OURS | $\mathbf{39.05}_{1.42}$ | $\mathbf{19.91}_{0.32}$ | $\mathbf{62.02}_{1.11}$ | $\mathbf{23.60}_{0.05}$ | $\mathbf{77.36}_{0.61}$ | $\mathbf{26.96}_{0.37}$ | $\mathbf{83.43}_{1.39}$ | $28.69_{0.67}$ | $\mathbf{89.52}_{0.18}$ | $30.63_{0.27}$ |

### 4.5 ABLATION STUDIES

Without the **Fusion process**, the results in the early sampling phase do not deviate significantly (Fig. S24), but when we do not perform an adaptive termination, the results show areas that are absent in the noisy data (Fig. S25), indicating that they have indeed drifted; Without the **"Di-" process**, the results lack some high-frequency information, and the overall gray value of the denoised images has also changed (Case 1 in Fig. S24), In Fig. S20, we demonstrate through experiments that the noise computed by "Di-" process has different statistical properties from $z$; Without **training the latter diffusion steps**, the denoising results are noticeably smoother and have more hallucinations (Fig. S24 and Fig. S25). The details of the above ablation studies can be found in Appendix G.1.

Furthermore, we balance the training epochs for different $T_c$[4] and show $R^2$ of microstructure model fitting results in Fig. 6. The results show that choosing any $T_c$ within a reasonable range ($T_c < 500$) will not have a significant influence on the denoising results and the training difficulty of every step is relatively consistent.

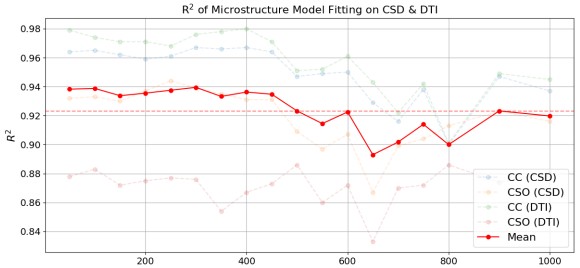

Figure 6: $R^2$ of microstructure model fitting on CSD & DTI obtained when $T_c$ is different. When $T_c < 500$, the performance is consistent.

During the reverse process, *Run-Walk* accelerated sampling not only enables accelerated sampling, but also ensures that the sampling quality remains mainly unchanged (Fig. S26). In Table S6, the sampling time indicates that the adaptive termination and *Run-Walk* accelerated sampling (Section 3.3) together greatly reduce the sampling time. The details of the above ablation studies can be found in Appendix G.2.

## 5 DISCUSSIONS

**On comparisons with related work** In Appendix A, we discuss the differences between Di-Fusion and Patch2Self, as well as DDM2. In essence, Di-Fusion surpasses Patch2Self by *significantly*

---

[3]We have clean ground truth in simulation settings.

[4]$1 \times 10^5$ epochs for $T_c = 300$, $2 \times 10^5$ epochs for $T_c = 600$, etc.

*reducing the dependence on the number of input volumes*, thereby expanding its scope of application. In comparison to DDM2, Di-Fusion not only *simplifies the method to a single-stage framework* but also implements *iterative and controllable refinement through methods mentioned in* Section 3. Moreover, in Section 4, Appendix E and H, Di-Fusion achieves state-of-the-art performance in the conducted experiments, showcasing its superior and stable results.

**On limitations**   *(i)* Possible longer inference duration. The inference time of diffusion models is already relatively long, and there are concerns that the additional computation introduced by the adaptive termination in Section 3.3 may further increase the inference duration. In Appendix G.2, we discuss the sampling burden associated with Di-Fusion. *(ii)* Possible hallucinations. As a class of generative models, diffusion models inevitably raise concerns regarding generating fake anatomical details (hallucinations). However, we find that with the methods introduced in Section 3, particularly the training the latter diffusion steps mentioned in Section 3.2, the generative capacity of diffusion models can be restricted (Appendix G.1), which helps reduce hallucinations. *(iii)* Noise-artifact conflation and additive noise assumptions. Our method explicitly targets thermal noise modeled as additive Gaussian processes, consistent with common denoising frameworks (Chen et al., 2019; Ramos-Llordén et al., 2021; Cordero-Grande et al., 2019). However, this intentionally excludes spatially varying distortions (e.g., cardiac pulsation) often classified as "physiological noise" (Chang et al., 2005; 2012; Walker et al., 2011) but better characterized as artifacts. While our assumption enables tractable solutions, it may limit effectiveness for signal-dependent noise.

## 6    CONCLUSION

This paper proposes Di-Fusion, an end-to-end self-supervised MRI denoising method that achieves iterative and stable refinement without relying on extra noise model training or clean data. The Fusion process aligns the trajectory of the forward process and avoids drifted results. The "Di-" process characterizes real-world noise, enabling the model to capture statistical properties of the real-world noise. By training the latter diffusion steps, our model achieves enhanced stability and performance. During the inference stage, Di-Fusion offers controllable results through an adaptive sampling process. Comprehensive experiments on real and simulated data demonstrate that Di-Fusion achieves state-of-the-art performance in microstructure modeling, tractography tracking, and other downstream tasks.

ACKNOWLEDGMENTS

This research was supported by Natural Science Foundation of China under Grant 62271465, Suzhou Basic Research Program under Grant SYG202338, and Open Fund Project of Guangdong Academy of Medical Sciences, China (No. YKY-KF202206).

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

## A    COMPARISONS WITH RELATED WORKS

**Comparison with Patch2Self**    Patch2Self (Fadnavis et al., 2020a) requires a minimum of ten additional diffusion vector volumes to denoise a single diffusion vector volume. Instead, our work only needs one additional volume, which is clinically meaningful as common clinical dMRI often scans fewer than ten diffusion vector volumes (Karayumak et al., 2019; Xiang et al., 2023). Moreover, our model does not require repetitive training, whereas Patch2Self necessitates training multiple regressors to perform voxel-by-voxel denoising. However, despite achieving better results, Di-Fusion takes relatively longer time than Patch2Self, which is due to the training time of the diffusion models.

**Comparison with DDM2**   Our work only requires a single stage for denoising, whereas DDM2 typically involves three stages. Furthermore, it's worth noting that the noise model in the first stage of DDM2 critically influences the ultimate denoising results, and finding an optimal solution that simultaneously maximizes evaluation metrics scores and minimizes training time can be challenging (See Fig. S32 for details).

## B  DI-FUSION

### B.1  FORWARD PROCESS

Consider $x = X_{*,*,i,j}$ ($i$: slice index, $j$: volume index) as the target slice to denoise, $x' = X_{*,*,i,j-1}$. $\beta_{1,\cdots,T}$ is a pre-defined noise schedule, $\sigma_t^2 := \beta_t$, $\alpha_t := 1 - \beta_t$ and $\bar{\alpha}_t := \prod_{s=1}^{t} \alpha_s$. We rewrite $\lambda_1^t = \frac{\sqrt{\bar{\alpha}_{t-1}}\beta_t}{1-\bar{\alpha}_t}$ and $\lambda_2^t = \frac{\sqrt{\alpha_t}(1-\bar{\alpha}_{t-1})}{1-\bar{\alpha}_t}$ for simplification.

Perform the Fusion process:
$$x_t^* = \lambda_1^t x + \lambda_2^t x'. \tag{14}$$

Then we get a linear interpolation between $x$ and $x'$, we compute $x_t$ based on $q\left(x_t|x_t^*\right)$:

$$x_t = \sqrt{\bar{\alpha}_t}x_t^* + \sqrt{1-\bar{\alpha}_t}z. \tag{15}$$

The forward process can be defined if using $z \sim \mathcal{N}\left(\mathbf{0}, \mathbf{I}\right)$ for perturbing data distribution:

$$q\left(x_t|x_t^*\right) := \mathcal{N}\left(x_t; \sqrt{\bar{\alpha}_t}x_t^*, (1-\bar{\alpha}_t)\mathbf{I}\right). \tag{16}$$

However, we use "Di-" process to compute a noise $\xi_{x-x'}$ to substitute for $z$:

$$\xi_{x-x'} = mess\left((x-x') - \mu_{x-x'}\right), \quad \mu_{x-x'} = \frac{\sum_{m=1}^{w}\sum_{n=1}^{h}(x_{mn} - x'_{mn})}{w \cdot h}. \tag{17}$$

So the forward process can't be represented as $\mathcal{N}\left(x_t; \sqrt{\bar{\alpha}_t}x_t^*, (1-\bar{\alpha}_t)\mathbf{I}\right)$, but could be computed using the following formula:

$$q\left(x_t|x_t^*\right) \rightarrow x_t = \sqrt{\bar{\alpha}_t}x_t^* + \sqrt{1-\bar{\alpha}_t}\xi_{x-x'}. \tag{18}$$

We leverage a dynamic combination (the Fusion process) and continuously varying noise (the "Di-" process) to provide the model with more augmented training data, thereby enhancing its robustness. This idea is similar to those in Noise2Void (Krull et al., 2019), Noisier2Noise (Moran et al., 2020), and Noisy-as-Clean (Xu et al., 2020), which also utilize data augmentation to construct training data.

### B.2  TRAINING PROCESS

Our simplified training objective is:

$$L_{\text{simple}}(\theta) := \mathbb{E}_{t,x_t^*,\xi_{x-x'}}\left[\left\|x - \mathcal{F}_\theta(\sqrt{\bar{\alpha}_t}x_t^* + \sqrt{1-\bar{\alpha}_t}\xi_{x-x'}, t)\right\|^2\right]. \tag{19}$$

We perform training the latter diffusion steps by sample $t \sim \text{Uniform}\left(\{1, \cdots, T_c\}\right)$.

### B.3  REVERSE PROCESS

The details of how to perform the reverse process in DDPM if a data predictor is used are in Appendix C.1. If it is a data predictor $\mathcal{F}_\theta$ that directly predict $x_0$, the reverse process for DDPM becomes:

$$x_{t-1} = \lambda_1^t \mathcal{F}_\theta\left(x_t, t\right) + \lambda_2^t x_t + (\sigma_t \cdot \eta)\xi_{x-x'}. \tag{20}$$

And $p_{\mathcal{F}}\left(x_{t-1}|x_t\right)$ can be defined as:

$$p_{\mathcal{F}}\left(x_{t-1}|x_t\right) \rightarrow x_{t-1} = \lambda_1^t \mathcal{F}_\theta\left(x_t, t\right) + \lambda_2^t x_t + (\sigma_t \cdot \eta)\xi_{x-x'}. \tag{21}$$

Now let us consider the forward process as defined not on all $\{x_t\}_1^{T_c}$, but on a subset $\{x_{\tau_1}, \ldots, x_{\tau_S}\}$, where $\tau$ is an increasing sub-sequence of $[1, \ldots, T_c]$ of length $S$. In particular, we define the sequential forward process over $x_{\tau_1}, \ldots, x_{\tau_S}$ ($x_{\tau_k} = \sqrt{\bar{\alpha}_{\tau_k}} (\lambda_1^{\tau_k} x + \lambda_2^{\tau_k} x') + \sqrt{1 - \bar{\alpha}_{\tau_k}} \xi_{x-x'}$, $1 \leq k \leq S$).

The *Run-Walk* accelerated sampling now sample according to reversed($\tau$) (In practice, $\tau = \{1, 2, \cdots, T_r - 1, T_r, T_r + p, \cdots, T_c - p, T_c\}$), then the reverse process become:

$$p_{\mathcal{F}_\theta} \left( x_{\tau_{k-1}} | x_{\tau_k} \right) \to x_{\tau_{k-1}} = \lambda_1^{\tau_k} \mathcal{F}_\theta \left( x_{\tau_k}, \tau_k \right) + \lambda_2^{\tau_k} x_{\tau_k} + \left( \sigma_{\tau_k} \cdot \eta \right) \xi_{x-x'}. \tag{22}$$

Before sampling, we define an universal value $\mathcal{CSNR}$ and compute $b_x$:

$$b_x = \frac{\sum_{m=1}^{w} \sum_{n=1}^{h} 1}{2 \cdot \sum_{m=1}^{w} \sum_{n=1}^{h} \mathbb{I}_{(x_{mn} > \beta_1)}} + \frac{\sum_{m=1}^{w} \sum_{n=1}^{h} 1}{2 \cdot \sum_{m=1}^{w} \sum_{n=1}^{h} \mathbb{I}_{(x_{mn} > \beta_2)}}, \tag{23}$$

During every $p_{\mathcal{F}_\theta} \left( x_{\tau_{k-1}} | x_{\tau_k} \right)$, we first get $x_{out} = \mathcal{F}_\theta \left( x_{\tau_k}, \tau_k \right)$, and $d_x = \|x - x_{out}\|^2 \times b_x$.

Then if $d_x$ is greater than $\mathcal{CSNR}$, the output $x_0 = x_{out}$ and the refinement iteration breaks. In contrast, the refinement iteration continues if $d_x$ is smaller than $\mathcal{CSNR}$.

## C ADDITIONAL DERIVATIONS

### C.1 THE DIFFERENCE BETWEEN THE TWO TRAJECTORIES

The original sampling process in the *Algorithm 2* of DDPM (Ho et al., 2020) is:

$$x_{t-1} = \frac{1}{\sqrt{\alpha_t}} \left( x_t - \frac{1 - \alpha_t}{\sqrt{1 - \bar{\alpha}_t}} \epsilon_\theta(x_t, t) \right) + \sigma_t z, \tag{24}$$

where $\epsilon_\theta$ is a noise predictor. However, we use a data predictor $\mathcal{F}_\theta$ to directly predict $x_0$ in our paper. We will demonstrate how to perform the reverse process in DDPM if a data predictor is used.

Given a data point sampled from a real data distribution $x_0 \sim q(x)$, let us define a forward diffusion process in which we add small amount of Gaussian noise to the sample in $T$ steps, producing a sequence of noisy samples $x_1, \ldots, x_T$. The step sizes are governed by a variance schedule $\{\beta_t \in (0, 1)\}_{t=1}^{T}$:

$$q(x_t | x_{t-1}) = \mathcal{N}(x_t; \sqrt{1 - \beta_t} x_{t-1}, \beta_t \mathbf{I}), \quad q(x_{1:T} | x_0) = \prod_{t=1}^{T} q(x_t | x_{t-1}). \tag{25}$$

As the step $t$ increases, the data sample $x_0$ gradually loses its distinguishable features. Ultimately, when $T \to \infty$, $x_T$ converges to an isotropic Gaussian distribution.

Let $\alpha_t = 1 - \beta_t$ and $\bar{\alpha}_t = \prod_{i=1}^{t} \alpha_i$. A nice property of the aforementioned process is that we can sample $x_t$ at any arbitrary time step $t$ in closed form using the reparameterization trick:

$$\begin{aligned} x_t &= \sqrt{\alpha_t} x_{t-1} + \sqrt{1 - \alpha_t} z_{t-1} & \text{;where } z_{t-1}, z_{t-2}, \cdots \sim \mathcal{N}(\mathbf{0}, \mathbf{I}). \\ &= \sqrt{\alpha_t \alpha_{t-1}} x_{t-2} + \sqrt{1 - \alpha_t \alpha_{t-1}} \bar{z}_{t-2} & \text{;where } \bar{z}_{t-2} \text{ merges two Gaussians.} \\ &= \ldots \\ &= \sqrt{\bar{\alpha}_t} x_0 + \sqrt{1 - \bar{\alpha}_t} z, \end{aligned} \tag{26}$$

where we merge two Gaussians with different variances, $\mathcal{N}(\mathbf{0}, \sigma_1^2 \mathbf{I})$ and $\mathcal{N}(\mathbf{0}, \sigma_2^2 \mathbf{I})$, resulting in a new distribution $\mathcal{N}(\mathbf{0}, (\sigma_1^2 + \sigma_2^2) \mathbf{I})$. Here, the merged standard deviation is given by $\sqrt{(1 - \alpha_t) + \alpha_t (1 - \alpha_{t-1})} = \sqrt{1 - \alpha_t \alpha_{t-1}}$. We can then derive:

$$q(x_t | x_0) = \mathcal{N}(x_t; \sqrt{\bar{\alpha}_t} x_0, (1 - \bar{\alpha}_t) \mathbf{I}). \tag{27}$$

Consider a reverse process, it is noteworthy that the reverse conditional probability is tractable when conditioned on $x_0$:

$$q(x_{t-1}|x_t, x_0) = \mathcal{N}(x_{t-1}; \tilde{\boldsymbol{\mu}}(x_t, x_0), \sigma_t^2 \mathbf{I}). \tag{28}$$

Using Bayes' rule, we then have (Ho et al., 2020):

$$
\begin{aligned}
&q(x_{t-1}|x_t, x_0) \\
&= q(x_t|x_{t-1}, x_0) \frac{q(x_{t-1}|x_0)}{q(x_t|x_0)} \\
&\propto \exp\Big( -\frac{1}{2}\big( \frac{(x_t - \sqrt{\alpha_t}x_{t-1})^2}{\beta_t} + \frac{(x_{t-1} - \sqrt{\bar{\alpha}_{t-1}}x_0)^2}{1-\bar{\alpha}_{t-1}} - \frac{(x_t - \sqrt{\bar{\alpha}_t}x_0)^2}{1-\bar{\alpha}_t} \big) \Big) \\
&= \exp\Big( -\frac{1}{2}\big( \frac{x_t^2 - 2\sqrt{\alpha_t}x_t x_{t-1} + \alpha_t x_{t-1}^2}{\beta_t} + \frac{x_{t-1}^2 - 2\sqrt{\bar{\alpha}_{t-1}}x_0 x_{t-1} + \bar{\alpha}_{t-1}x_0^2}{1-\bar{\alpha}_{t-1}} - \frac{(x_t - \sqrt{\bar{\alpha}_t}x_0)^2}{1-\bar{\alpha}_t} \big) \Big) \\
&= \exp\Big( -\frac{1}{2}\big( (\frac{\alpha_t}{\beta_t} + \frac{1}{1-\bar{\alpha}_{t-1}})x_{t-1}^2 - (\frac{2\sqrt{\alpha_t}}{\beta_t}x_t + \frac{2\sqrt{\bar{\alpha}_{t-1}}}{1-\bar{\alpha}_{t-1}}x_0)x_{t-1} + C(x_t, x_0) \big) \Big),
\end{aligned}
\tag{29}
$$

where $C(x_t, x_0)$ is some function that does not involve $x_{t-1}$, and the details are omitted. Following the standard Gaussian density function, the mean and variance can be parameterized as follows (recall that $\alpha_t = 1 - \beta_t$ and $\bar{\alpha}_t = \prod_{i=1}^T \alpha_i$):

$$\sigma_t^2 = 1/(\frac{\alpha_t}{\beta_t} + \frac{1}{1-\bar{\alpha}_{t-1}}) = 1/(\frac{\alpha_t - \bar{\alpha}_t + \beta_t}{\beta_t(1-\bar{\alpha}_{t-1})}) = \frac{1-\bar{\alpha}_{t-1}}{1-\bar{\alpha}_t} \cdot \beta_t. \tag{30}$$

$$
\begin{aligned}
\tilde{\boldsymbol{\mu}}_t(x_t, x_0) &= (\frac{\sqrt{\alpha_t}}{\beta_t}x_t + \frac{\sqrt{\bar{\alpha}_{t-1}}}{1-\bar{\alpha}_{t-1}}x_0)/(\frac{\alpha_t}{\beta_t} + \frac{1}{1-\bar{\alpha}_{t-1}}) \\
&= (\frac{\sqrt{\alpha_t}}{\beta_t}x_t + \frac{\sqrt{\bar{\alpha}_{t-1}}}{1-\bar{\alpha}_{t-1}}x_0)\frac{1-\bar{\alpha}_{t-1}}{1-\bar{\alpha}_t} \cdot \beta_t \\
&= \frac{\sqrt{\alpha_t}(1-\bar{\alpha}_{t-1})}{1-\bar{\alpha}_t}x_t + \frac{\sqrt{\bar{\alpha}_{t-1}}\beta_t}{1-\bar{\alpha}_t}x_0.
\end{aligned}
\tag{31}
$$

Thus, if it is a data predictor $\mathcal{F}_\theta$ that directly predict $x_0$, based on $q(x_{t-1}|x_t, x_0)$, the reverse process for DDPM becomes:

$$
\begin{aligned}
x_{t-1} &= \tilde{\boldsymbol{\mu}}_t(x_t, x_0) + \sigma_t^2 z \\
&= \frac{\sqrt{\alpha_t}(1-\bar{\alpha}_{t-1})}{1-\bar{\alpha}_t}x_t + \frac{\sqrt{\bar{\alpha}_{t-1}}\beta_t}{1-\bar{\alpha}_t}\mathcal{F}_\theta(x_t, t) + \sigma_t^2 z.
\end{aligned}
\tag{32}
$$

DDPM (Ho et al., 2020) found that both $\sigma_t^2 = \beta_t$ and $\sigma_t^2 = \frac{1-\bar{\alpha}_{t-1}}{1-\bar{\alpha}_t}\beta_t$ had similar results through experiments. We set $\sigma_t^2 = \beta_{1,\cdots,T}$ and hold $\beta_{1,\cdots,T}$ as hyperparameters.

Now we know how to perform the reverse process if a data predictor is used. According to Eq. (4), we know $x + \epsilon_t = x_{out} = \mathcal{F}_\theta(x_t, t)$, then we can get:

$$x_{t-1} = \frac{\sqrt{\alpha_t}(1-\bar{\alpha}_{t-1})}{1-\bar{\alpha}_t}x_t + \frac{\sqrt{\bar{\alpha}_{t-1}}\beta_t}{1-\bar{\alpha}_t}(x + \epsilon_t) + \sigma_t^2 z, \tag{33}$$

Now let us consider directly performing the forward process ($q(x_t|x_0)$) on $x'(x = X_{*,*,i,j}, x' = X_{*,*,i,j-1})$ without the Fusion process (Eq. (6)):

$$\bar{x}_{t-1} = \sqrt{\bar{\alpha}_{t-1}}x' + \sqrt{1-\bar{\alpha}_{t-1}}z, \tag{34}$$

thus the trajectory $\{\bar{x}_t\}_1^T$ obtained by directly performing the forward process in DDPM and the trajectory $\{x_t\}_1^T$ obtained from the reverse process of DDPM are different, and the major difference is brought by $(x + \epsilon_t)$:

$$x_{t-1} = \underbrace{\frac{\sqrt{\bar{\alpha}_{t-1}}\beta_t}{1-\bar{\alpha}_t}(x+\epsilon_t)}_{\text{major difference}} + \frac{\sqrt{\alpha_t}(1-\bar{\alpha}_{t-1})}{1-\bar{\alpha}_t}x_t + \sigma_t z \neq \bar{x}_{t-1} = \sqrt{\bar{\alpha}_{t-1}}x' + \sqrt{1-\bar{\alpha}_{t-1}}z. \quad (35)$$

This is because component $\epsilon_t$ decays as $t \to 0$, then a larger proportion of components in $x_{t-1}$ becomes closer to $x$.

If we directly feed $x_{t-1}$ and $t-1$ into $\mathcal{F}_\theta$, the output would deviate slightly further from $x$. This occurs because during training, $\mathcal{F}_\theta$ is optimized only with the objective: $\|x - \mathcal{F}_\theta(\sqrt{\bar{\alpha}_{t-1}}x' + \sqrt{1-\bar{\alpha}_{t-1}}z, t-1)\|^2$ (the training objective without the Fusion process). Importantly, $x_{t-1}$ is one step closer to $x$. ($x_{t-1} = \frac{\sqrt{\bar{\alpha}_{t-1}}\beta_t}{1-\bar{\alpha}_t}(x+\epsilon_t) + \frac{\sqrt{\alpha_t}(1-\bar{\alpha}_{t-1})}{1-\bar{\alpha}_t}x_t + \sigma_t z$), rather than simply being a noisy version of $x'$. This drift accumulates over the sampling chain, ultimately leading the result to drift toward another slice.

## C.2 Variance information of noise in "Di-" process

$\xi_{x-x'}$ theoretically preserves the variance information of the noise:

$$
\begin{aligned}
\text{Var}(x-x') &= \text{Var}(y+n_1-(y+n_2)) \\
&= \text{Var}(n_1-n_2) \\
&= \text{Var}(n_1) + \text{Var}(n_2) - 2\text{Cov}(n_1,n_2) \\
&= \text{Var}(n_1) + \text{Var}(n_2),
\end{aligned}
\quad (36)
$$

where $\text{Cov}(\cdot)$ is the covariance, $\text{Var}(\cdot)$ is the variance, $\text{Cov}(n_1,n_2) = 0$ since $n_1$ and $n_1$ are independent. Assuming that $n_1$ and $n_2$ follow the same distribution, the variance information of this distribution is retained.

In Fig. S20, we show that the noise in "Di-" process has different statistical properties compared to Gaussian noise.

## C.3 Speed towards the target

The difference between $x_{t-1}$ and $x_t$ can be formulated as:

$$
\begin{aligned}
x_{t-1} - x_t &= \lambda_1^t \mathcal{F}_\theta(x_t,t) + \lambda_2^t x_t + (\sigma_t \cdot \eta)\xi_{x-x'} - x_t \\
&= \lambda_1^t x_{out} + (1-\lambda_1^t)x_t + (\sigma_t \cdot \eta)\xi_{x-x'} - x_t.
\end{aligned}
\quad (37)
$$

According to Eq. (4), we know $x + \epsilon_t = x_{out}$, then we can substitute $x_{out}$ into the Eq. (37) and get:

$$
\begin{aligned}
x_{t-1} - x_t &= \lambda_1^t x_{out} + (1-\lambda_1^t)x_t + (\sigma_t \cdot \eta)\xi_{x-x'} - x_t \\
&= \lambda_1^t(x+\epsilon_t) + (1-\lambda_1^t)x_t + (\sigma_t \cdot \eta)\xi_{x-x'} - x_t \\
&= \underbrace{\lambda_1^t(x-x_t)}_{\text{speed}} + \underbrace{(\sigma_t \cdot \eta)\xi_{x-x'}}_{\text{noise}} + \underbrace{\lambda_1^t \epsilon_t}_{\text{perturbation}}.
\end{aligned}
\quad (38)
$$

In DDIM (Song et al., 2020a), $\eta = 0$, so typically, the term "noise" disappears, leading to the following expression:

$$
x_{t-1} - x_t = \underbrace{\lambda_1^t(x-x_t)}_{\text{speed}} + \underbrace{\lambda_1^t \epsilon_t}_{\text{perturbation}}.
\quad (39)
$$

We know that $\epsilon_t$ is a perturbation term that depends on $t$, as $t \to 0$, $\epsilon_t \to 0$, $\lambda_1^t \to 1$ at the same time. So the value of "perturbation" item does not change significantly when $t$ decreases; thus difference between $x_{t-1}$ and $x_t$ are mainly caused by the "speed" item.

# D EXPERIMENTAL DETAILS

## D.1 EXPERIMENT AND REPRODUCIBILITY DETAILS

**Noise schedule**  A typical noise schedule  (Ho et al., 2020; Saharia et al., 2022b) follows a "warm-up" scheduling strategy. Inspired by DDM2, we implement a reverse "warm-up" strategy where $\beta_t$ remains at $5e^{-5}$ for the first 300 iterations and then linearly increases to $1e^2$ between $(300, 1000]$ iterations (Xiang et al., 2023).

**Training details**  Following DDPM (Ho et al., 2020), we set $\sigma_t^2 = \beta_{1,\cdots,T}$ and hold $\beta_{1,\cdots,T}$ as hyperparameters. Since we are performing a deterministic sampling process, $\eta$ in Eq. (10) is set to 0 (we talk about how $\eta$ influences the final results in Appendix F.2). We implement denoising functions $\mathcal{F}_\theta$ via U-Net (Ronneberger et al., 2015) with modifications suggested in  (Saharia et al., 2022b; Song et al., 2020b). Inspired by (Chen et al., 2020; Song & Ermon, 2019), we train $\mathcal{F}_\theta$ to condition on $\overline{\alpha}_t$, $t \sim \text{Uniform} (\{1, \cdots, T_c\})$, $T_c = 300$. Adam optimizer was used to optimize $\theta$ with a fixed learning rate of $1e^{-4}$ and a batch size of 32, and $\mathcal{F}_\theta$ was trained $1e^5$ steps from scratch. All experiments were performed on RTX GeForce 3090 GPUs in PyTorch (Paszke et al., 2019). The training duration for one $\mathcal{F}_\theta$ is approximately five hours on a single RTX GeForce 3090 GPU with 5578MB of VRAM.

**Sampling details**  During sampling, $T_r = 50$. $\beta_1 = -0.93$ and $\beta_2 = -0.95$ and changing their values has little impact on the results (We set these two factors as a simple way to extract the brain mask). $\eta = 0$ and $p = 10$ if no special instructions are provided. $\mathcal{CSNR}$ are provided in the figure caption.

**Competing methods details**  In the main paper, Di-Fusion is compared against four state-of-the-art self-supervised deep learning-based denoising methods (ASCM isn't deep learning-based). For fair comparisons DIP, Nr2N, and DDM2 adopt the architecture used in Di-Fusion. We follow the official repository[5] and use the parameters that should give the optimal denoising performance for P2S (Fadnavis et al., 2020a). *(i)* Deep Image Prior (Ulyanov et al., 2018) trains a network on a random input to target a noisy image. Thus, network parameter optimization must be performed for each image. In our experiments, we use their official repository[6] to identify the optimal training iterations on a single image and then apply the same number of iterations for denoising the entire volume. *(ii)* Noisier2Noise (Moran et al., 2020) trains a network on a noisier input to target a noisy image. In our experiments, we add additional randomly sampled noise to $x'$, and the training is performed to reconstruct the noisy image $x$ (Noise2Noise wasn't used due to its pronounced over-smoothing denoising effect in the DDM2 experiments (Xiang et al., 2023). We want to evaluate our method further by using an advanced version, Noisier2Noise). *(iii)* Patch2Self (Fadnavis et al., 2020a) generalizes Noise2Noise (Lehtinen et al., 2018) and Noise2Self (Batson & Royer, 2019) for voxel-by-voxel dMRI denoising. In our experiments, we followed their official implementation without adjusting their hyperparameters. *(iv)* DDM2 (Xiang et al., 2023) proposes a three-stage framework that integrates statistic-based denoising theory into diffusion models and performs denoising through conditional generation. In our experiments, we follow their official repository[7] without adjusting their hyperparameters.

Additionally, more comparisons with other denoising methods, including MPPCA (Veraart et al., 2016), Noise2Score (Kim & Ye, 2021), Recorrupted2Recorrupted (Pang et al., 2021), and Patch2Self2 (Fadnavis et al., 2024), are provided in Appendix E. We implemented MPPCA using the code from DIPY (Garyfallidis et al., 2014). For Noise2Score (N2S), we utilized their official repository[8]. Recorrupted2Recorrupted (R2R) was implemented using its repository[9]. For Patch2Self2 (P2S2), we directly used the denoised data provided in their supplementary material[10] (Fadnavis et al., 2024).

---

[5]https://github.com/ShreyasFadnavis/patch2self

[6]https://github.com/DmitryUlyanov/deep-image-prior

[7]https://github.com/StanfordMIMI/DDM2

[8]https://github.com/cubeyoung/Noise2Score

[9]https://github.com/PangTongyao/Recorrupted-to-Recorrupted-Unsupervised-Deep-Learning-for-Image-Denoising

[10]The denoised data is shared at https://figshare.com/s/87f6ffee972510bfda76

### D.2 DOWNSTREAM CLINICAL TASKS IMPLEMENTATION DETAILS

**On tractography**  To reconstruct white-matter pathways in the brain, one integrates orientation information of the underlying axonal bundles (streamlines) obtained by decomposing the signal in each voxel using a microstructure model (Behrens et al., 2014; Fadnavis et al., 2020a; Garyfallidis et al., 2014). Noise that corrupts the acquired DWI may impact the tractography results, leading to spurious streamlines generated by the tracking algorithm. We explore the effect of denoising on probabilistic tracking (Girard et al., 2014) by employing the Fiber Bundle Coherency (FBC) metric (Portegies et al., 2015). We first fit the data to the Constrained Spherical Deconvolution (CSD) model (Tournier et al., 2007). The fiber orientation distribution information required to perform the tracking is obtained from the Constrained Spherical Deconvolution (CSD) model fitted to the same data. The Optic Radiation (OR) is reconstructed by tracking fibers (3x3x3 voxels ROI cube, and the seed density is 6) from the calcarine sulcus (visual cortex V1) to the lateral geniculate nucleus (LGN). After the streamlines are generated, their coherency is measured with the local FBC algorithm (Portegies et al., 2015; Duits & Franken, 2011), with yellow-orange representing - spurious/incoherent fibers and red-blue representing valid/coherent fibers. Since low FBCs indicate which fibers are isolated and poorly aligned with their neighbors, we further clean the results of tractography algorithms by using the stopping criterion outlined in (Meesters et al., 2016) (the stopping criterion was only performed on noisy data's density map of FBC; thus, its results are captioned by "Noisy_filtering" and can be considered as the reference for high FBCs).

**On microstructure model fitting**  Microstructure modeling poses a complicated inverse problem and often leads to degraded parameter estimates due to the low SNR of dMRI (Novikov et al., 2018). Different denoising methods can be compared based on their accuracy in fitting the diffusion signal. We apply two commonly used diffusion microstructure models, the diffusion tensor model (DTI) (Basser et al., 1994) and Constrained Spherical Deconvolution (CSD) (Tournier et al., 2007) (DIPY (Garyfallidis et al., 2014) has made available of these two models), on raw and denoised data. DTI is a simple model that captures the local diffusion information within each voxel by modeling it in the form of a 6-parameter tensor. CSD is a more complex model using a spherical harmonic representation of the underlying fiber orientation distributions. In order to quantify the results, we perform a 3-fold cross-validation (Hastie et al., 2009) at two exemplary voxel locations, corpus callosum (CC), a single-fiber structure, and centrum semiovale (CSO), a crossing-fiber structure. The data is divided into three different subsets for the selected voxels, and data from two folds are used to fit the model, which predicts the data on the held-out fold. We quantify the goodness of fit of the models by calculating the $R^2$ score ($R^2$ metric is computed from each model fit on the corresponding data) (Fadnavis et al., 2020a).

**On diffusion signal estimates**  We examine how the denoising quality translates to downstream clinical tasks such as creating DTI (Basser et al., 1994) diffusion signal estimates using the various denoising methods. To do the comparisons, we use the volumes acquired by the first ten diffusion directions and the ten b-value=0 volumes. Before fitting the data, we perform data pre-processing. We first use the method in (Ostu, 1979) to compute a brain mask to avoid unnecessary calculations on the background of the image. Now that we have loaded and pre-processed the data we can go forward with DTI (Basser et al., 1994) fitting. We can extract the fractional anisotropy (FA), the mean diffusivity (MD), the axial diffusivity (AD) and the radial diffusivity (RD) from the DTI model.

### D.3 SNR AND CNR IMPLEMENTATION DETAILS

To quantify the denoising performance, we employ Signal-to-Noise Ratio (SNR) and Contrast-to-Noise Ratio (CNR) metrics, which are also used in DDM2 (Xiang et al., 2023). We differentiate foreground and background signals following Patch2Self (Fadnavis et al., 2020a): **1.** Perform uniform normalization on all the data; **2.** Use the method in (Ostu, 1979) to compute a brain mask; **3.** fit DTI (Basser et al., 1994) model to perform corpus callosum segementation; **4.** signal is corpus callosum signal, background is the signal out of the brain mask. **5.** Compute SNR and CNR using:

$$\text{SNR} = \frac{\text{Mean (signal)}}{\text{Var (background)}}, \quad \text{CNR} = \frac{\text{Mean (signal)} - \text{Mean (background)}}{\text{Var (background)}}, \qquad (40)$$

Table S2: $\uparrow R^2$ of microstructure model fitting on CSD & DTI. **Bold** and Underline fonts denote the best and the second-best performance, respectively. As measured by $R^2$, Di-Fusion achieves the best results across all four different settings.

| | CSD | | DTI | |
| --- | --- | --- | --- | --- |
| Method | CC | CSO | CC | CSO |
| Noisy | 0.797 | 0.614 | 0.789 | 0.484 |
| ASCM | 0.934 | 0.844 | 0.942 | 0.789 |
| DIP | 0.868 | 0.477 | 0.875 | 0.381 |
| Nr2N | 0.959 | 0.908 | 0.961 | 0.872 |
| P2S | 0.927 | 0.754 | 0.725 | 0.675 |
| DDM2 | 0.863 | 0.810 | 0.845 | 0.790 |
| OURS | **0.967** | **0.939** | **0.976** | **0.876** |

where $\mathrm{Mean}\,(\cdot)$ is the mean, $\mathrm{Var}\,(\cdot)$ is the variance; **6.** Statistics are performed on all computed SNR and CNR values, and a box plot like Fig. S12 is created.

## D.4 Simulated data implementation details

**Details on making simulated data**   Apart from the experiments on *in-vivo* datasets, we further simulate noisy k-space data to demonstrate that Di-Fusion can still be used for denoising tasks on simulated noisy MRI data, which is done on fastMRI datasets (fastMRI provides raw, complex, multi-echo, and multi-coil k-space MRI data) (Tibrewala et al., 2023; Zbontar et al., 2018). To simulate the effects of adding additional complex noise to the k-space data, we employ k-space noise addition strategies that have been previously validated in prior work (Desai et al., 2021a;b; Xiang et al., 2023). Specifically, we start by sampling Gaussian noise with different standard deviations (to simulate different noise intensities) and add it to the real and imaginary parts of each coil's k-space data. Next, we utilize the inverse transformation function implemented in fastMRI (Tibrewala et al., 2023; Zbontar et al., 2018) to convert the k-space data into simulated noisy datasets with varying degrees of noise. We simulate five datasets with different noise intensities.

**Declaration**   DIP is not considered as a comparison method due to its long computational time (the need for retraining on each image) and the mild blurring shown in Fig. 5, Fig. S13, S14 and Fig. S15. The original Patch2Self is not included as a comparison method because it typically requires at least ten 3D volumes to achieve good results (Fadnavis et al., 2020b;a; Garyfallidis et al., 2014). In contrast, the input 3D volumes in the simulated experiments are limited to a maximum of two (two for DDM2, one for Di-Fusion and Noisier2Noise). Directly comparing it with other methods on simulated data would be unfair. However, we still develop a reimplementation of Patch2Self, with modifications to the volume extraction part to limit the input volumes (two in our modified Patch2Self). *It should be noted that our modified Patch2Self is solely utilized in the simulated experiments, where the limited input of two volumes does not yield optimal results. The original Patch2Self is still used in the remaining experiments carried out in this paper.* Nevertheless, Patch2Self remains a useful approach when applied to a larger number of volumes (e.g., ten).

## E Supplementary experimental results

In this section, we include comparisons with additional denoising methods, including MPPCA (Veraart et al., 2016), Noise2Score (N2S) (Kim & Ye, 2021), Recorrupted2Recorrupted (R2R) (Pang et al., 2021), and Patch2Self2 (P2S2) (Fadnavis et al., 2024) (reproduction details are provided in Appendix D.1).

### E.1 Microstructure model fitting and data distribution plots

In Table S2, as measured by $R^2$, our Di-Fusion achieves the best results across all four different settings. An intriguing observation is that the denoised data from DDM2 exhibits a relatively higher

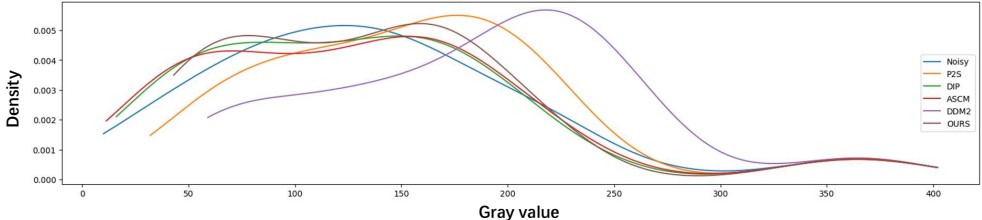

Figure S7: Data distribution plots on raw and denoised data. Note that DDM2 denoised data distribution has shifted from the noisy data.

Table S3: $\uparrow R^2$ of microstructure model fitting on CSD & DTI. **Bold** and Underline fonts denote the best and the second-best performance, respectively. As measured by $R^2$, Di-Fusion achieves the best results across all four different settings.

| Method | Noisy | ASCM | MPPCA | DIP | Nr2N | N2S | R2R | P2S | DDM2 | P2S2 | OURS |
|---|---|---|---|---|---|---|---|---|---|---|---|
| CSD-CC | 0.797 | 0.934 | 0.884 | 0.868 | 0.959 | 0.823 | 0.879 | 0.927 | 0.863 | 0.957 | **0.967** |
| CSD-CSO | 0.614 | 0.844 | 0.750 | 0.477 | 0.908 | 0.468 | 0.731 | 0.754 | 0.810 | 0.934 | **0.939** |
| DTI-CC | 0.789 | 0.942 | 0.881 | 0.875 | 0.961 | 0.831 | 0.872 | 0.725 | 0.845 | 0.973 | **0.976** |
| DTI-CSO | 0.484 | 0.789 | 0.614 | 0.381 | 0.872 | 0.348 | 0.677 | 0.675 | 0.790 | 0.867 | **0.876** |

distribution (higher mean value) than other methods in Fig. S7. An observation is that the data from DDM2 exhibits a higher distribution than other methods. This may explain the improvement of DDM2 in CNR/SNR metrics in Fig. S12, as the foreground signals have higher values. Our experiments on downstream clinical tasks in Section 4.2 have shown no correlation between high or low scores on CNR/SNR metrics and the performance of the downstream clinical tasks.

In Table S3, we summarize the quantitative $R^2$ metrics of all comparison methods. As measured by $R^2$, our Di-Fusion still achieves the best results across all four different settings.

## E.2 ADDITIONAL COMPARISONS ON TRACTOGRAPHY

In Fig. E.2, we illustrate the effect on the tractography of OR using additional denoising methods. Di-Fusion effectively performs denoising while maintaining high FBCs, with "Noisy_filtering" serving as the reference for high FBCs.

## E.3 DTI DIFFUSION SIGNAL ESTIMATES COMPARISONS

We further examine how the denoising quality translates to downstream clinical tasks such as creating DTI (Basser et al., 1994) diffusion signal estimates using the various denoising methods. Details are in Appendix D.2. In Fig. S9, we show fractional anisotropy, axial diffusivity, mean diffusivity, and radial diffusivity comparisons. Apart from the poor performance of ASCM, we observe that other methods performed well on diffusion signal estimates.

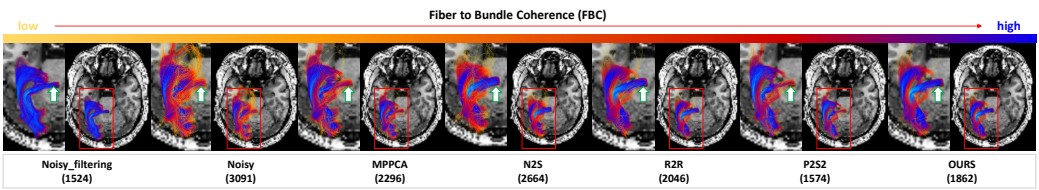

Figure S8: Density map of FBC projected on the streamlines of the OR bundles. The numbers in parentheses represent the number of streamlines. Di-Fusion maintains high FBCs (consider "Noisy_filtering" as references for high FBCs).

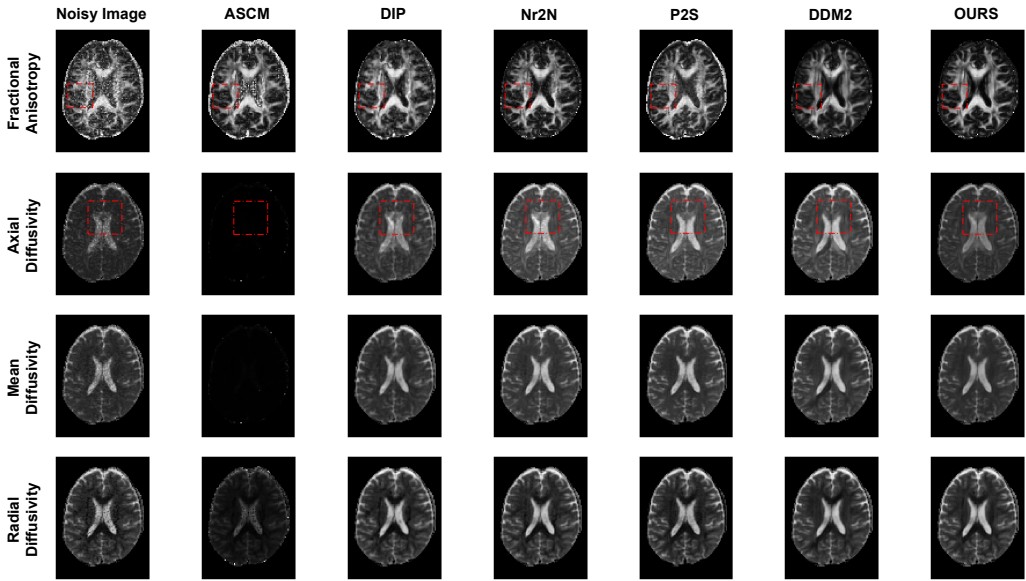

Figure S9: Fractional anisotropy, axial diffusivity, mean diffusivity, and radial diffusivity comparisons. The main differences are highlighted within the red dashed box. Our method effectively suppresses noise and reconstructs fiber tracts while maintaining a grayscale consistency with the original data (No overall brightening of diffusion signal estimates, especially on axial diffusivity)

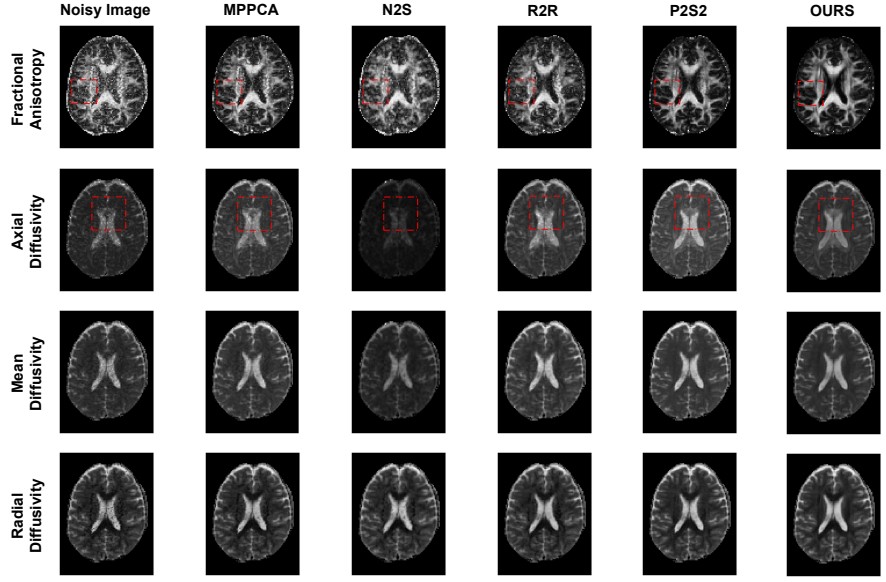

Figure S10: Fractional anisotropy, axial diffusivity, mean diffusivity, and radial diffusivity comparisons. The main differences are highlighted within the red dashed box. Our method effectively suppresses noise and reconstructs fiber tracts while maintaining a grayscale consistency with the original data (No overall brightening of diffusion signal estimates, especially on axial diffusivity)

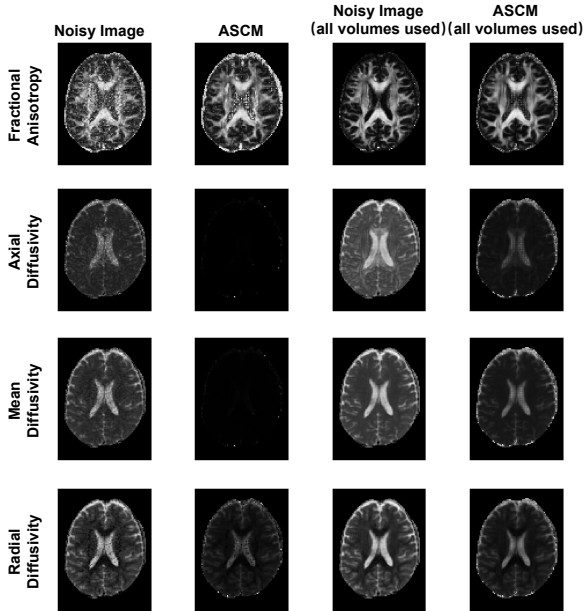

Figure S11: Fractional anisotropy, axial diffusivity, mean diffusivity and radial diffusivity comparisons of previous version results and all volumes used results.

On radial diffusivity, all methods exhibited an improved and less noisy representation of the diffusion directions of fiber tracts. However, on fractional anisotropy, and axial diffusivity, DDM2 shows inconsistencies with the noisy image at specific locations (highlighted by the red dashed box), indicating excessive denoising. Our method effectively suppresses noise and reconstructs fiber tracts while maintaining a grayscale consistency with the original data (no overall brightening of diffusion signal estimates, especially on axial diffusivity).

In Fig. E.3, we further compare DTI diffusion signal estimates with those obtained using additional denoising methods. We follow the same steps in Appendix D.2 to estimate fractional anisotropy (FA), axial diffusivity (AD), mean diffusivity (MD), and radial diffusivity (RD). Additionally, we computed the reference for FA, AD, MD, and RD using all available dMRI data. Based on this, we performed the calculation of PSNR and SSIM for all slices. The quantitative results are summarized in Table E.3

Questions on ASCM diffusion signal estimates. In Fig. S9, minimal signals are observed in the ASCM axial and mean diffusivity. We further utilize all volumes to perform diffusion signal estimates and show results in Fig. S11. A noticeable signal should be revealed if all volumes are used in the diffusion signal estimates. This finding demonstrates that the denoising results of ASCM could significantly hinder the DTI diffusion signal estimates.

### E.4 QUANTITATIVE RESULTS ON *in-vivo* DATA

Given the absence of a consensus on image quality metrics, particularly in unsupervised reference-free settings (Chaudhari et al., 2020; Woodard & Carley-Spencer, 2006), the task of assessing perceptual MRI quality becomes a challenging research problem (Mittal et al., 2011). Considering the infeasibility of using PSNR and SSIM metrics (no ground truth reference images) and their limited correlation with clinical utility (Mason et al., 2019), computing metrics in downstream clinical regions of interest is more reasonable (Adamson et al., 2021). We follow the procedure outlined in (Xiang et al., 2023) to calculate SNR/CNR metrics (Details are in D.3). The quantitative denoising results were reported as mean and standard deviation scores for the complete 4D volumes in Fig. S12. Di-Fusion indicates better performance against competing methods. Our experiments in Section 4.2 have shown no correlation between high or low scores on CNR/SNR metrics and the performance of the downstream clinical tasks.

| Method | Metric | FA | MD | RD | AD |
|--------|--------|------|------|------|------|
| Noisy | PSNR | 23.47 | 31.79 | 33.56 | 26.39 |
| | SSIM | 0.8760 | 0.9635 | 0.9637 | 0.9247 |
| ASCM | PSNR | 22.63 | 21.81 | 24.70 | 20.89 |
| | SSIM | 0.8897 | 0.8188 | 0.8921 | 0.8102 |
| DIP | PSNR | 24.76 | 33.01 | 32.42 | 30.21 |
| | SSIM | 0.8894 | 0.9675 | 0.9630 | 0.9453 |
| MPPCA | PSNR | 26.37 | 35.28 | 36.35 | 29.63 |
| | SSIM | 0.9048 | 0.9751 | 0.9780 | 0.9534 |
| Nr2N | PSNR | 29.47 | 38.42 | 38.48 | 34.87 |
| | SSIM | 0.9354 | 0.9851 | 0.9865 | 0.9712 |
| N2S | PSNR | 23.13 | 29.71 | 31.08 | 25.84 |
| | SSIM | 0.8691 | 0.9469 | 0.9491 | 0.8973 |
| R2R | PSNR | 25.93 | 34.54 | 33.82 | 30.21 |
| | SSIM | 0.8809 | 0.9708 | 0.9678 | 0.9394 |
| P2S | PSNR | 24.18 | 32.16 | 30.62 | 33.27 |
| | SSIM | 0.9090 | 0.9708 | 0.9632 | 0.9677 |
| DDM2 | PSNR | 26.77 | 37.53 | 37.39 | 33.21 |
| | SSIM | 0.9041 | 0.9872 | 0.9848 | 0.9610 |
| P2S2 | PSNR | 30.08 | 39.28 | 40.23 | 34.06 |
| | SSIM | 0.9432 | 0.9894 | 0.9921 | 0.9701 |
| OURS | PSNR | **30.79** | **40.26** | **40.35** | **35.30** |
| | SSIM | **0.9450** | **0.9931** | **0.9923** | **0.9763** |

Table S4: PSNR and SSIM comparisons for DTI diffusion signal estimates. Here "AD" represents axial diffusivity, "RD" represents radial diffusivity, "MD" represents mean diffusivity and "FA" represents fractional anisotropy.

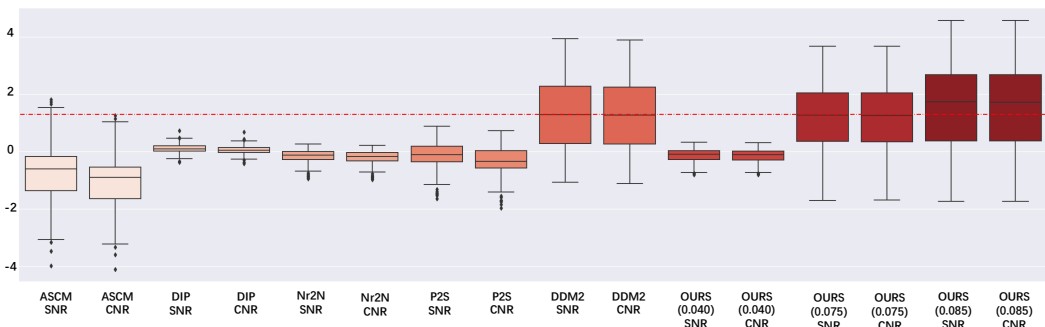

Figure S12: Box plots of Quantitative SNR/CNR metrics scores. The numbers within parentheses under OURS represent the value of $\mathcal{CSNR}$ (Section 3.3). Di-Fusion indicates better performance in terms of SNR/CNR metrics.

We summarize the CNR and SNR metrics of all comparison methods in Table S5, where our method achieves better results in both CNR and SNR metrics.

Table S5: Comparison of SNR and CNR. **Bold** and Underline fonts denote the best and the second-best performance, respectively.

| Method | ASCM | MPPCA | DIP | R2R | N2S | Nr2N | P2S | P2S2 | DDM2 | OURS |
|--------|------|-------|-----|-----|-----|------|-----|------|------|------|
| SNR | -0.7251 | 0.2372 | 0.1035 | -0.0099 | 0.2266 | -0.1598 | -0.1616 | 0.1526 | 1.3040 | **1.5735** |
| CNR | -1.0513 | 0.2191 | 0.0567 | -0.1161 | -0.0304 | -0.2004 | -0.3694 | 0.1177 | 1.2725 | **1.5687** |

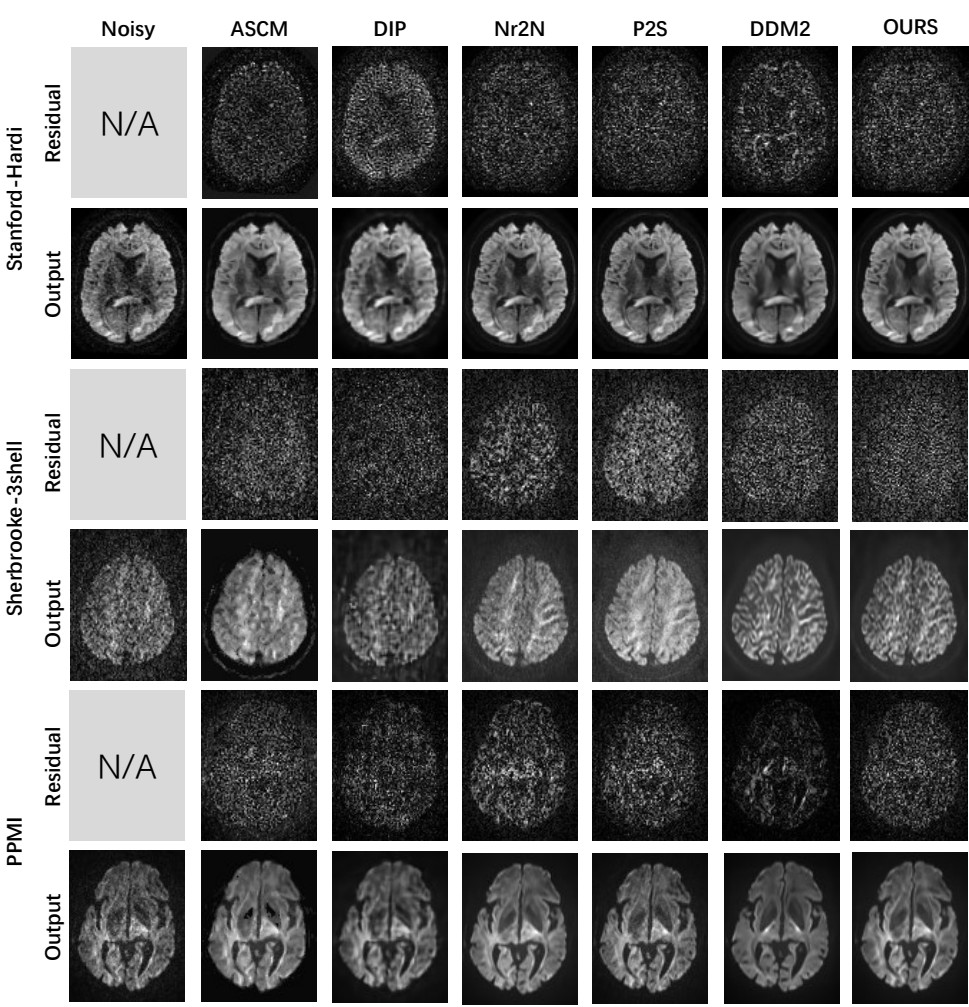

Figure S13: More qualitative results on Stanford-Hardi. "OURS" results are obtained when $\mathcal{CSNR} = 0.040$ (Section 3.3). Notice that Di-Fusion suppresses noise and does not show any anatomical structure in the residual plots.

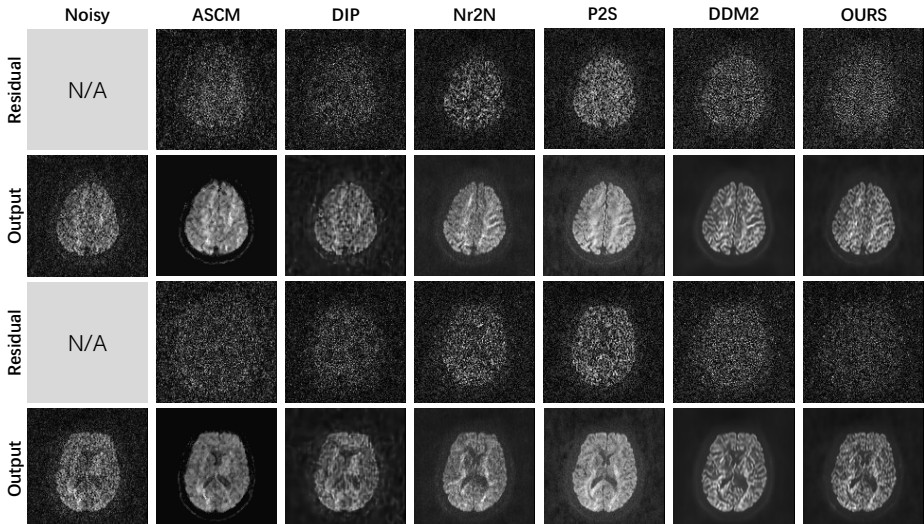

Figure S14: More qualitative results on Sherbrooke 3-Shell. "OURS" results are obtained when $\mathcal{CSNR} = 0.040$ (Section 3.3). Notice that Di-Fusion suppresses noise and does not show any anatomical structure in the residual plots.

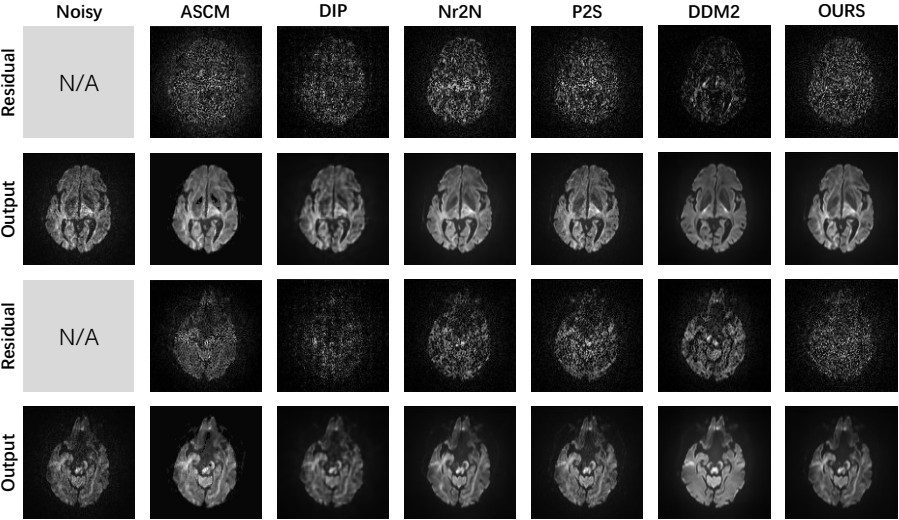

Figure S15: More qualitative results on PPMI. "OURS" results are obtained when $\mathcal{CSNR} = 0.040$ (Section 3.3). Notice that Di-Fusion suppresses noise and does not show any anatomical structure in the residual plots.

### E.5 MORE QUALITATIVE RESULTS

In Fig. S13, Fig. S14 and Fig. S15, we show more qualitative results. For each of the datasets, we show the axial slice of a randomly chosen 3D volume and the corresponding residuals (squared differences between the noisy data and the denoised output). We can observe that the results are generally consistent with those presented in Section 4.3. From the residuals of "DDM2", it can be observed that particular regions are suppressed (especially in Fig. S13, the DDM2 results for the second slice show that the residuals contain a significant amount of anatomical information). Notice that Di-Fusion suppresses noise and does not show any anatomical structure in the residual plots.

### E.6 QUANTITATIVE AND QUALITATIVE RESULTS ON SIMULATED DATA

We show the quantitative and qualitative results in Fig. S16. When the noise intensity is high (left three columns), our method performs the best. When the noise intensity is low (right two columns), denoising results are comparable to other methods. Considering the high PSNR and SSIM in the right two columns, it suggests that in real-world scenarios, such data may not require denoising and can still enable effective clinical use. Di-Fusion has more potential for generalization and applicability as it performs better under high noise intensity.

### E.7 COMPARE UNDER MIXED B-VALUE IMAGES

In Fig. S17, we show additional qualitative results when training on mixed b-value images (Sherbrooke 3-Shell has 1000, 2000, and 3500 b-values). Nr2N, P2S, DDM2, and our method both show minimal sensitivity to mixed b-values training data. Minor brightness variations in DDM2 and P2S for multiple b-values have a negligible impact on the overall results. Our method primarily learns the mapping from one volume to another, making it less affected by varying b-values in different volumes (P2S uses all the other different volumes, DDM2 uses two different input volumes at Stage 1, and Di-Fusion only uses one different volume). This suggests that the performance of Di-Fusion is relatively robust and not reliant on specific b-value configurations.

### E.8 QUALITATIVE RESULTS WHEN USING FEWER DMRI VOLUMES

As shown in Fig. E.8, when using fewer dMRI volumes (20% of original dMRI volumes), Di-Fusion still demonstrates effective denoising capabilities. Please pay special attention that the 30 dMRI volumes here refer to the total training data. Additionally, using a portion of dMRI data from different individuals for model training is a more clinically feasible approach.

## F VISUALIZATION OF DI-FUSION

### F.1 FUSION PROCESS: LINEAR INTERPOLATION BETWEEN THE TWO ENDPOINTS

The noise schedule can be found in Appendix D.1. In Fig. S19, we provide a visual demonstration of $x_t^*$ (Eq. (6)). Without the Fusion process, the model output would deviate from $x_{out}$, resulting in drifted results. By incorporating the Fusion process, where each linear interpolation $x_t^*$ from $x'$ to $x$ has $x$ as the target, the inference process avoids drifted results (Fig. 1 (a)). We further conducted ablation studies to demonstrate the significance of the Fusion process in Appendix G.1.

### F.2 "DI-" PROCESS: DIFFERENT NOISE DISTRIBUTION

Experiment details: We computed all the $\xi_{x-x'}$ in Stanford HARDI dataset (meaning a total of $76 * 150 = 11400$ noisy images), calculated the grayscale histogram, mean and variance of these noisy images, and presented the calculated mean and variance in the form of a histogram. At the same time, we randomly sampled 11400 Gaussian noisy images and performed the same statistical operation.

Statistical properties of $\xi_{x-x'}$: From Fig. S20, the noise calculated by the "Di-" process has significantly different statistical properties from Gaussian noise. This is reflected in the fact that: **1.** the variance of the calculated noise is relatively small and does not follow a normal distribution **2.**

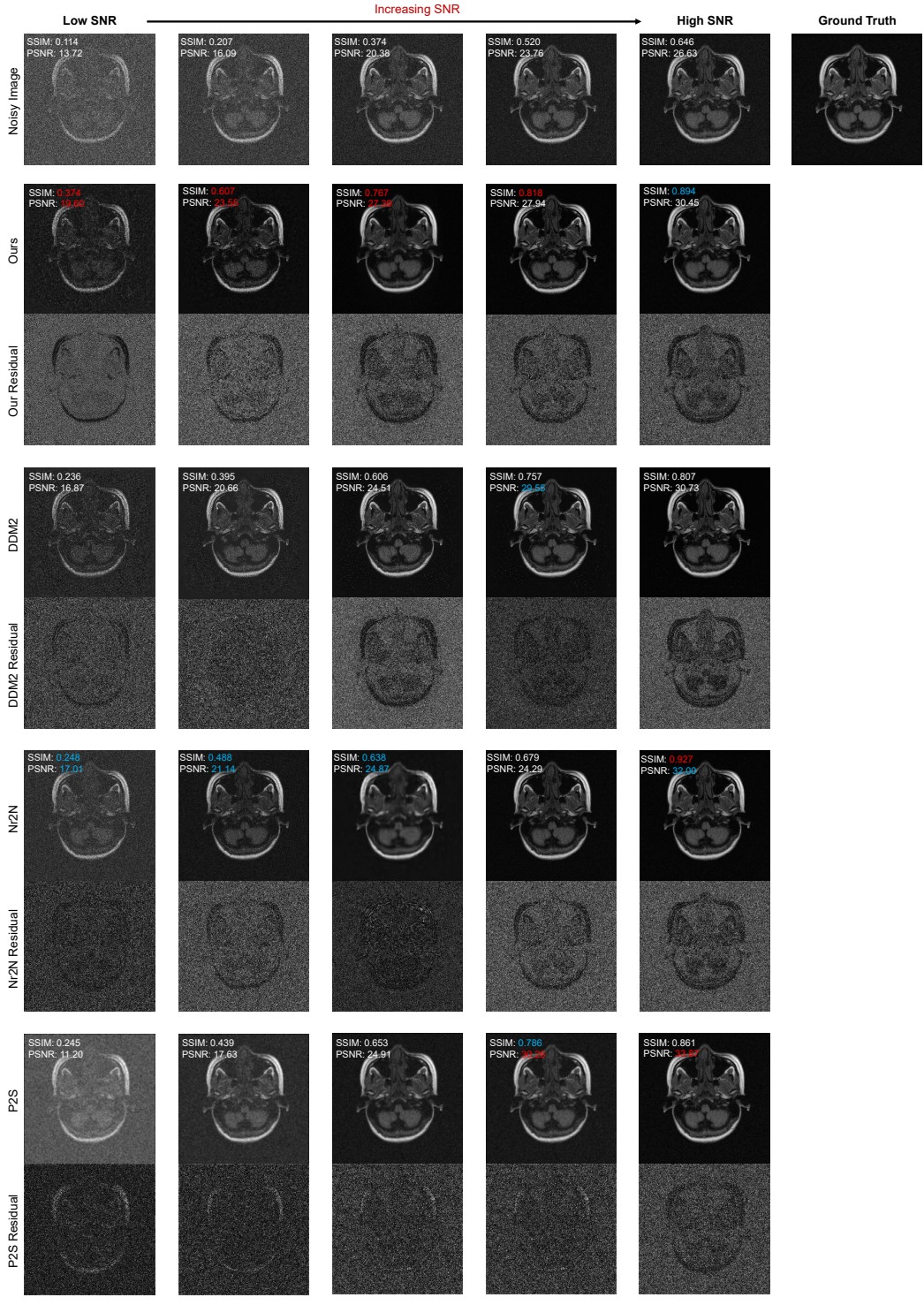

Figure S16: Quantitative and qualitative results on simulated data. In our experiments, $\mathcal{CSNR} = 0.040$. The red color represents the highest value for the metric, while the blue color represents the second-highest value. Please note that these are the results of a single round of simulated experiments and their corresponding PSNR and SSIM metrics scores.

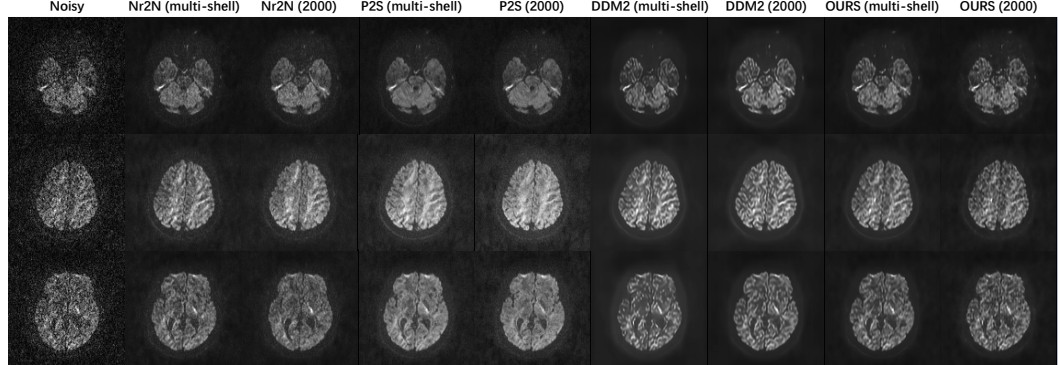

Figure S17: Additional results when training on mixed b-value images (All our results are obtained when $\mathcal{CSNR} = 0.040$). "(2000)" indicates using data with only a b-value of 2000. "(multi-shell)" represents using data with mixed b-values, including 1000, 2000, and 3500. The performance of Di-Fusion is relatively robust and not reliant on specific b-value configurations

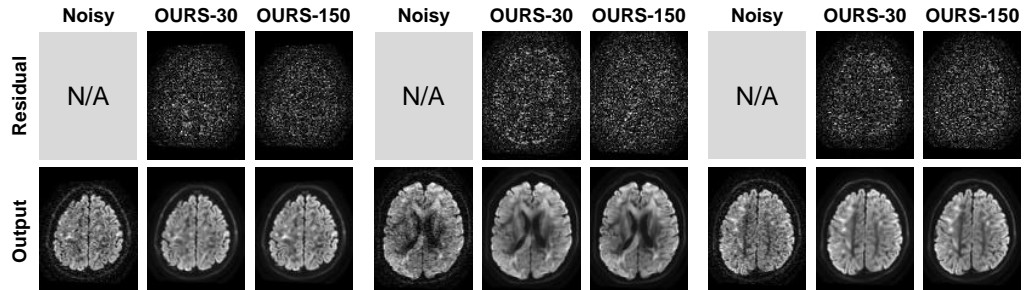

Figure S18: Qualitative results when using fewer dMRI volumes. OURS-30 indicates using 30 dMRI volumes, while OURS-150 represents using 150 dMRI volumes.

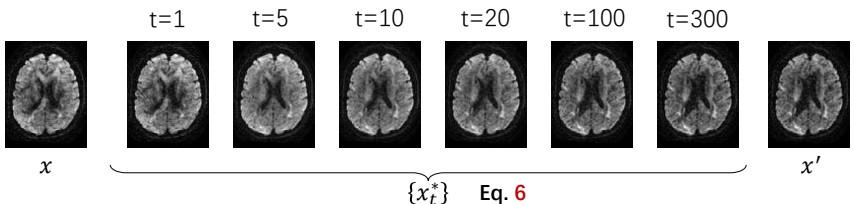

Figure S19: Visual demonstration of $x_t^*$ obtained by different $t$.

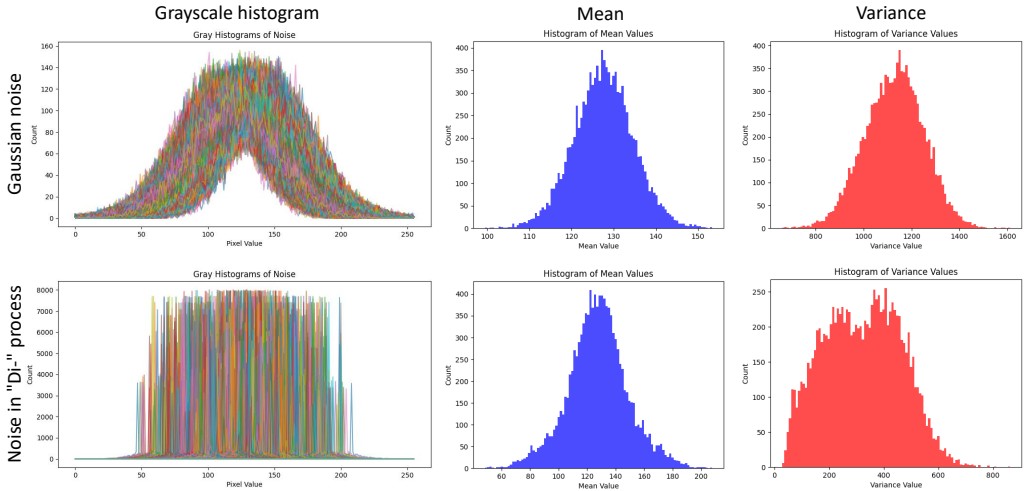

Figure S20: Grayscale histogram, mean and variance of these noisy images. We computed all the $\xi_{x-x'}$ in Stanford HARDI dataset (meaning a total of $76 * 150 = 11400$ noisy images), calculated the grayscale histogram, mean and variance of these noisy images, and presented the calculated mean and variance in the form of a histogram. At the same time, we randomly sampled 11400 Gaussian noisy images and performed the same statistical operation. The noise calculated by the "Di-" process has significantly different statistical properties from Gaussian noise. This is reflected in the fact that: **1.** the variance of the calculated noise is relatively small and does not follow a normal distribution **2.** the counts of each pixel value on the grayscale histogram of $\xi_{x-x'}$ are similar, rather than a normal distribution in Gaussian noise.

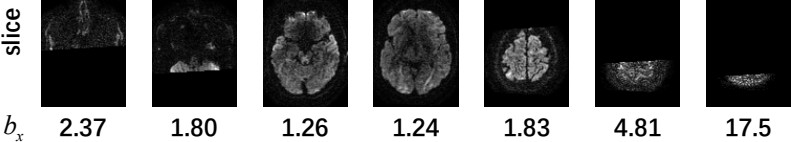

Figure S21: Slices and their corresponding $b_x$. The $b_x$ values of the edge slices are relatively larger.

the counts of each pixel value on the grayscale histogram of $\xi_{x-x'}$ are similar, rather than a normal distribution in Gaussian noise. Different noise distribution makes $\mathcal{F}_\theta$ more capable of modeling real-world noise.

## F.3 SAMPLING PROCESS: ITERATIVE AND STABLE REFINEMENT

**Value of $b_x$** In Section 3.3, we adopt a simple definition (Eq. (12)) to calculate a coefficient $b_x$ that accounts for the ratio of brain tissue to the entire image. Fig. S21 displays the slices accompanied by their corresponding $b_x$. It can be observed that Eq. (12) is a simple method for evaluating the proportion of the brain tissue and $b_x$ can be used to correct $d_x$ in Eq. (13).

**Iterative and controllable refinement** In Section 3.3, we propose an adaptive termination during the sampling process. This allows us to control the sampling process by setting the value of $\mathcal{CSNR}$. In general, setting a lower $\mathcal{CSNR}$ will preserve more anatomical details. On the other hand, setting a higher $\mathcal{CSNR}$ will remove more noise at the cost of losing some anatomical details (see Fig. S22 for visual demonstrations).

$d_x$ **plots** In Section 3.3, we calculate $d_x$ (Eq. (13)) to represent the degree of denoising in $x_{out}$. In Fig. S23, we illustrate the variation of $d_x$ during the reverse sampling process and present the results when implementing an adaptive termination. It can be observed that with such an adaptive termination, it is possible to quickly obtain denoised results (low $\mathcal{CSNR}$ results) or further remove noise effectively (high $\mathcal{CSNR}$ results).

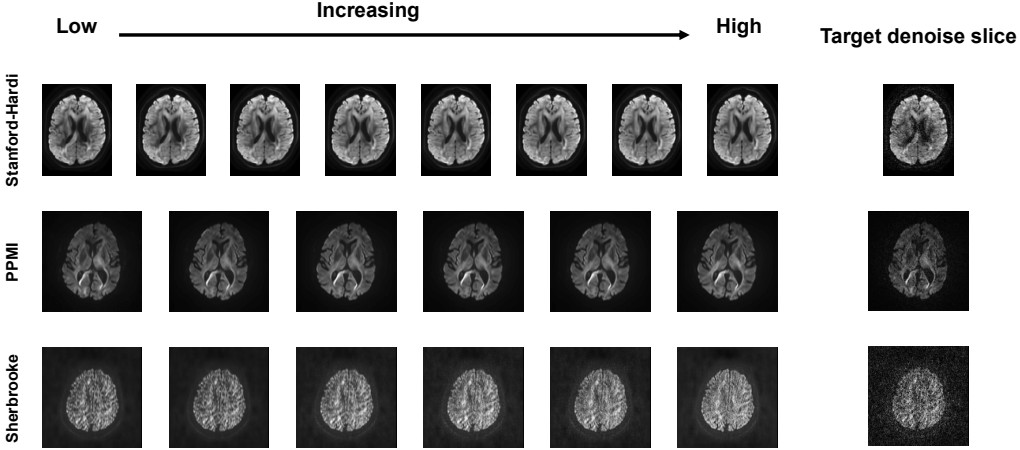

Figure S22: The results of sampling process obtained by different $\mathcal{CSNR}$.

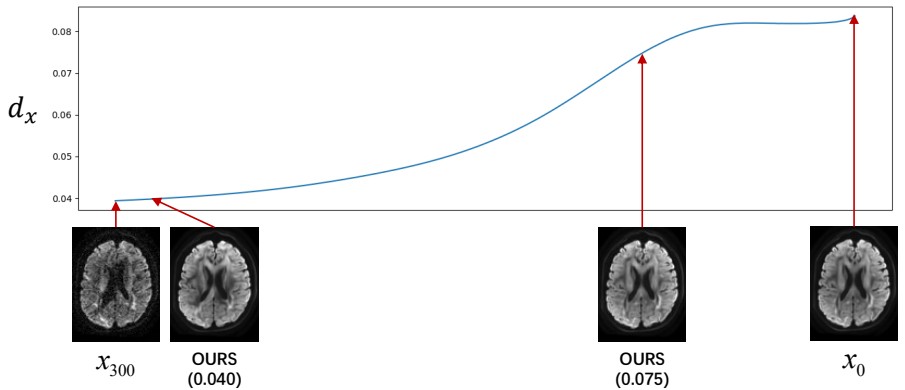

Figure S23: Variation of $d_x$ during the sampling process. The numbers within parentheses below OUR represent the value of $\mathcal{CSNR}$ (Section 3.3).

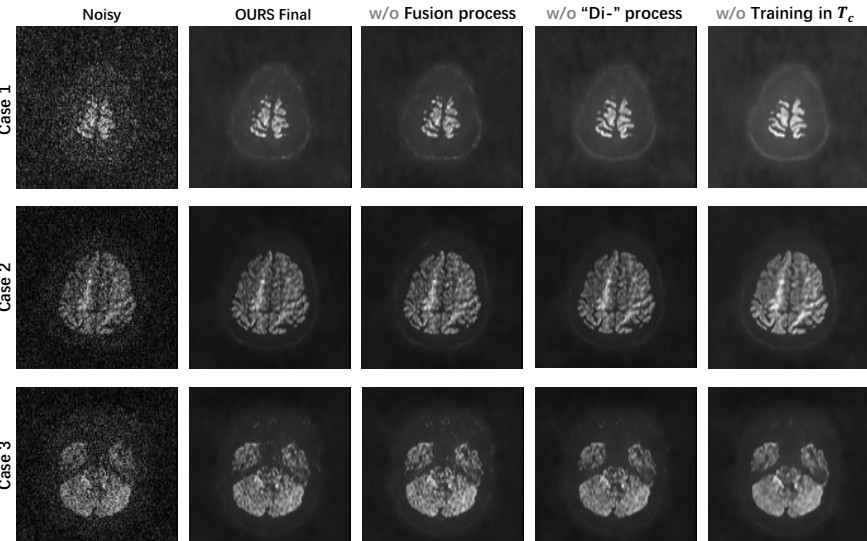

Figure S24: Qualitative results of ablation studies (Implement an adaptive termination mentioned in Section 3.3 during the sampling process and all the experiments $\mathcal{CSNR} = 0.040$). Headings distinguish results obtained using different ablation settings.

# G ABLATION STUDIES

## G.1 ON TRAINING IN DI-FUSION

**On Fusion process** In Section 3.1, we utilize Eq. (6) to compute linear interpolation from $x'$ to $x$, aiming to reduce drift in final results. We disable the Fusion process by substituting $x'$ for $x_t^*$. In Fig. S24, when going through several reverse steps (low $\mathcal{SCNR}$), the results without the Fusion process do not exhibit significant deviations. However, when the adaptive termination is not implemented (which means completing all the sampling steps), noticeable slice misalignment occurs in Fig. S25 (highlighted by the red box).

**On "Di-" process** In Section 3.1, we utilize Eq. (8) to compute a noise distribution $\xi_{x-x'}$ and use it in $q(x_t|x_t^*)$ and $p_\mathcal{F}(x_{t-1}|x_t)$. We directly replace $\xi_{x-x'}$ calculated in the "Di-" process with Gaussian noise. Without the "Di-" process, results lack some high-frequency information, and the overall gray value of the denoised images has also changed (Case 1 in Fig. S24). Some may consider using $\xi_{x-x'}$ only during the diffusion process $q(x_t|x_t^*)$ and Gaussian noise during the sampling process $p_\mathcal{F}(x_{t-1}|x_t)$. We present the results of this setting in Fig. S27, where it can be observed that artifacts occur along the edge slices.

**On training the latter diffusion steps** In Section 3.2, we preform training the latter diffusion steps by optimizing $\mathcal{F}_\theta$ to condition on $\overline{\alpha}_t$, $t \sim \text{Uniform}(\{1, \cdots, T_c\})$, $T_c = 300$. We disable training the latter diffusion steps by optimizing $\mathcal{F}_\theta$ to condition on $\overline{\alpha}_t$, $t \sim \text{Uniform}(\{1, \cdots, 1000\})$ and balance the training iterations (training the latter diffusion steps iterations: $1e^5$, training all diffusion steps: $3.5e^5$). Without training the latter diffusion steps, the denoising results are noticeably smoother and have more hallucinations (Fig. S24 and Fig. S25). We recommend training the latter diffusion steps based on its potential advantages, which include *(i)* mitigating hallucinations and *(ii)* reducing training time with improved stability.

## G.2 ON SAMPLING IN DI-FUSION

**On adaptive termination** In Section 3.3, we introduce an adaptive termination to enable iterative and adjustable refinement. In Fig. S21, we show slices and their corresponding $b_x$, the $b_x$ values of the edge slices are relatively larger. In Fig. S22, we illustrate the impact of $\mathcal{CSNR}$ on the sampling results. In Fig. S23, we show variation of $d_x$ during the sampling process.

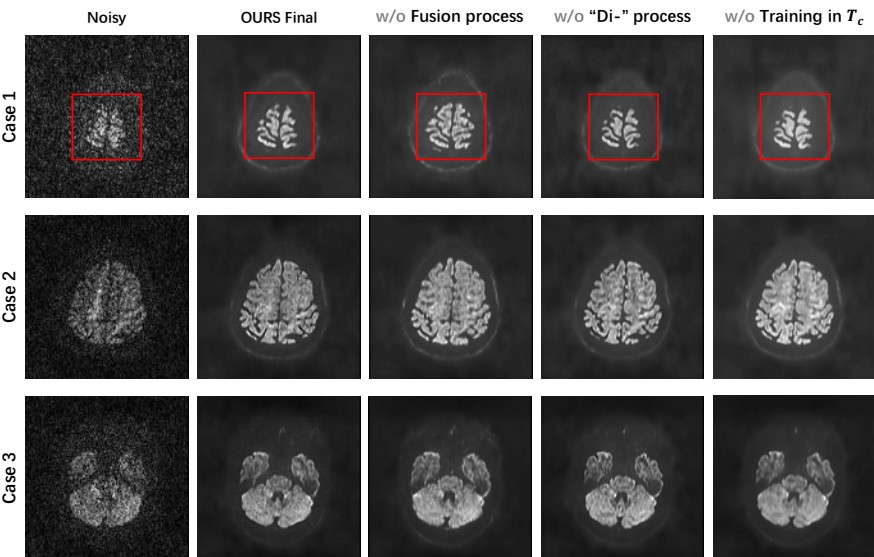

Figure S25: Qualitative results of ablation studies (Didn't implement an adaptive termination mentioned in Section 3.3 during the sampling process). Headings distinguish results obtained using different ablation settings. The red box highlights the main differences.

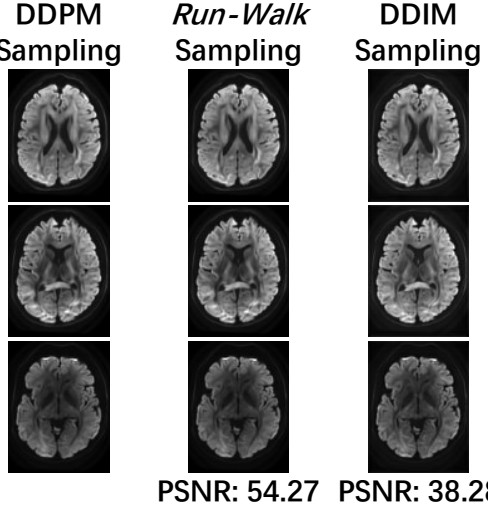

Figure S26: DDPM sampling *v.s. Run-Walk* accelerated sampling (didn't implement an adaptive termination in Section 3.3) *v.s.* DDIM sampling. For all results, $\eta = 0$. PSNR, SSIM are calculated using DDPM sampling results as references. This indicates that the sampling results from *Run-Walk* sampling are closer to the sampling results when accelerated sampling is not used.

***Run-Walk* accelerated sampling maintains the sampling quality**    In Fig. S26, we show results obtained by different sampling strategies and metrics (averaged PSNR and SSIM on all volumes) calculated using DDPM sampling results as references. Directly performing DDIM sampling on a pre-trained model may lead to biased results (use DDPM sampling results as references). *Run-Walk* accelerated sampling significantly improves sampling speed and reduces inference time while maintaining the sampling quality relatively unchanged.

**About sampling time**    We do experiments to demonstrate that the additional computations in Section 3.3 do not impact the sampling speed. Firstly, we set $\mathcal{CSNR}$ to 1, which means that all

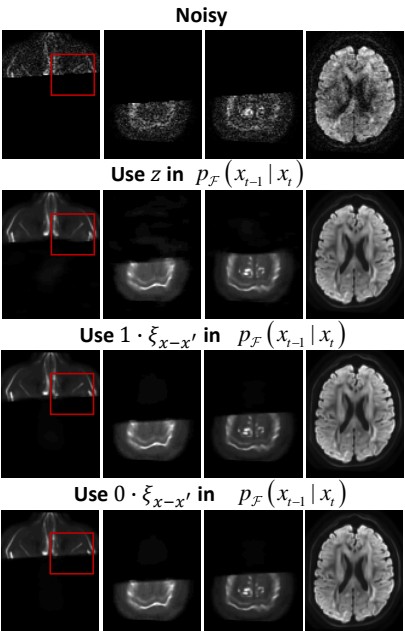

Figure S27: Using $z$ during $p_{\mathcal{F}}\left(x_{t-1}|x_t\right)$ *v.s.* Using $1\cdot\xi_{x-x'}$ during $p_{\mathcal{F}}\left(x_{t-1}|x_t\right)$ *v.s.* Using $0\cdot\xi_{x-x'}$ during $p_{\mathcal{F}}\left(x_{t-1}|x_t\right)$ (all results didn't implement an adaptive termination mentioned in Section 3.3 during the sampling process).

slices undergo the extra computations and the whole sampling process since $\mathcal{CSNR}$ is sufficiently large. Subsequently, we remove the extra computational operations and perform sampling again. The sampling time in the first experiment was 1.19 seconds per slice. In contrast, in the second experiment, it was 1.18 seconds per slice, which indicates that the additional operations have minimal impact on the sampling speed. In Table S6, the sampling time per individual slice is presented for different $\mathcal{CSNR}$. We find that When $\mathcal{CSNR}$ is low ($\mathcal{CSNR} = 0.040$), *the sampling time is 0.0395 seconds per slice*. This indicates that our adaptive termination and *Run-Walk* accelerated sampling greatly reduce the sampling time.

**Using $\xi_{x-x'}$ in reverse process** In Fig. S27, we demonstrate the importance of using $\xi_{x-x'}$ and setting $\eta = 0$ in the sampling process. It can be observed from the final results that in the central slices (with more brain tissue), using $z \sim \mathcal{N}\left(\mathbf{0}, \mathbf{I}\right)$ during the sampling process $p_{\mathcal{F}}(x_{t-1}|x_t)$ only leads to subtle differences in the denoised results. However, in the edge slices (with less brain tissue), using $z$ significantly impacts the sampling results, resulting in additional regions that appear inexplicably (highlighted by the red box, and these additional regions don't appear in noisy data). During the sampling process, DDM2 uses $z$. Because our sampling process is deterministic, according to the experiments in DDIM (Song et al., 2020b), we set $\eta = 0$. We further demonstrated the sampling results in the figure with $\eta = 1$ and $\eta = 0$. When $\eta = 1$, although the presence of unexpected regions is reduced, some still remain. However, when $\eta = 0$, such issues don't arise.

Table S6: Sampling time per slice for different $\mathcal{CSNR}$ (Stanford HARDI). We set different $\mathcal{CSNR}$ parameters for Run-Walk accelerated sampling and DDPM sampling to perform adaptive termination.

| $\mathcal{CSNR}$ | Time (s) for Run-walk | Time (s) for DDPM |
|---|---|---|
| 0.04 | 0.0395 | 0.327 |
| 0.045 | 0.115 | 0.739 |
| 0.05 | 0.626 | 2.01 |
| 0.055 | 1.08 | 5.48 |
| 0.06 | 1.11 | 6.97 |
| 1 | 1.18 | 11.5 |

Table S7: $\uparrow R^2$ of microstructure model fitting on CSD & DTI. **Bold** and Underline fonts denote the best and the second-best performance, respectively.

| Method | CSD | | DTI | |
|---|---|---|---|---|
| | CC | CSO | CC | CSO |
| Noisy | 0.797 | 0.614 | 0.789 | 0.484 |
| ASCM | 0.934 | 0.844 | 0.942 | 0.789 |
| DIP | 0.868 | 0.477 | 0.875 | 0.381 |
| Nr2N | 0.959 | 0.908 | 0.961 | 0.872 |
| P2S (OLS) | 0.927 | 0.754 | 0.725 | 0.675 |
| P2S (Ridge) | 0.927 | 0.757 | 0.927 | 0.673 |
| P2S (Lasso) | 0.824 | 0.471 | 0.816 | 0.429 |
| P2S (OLS, r=1) | 0.934 | 0.832 | 0.950 | 0.735 |
| DDM2 | 0.863 | 0.810 | 0.845 | 0.790 |
| OURS | **0.967** | **0.939** | **0.976** | **0.876** |

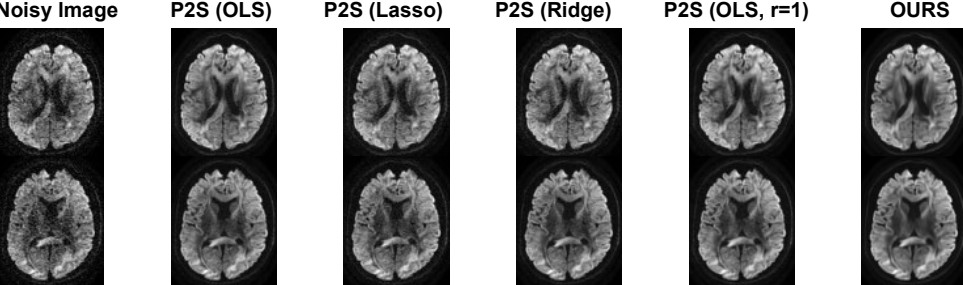

Figure S28: Comparisons with different Patch2Self experimental settings. "OURS" results are obtained when $\mathcal{CSNR} = 0.040$.

## H MORE COMPARISONS WITH COMPETING METHODS

### H.1 COMPARE WITH DIFFERENT PATCH2SELF SETTINGS

In Fig. S28, we show additional results on comparisons with different Patch2Self experimental settings. Our modifications are limited to the denoiser type (OLS, Lasso, Ridge) and patch radius, following the official repository of Patch2Self (Fadnavis et al., 2020b;a; Garyfallidis et al., 2014). The term "(r=1)" indicates changing the patch radius to 1, while the patch radius is assumed to be 0 if not specified. Modifying the denoiser type and patch radius in Patch2Self does not yield substantial improvements in the results. Altering the denoiser type does not impact the overall denoising time, whereas changing the patch radius significantly increases the overall denoising time. In our experiments, employing the OLS denoiser required 4 hours, while utilizing OLS with a patch radius of 1 took 26 hours.

In Fig. S29, we show additional tractography results on comparisons with different Patch2Self experimental settings. Although the number of streamlines is the lowest in the "(OLS, r=1)" experimental setting, *it still misses the high FBCs indicated by the white arrows*. There are no significant differences in the results in the remaining experimental settings. Considering the computational burden when setting the patch radius to 1, we suggest setting the patch radius of Patch2Self to 0 to improve efficiency.

In Table S7, we show quantitative results (on microstructure model fitting) on comparisons with different Patch2Self experimental settings. Varied experimental settings can influence the performance of microstructure model fitting. Nonetheless, these modifications do not change the rankings of the best and second-best results.

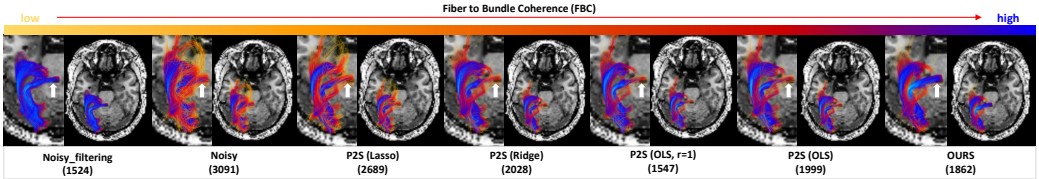

Figure S29: Density map of FBC projected on the streamlines of the OR bundles. The numbers in parentheses represent the number of streamlines.

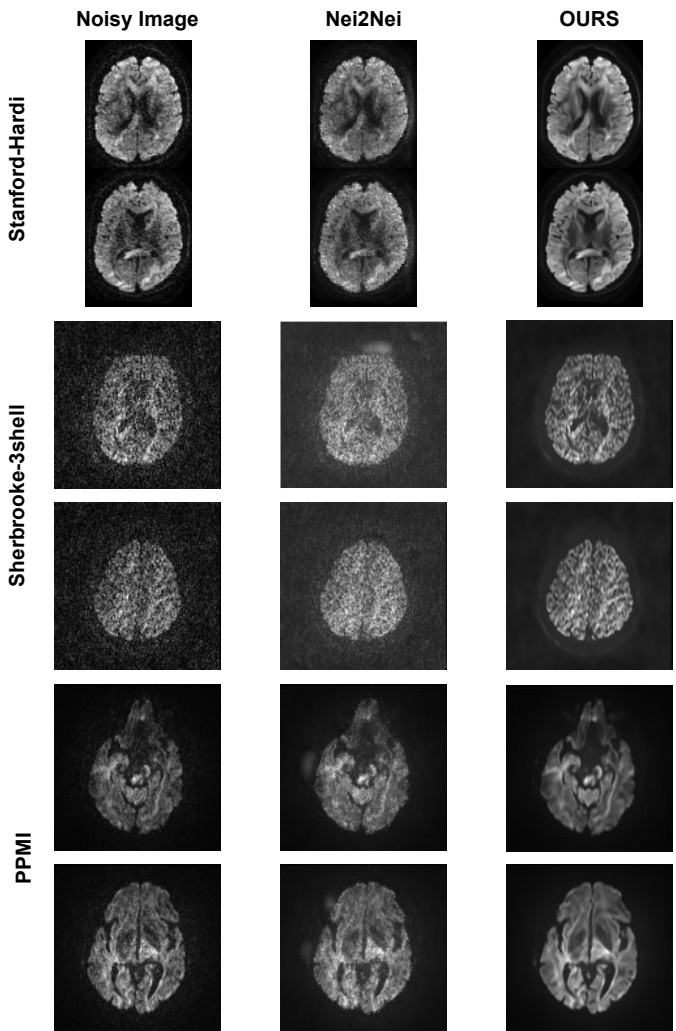

Figure S30: Qualitative comparisons with Neighbor2Neighbor. "OURS" results are obtained when $\mathcal{CSNR} = 0.040$.

## H.2 COMPARE WITH NEIGHBOR2NEIGHBOR

In Section 2.1, the mentioned methods exhibit a significant drop in performance when confronted with real-world noisy images, particularly when the explicit noise model is unknown. To make up for this, Neighbor2Neighbor (Nei2Nei) (Huang et al., 2021) and Zero-shot Noise2Noise (Mansour & Heckel, 2023) sub-sample individual noisy images to create training pairs and are more robust against

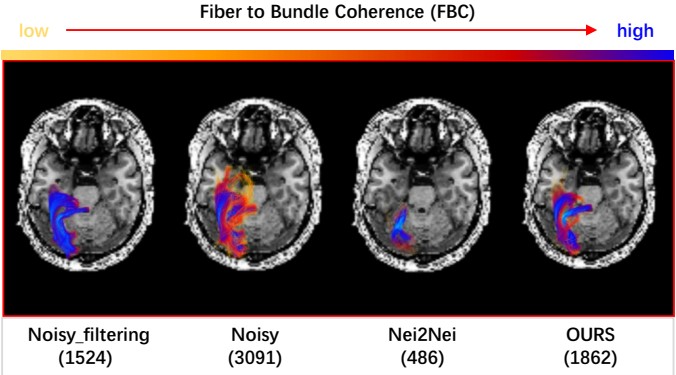

Figure S31: Density map of FBC projected on the streamlines of the OR bundles. The numbers in parentheses represent the number of streamlines.

real-world noise. We compare our method with Nei2Nei using the same model in Di-Fusion. We implement Nei2Nei with parameters set to the default values specified in their official repository[11].

The qualitative results are in Fig. S30. It can be observed that Nei2Nei does not perform well in denoising, as there are partial jagged artifacts in the image and significant changes in grayscale. This may be the reason that Nei2Nei denoising relies on the structural similarity of the neighboring regions in the image. This can also be seen in the qualitative results of Nei2Nei, where the images maintain structural similarity in sub-sampled noisy images, leading to better denoising results. In Fig. S31, it can be observed that the density map of FBC projected on the streamlines of the OR bundles is missing a significant number of FBCs; thus the denoising results of Nei2Nei are unsuitable for modeling tasks.

## I  DDM2: STAGE 1 HAS A HUGE IMPACT ON FINAL RESULTS

In Fig. S32, we present the results of DDM2's first stage and corresponding third stage on the Stanford HARDI dataset. By utilizing the hyperparameters from the DDM2 official repository and conducting experiments (*only the training iteration in stage 1 was modified*, the official training iteration is set to $10e^4$), we have observed that coarser outcomes in the first stage yield more striking yet less stable denoising results in the final stage. Conversely, deterministic outcomes in the first stage result in more stable but uninteresting denoising results in the final stage. Different first stage results lead to drastically distinct outcomes in the third stage. The CNR and SNR scores show significant differences between different first stage results (Fig. S32 (right below)). Please note that the subsequent experiments we conduct on DDM2 are using their best results.

---

[11]https://github.com/TaoHuang2018/Neighbor2Neighbor

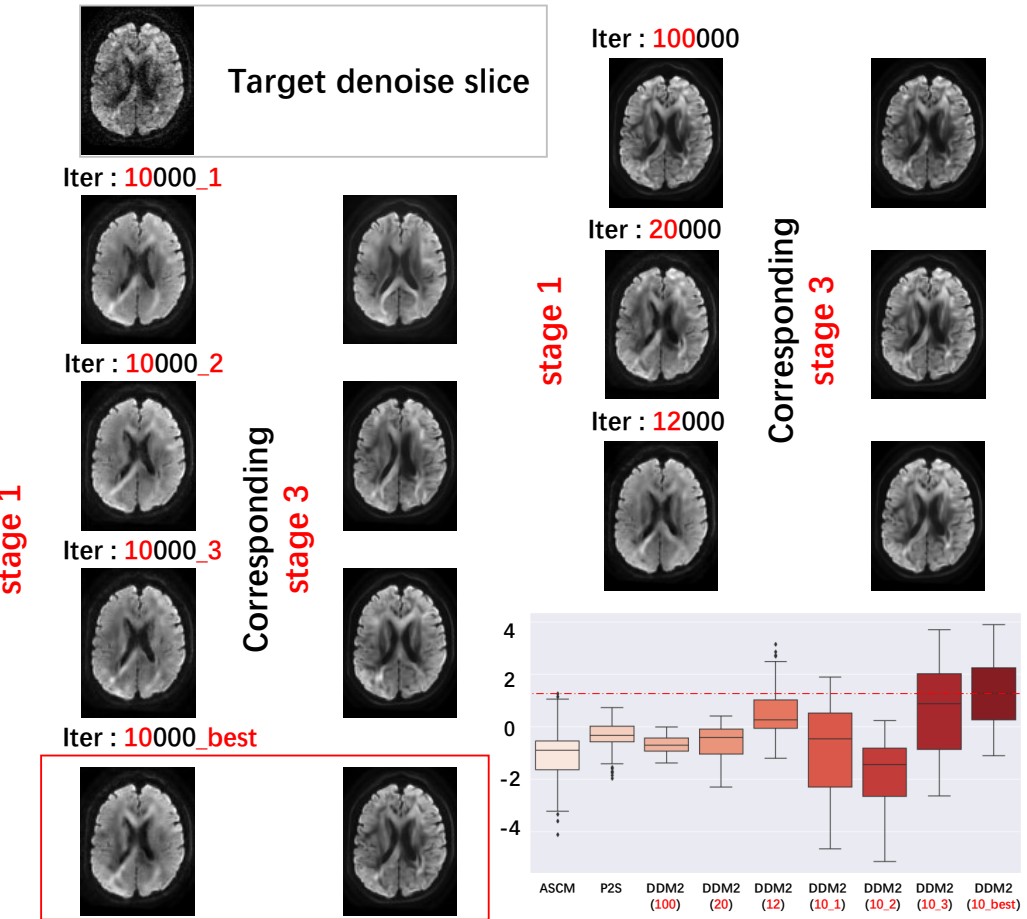

Figure S32: DDM2 unstable model outcomes. (right below) Show the CNR metric of different experiment settings; CNR/SNR metrics show the same trend. The red box highlights the best results obtained using the parameters from the official code repository. Having a stable Stage 1 often leads to poor performance in CNR/SNR metrics, the red color within the parentheses represents the settings corresponding to each experiment.

