# OpenReview forum: "Self-Supervised Diffusion MRI Denoising via Iterative and Stable Refinement"
_ICLR.cc/2025/Conference — ICLR 2025 Poster_

### Official Review · Reviewer_9LpF · 2024-10-27

**Soundness:** 3
**Presentation:** 2
**Contribution:** 2
**Rating:** 5
**Confidence:** 4

**Summary:**

The paper proposes a method for denoising diffusion MRI data sets.

This is a well-studied problem with many solutions in the literature. It is an important problem, as diffusion MRI is widely used for neuroscience and for clinical medicine. Recent years have seen a trend towards using self-supervised approaches to characterise the noise distribution and separate noise from the underlying signal.  This submission falls very much in this category, but proposes a different algorithm to those that are popular in the literature.

Experiments compare against five baselines and results appear competitive with other methods, sometimes surpassing them.

**Strengths:**

The algorithm appears novel, although I found it hard to tell from the literature review how novel it is - whether it takes ideas from other areas and repurposes them for this problem, or if this is an algorithm specifically designed for diffusion MRI.

The problem is an important one with widespread application.

Results appear competitive on a few example images shown in the figures.

**Weaknesses:**

The baselines chosen do not include the most widely used denoising methods.  A clear omission is the random-matrix theory approaches proposed by Veraart et al in a series of very highly cited papers starting with Neuroimage 2016.

The only quantitative results use simulations, which seem likely to be skewed towards to capabilities of the proposed algorithm.

The qualitative results on actual human data are questionable as to whether they show improvement over baselines.  Even if they do, these are single cherry-picked examples and it is not clear whether these are advantages that manifest over large collections of images/scenarios.

**Questions:**

Corresponding to weaknesses listed above.

---

> ### Author Response · Authors · 2024-11-21
>
> ----------
> Thank you for your constructive feedback.
>
> ----------
> **Q1:** The baselines chosen do not include the most widely used denoising methods. A clear omission is the random-matrix theory approaches proposed by Veraart et al in a series of very highly cited papers starting with Neuroimage 2016.
>
> **A1:** We will include MPPCA (Neuroimage2016) [1], Noise2Score (N2S, NeurIPS 2021) [3], Recorrupted2Recorrupted (R2R, CVPR 2021) [4], and Patch2Self2 (P2S2, CVPR 2024) [2] for further comparisons on **tractography, microstructure model fitting, diffusion signal estimates, CNR and SNR metrics**. Due to page limitations, we will include the quantitative results and its visualizations in the appendix. Here is an abstract of the quantitative results:
>
> **Table 1**. ↑$R^2$ of microstructure model fitting on CSD & DTI. **Bold** and *italic* fonts denote the best and the second-best performance, respectively. As measured by $R^2$, Di-Fusion achieves the best results across all four different settings.
>
> | Method   | Noisy  | ASCM   | MPPCA  | DIP    | Nr2N   | N2S    | R2R    | P2S    | DDM2   | P2S2   | OURS   |
> |----------|--------|--------|--------|--------|--------|--------|--------|--------|--------|--------|--------|
> | **CSD-CC** | 0.797  | 0.934  | 0.884  | 0.868  | *0.959*  | 0.823  | 0.879  | 0.927  | 0.863  | 0.957  | **0.967** |
> | **CSD-CSO** | 0.614  | 0.844  | 0.750  | 0.477  | 0.908  | 0.468  | 0.731  | 0.754  | 0.810  | *0.934*  | **0.939** |
> | **DTI-CC**  | 0.789  | 0.942  | 0.881  | 0.875  | 0.961  | 0.831  | 0.872  | 0.725  | 0.845  | *0.973*  | **0.976** |
> | **DTI-CSO** | 0.484  | 0.789  | 0.614  | 0.381  | *0.872*  | 0.348  | 0.677  | 0.675  | 0.790  | 0.867  | **0.876** |
>
> **Table 2**. Comparison of SNR and CNR. **Bold** and *italic* fonts denote the best and the second-best performance, respectively.
>
> | Method   | ASCM   | MPPCA  | DIP    | Nr2N    | N2S    | R2R   | P2S    | DDM2   |  P2S2  | OURS   |
> |----------|--------|--------|--------|--------|--------|--------|--------|--------|--------|--------|
> | **SNR**  | -0.7251 | 0.2372 | 0.1035 | -0.1598 | 0.2266 | -0.0099 | -0.1616 | *1.3040* | 0.1526 | **1.5735** |
> | **CNR**  | -1.0513 | 0.2191 | 0.0567 | -0.2004 | -0.0304 | -0.1161 | -0.3694 | *1.2725* | 0.1177 | **1.5687** |
> ----------
> **Q2:** The only quantitative results use simulations, which seem likely to be skewed towards to capabilities of the proposed algorithm.
>
> **A2:** Thank you for your valuable feedback. In Patch2Self [2], Microstructure Model Fitting is evaluated using quantitative results (i.e., the $R^2$ metric in Figure 4 of our submission). In DDM2 [3], CNR and SNR metrics are considered quantitative results (in Section 4.3 "Quantitative results on SNR/CNR metrics" of our submission), while Fractional Anisotropy (in Section 4.2 "Effect on diffusion signal estimates") is regarded as both quantitative and clinically relevant. We will further emphasize the quantitative evaluation of these tasks in the main text to enhance clarity or directly include the tables as shown in **A1**.
>
> ----------
> **Q3:** The qualitative results on actual human data are questionable as to whether they show improvement over baselines. Even if they do, these are single cherry-picked examples and it is not clear whether these are advantages that manifest over large collections of images/scenarios.
>
> **A3:**  We include additional qualitative results in the Appendix and directly refer to the correspoding Figure in the main paper. We believe that this would provide a more complete picture of the method's performance. Regarding performance over large collections of images or scenarios, the effectiveness of our method can be demonstrated using the quantitative results of microstructure model fitting, as well as CNR and SNR metrics.
>
> ----------
> **References:**
>
> [1] Veraart J, Novikov D S, Christiaens D, et al. Denoising of diffusion MRI using random matrix theory[J]. Neuroimage, 2016, 142: 394-406.
>
> [2] Fadnavis S, Batson J, Garyfallidis E. Patch2Self: Denoising Diffusion MRI with Self-Supervised Learning​[J]. Advances in Neural Information Processing Systems, 2020, 33: 16293-16303.
>
> [3] Xiang T, Yurt M, Syed A B, et al. DDM $^2$: Self-Supervised Diffusion MRI Denoising with Generative Diffusion Models[C]//The Eleventh International Conference on Learning Representations.

---

> > ### Comment · Reviewer_9LpF · 2024-11-26
> >
> > THanks for the efforts to address the reviewers comments. The paper has improved with the addition of baselines. I'm leaving my score as it is, as I still don't really see compelling evidence that this work stands out from the crowd of methods addressing this problem on real data.  I wouldn't strongly object to this paper being accepted, but I don't think it is high priority.

---

> > > ### Author Response · Authors · 2024-11-27
> > >
> > > Thank you for your valuable feedback. The discussion phase will conclude on `December 2nd`, and we would be grateful for any additional questions or suggestions you may have.

---

### Official Review · Reviewer_K1kH · 2024-11-02

**Soundness:** 4
**Presentation:** 3
**Contribution:** 4
**Rating:** 10
**Confidence:** 4

**Summary:**

This is a new denoising method for dMRI data. Combines DL-based diffusion models with a bit of fusion. The fusion process stabilizes issues that DDM2 has.

I suggest that the authors refrain from large claims. For example, it says that it outperforms the other methods. But I do not see any speed or memory comparisons.

In the comparisons I would also add MPPCA. I would also cite Patch2Self2 (CVPR 24). Patch2Self has clearly outperformed MPPCA however still many people use MPPCA.

The paper does a great job on the methodological sections.
In providing code and using open source standards.

However, at least a thorough review of language is required.

Qualitatively it is hard to see large advantages over Patch2Self but nonetheless the method is useful.

In the revision please report time and memory usage. I would also compare against Patch2Self2 if possible.

Also it would be important to explain the setup. What GPUs were used for training?

**Strengths:**

Great way to stabilize the diffusion process.

**Weaknesses:**

Refrain from large claims.

Report time and memory usage.

Report setup (GPU types, numbers and VRAM).

Check statements.

**Questions:**

What is the actual minimum number of B0 and DWI volumes required ?

Can this work with data that have a single B0? Would that be denoised?

---

> ### Author Response · Authors · 2024-11-21
>
> ----------
> Thank you for your valuable feedback.
>
> ----------
> **Q1:** Refrain from large claims.
>
> **A1:** We will check the claims carefully in the revised version (Specifically, "outperforms other methods" to "achieves SOTA performance in microstructure modeling, tractography tracking, and various other downstream tasks", etc.).
>
> ----------
> **Q2:** Report time and memory usage, setup (GPU types, numbers and VRAM).
>
> **A2:** GPU types: RTX GeForce 3090; numbers of GPUs: 1; VRAM: 5578MB. The training duration for one ${\mathcal{F}_\theta }$ is approximately five hours on a single RTX GeForce 3090 GPU. These details will be reported in the "Training details" section of "Experiment and Reproducibility Details" in Appendix D.
>
> ----------
> **Q3:** Check statements.
>
> **A3:** We will carefully review the statements in the updated version to ensure clarity.
>
> ----------
> **Q4:** More comparisons.
>
> **A4:** We will include MPPCA (Neuroimage2016) [1], Noise2Score (N2S, NeurIPS 2021) [3], Recorrupted2Recorrupted (R2R, CVPR 2021) [4], and Patch2Self2 (P2S2, CVPR 2024) [2] for further comparisons on **tractography, microstructure model fitting, diffusion signal estimates, CNR and SNR metrics**. Due to page limitations, we will include the quantitative results and its visualizations in the appendix. Here is an abstract of the quantitative results:
>
> **Table 1**. ↑$R^2$ of microstructure model fitting on CSD & DTI. **Bold** and *italic* fonts denote the best and the second-best performance, respectively. As measured by $R^2$, Di-Fusion achieves the best results across all four different settings.
>
> | Method   | Noisy  | ASCM   | MPPCA  | DIP    | Nr2N   | N2S    | R2R    | P2S    | DDM2   | P2S2   | OURS   |
> |----------|--------|--------|--------|--------|--------|--------|--------|--------|--------|--------|--------|
> | **CSD-CC** | 0.797  | 0.934  | 0.884  | 0.868  | *0.959*  | 0.823  | 0.879  | 0.927  | 0.863  | 0.957  | **0.967** |
> | **CSD-CSO** | 0.614  | 0.844  | 0.750  | 0.477  | 0.908  | 0.468  | 0.731  | 0.754  | 0.810  | *0.934*  | **0.939** |
> | **DTI-CC**  | 0.789  | 0.942  | 0.881  | 0.875  | 0.961  | 0.831  | 0.872  | 0.725  | 0.845  | *0.973*  | **0.976** |
> | **DTI-CSO** | 0.484  | 0.789  | 0.614  | 0.381  | *0.872*  | 0.348  | 0.677  | 0.675  | 0.790  | 0.867  | **0.876** |
>
> **Table 2**. Comparison of SNR and CNR. **Bold** and *italic* fonts denote the best and the second-best performance, respectively.
>
> | Method   | ASCM   | MPPCA  | DIP    | Nr2N    | N2S    | R2R   | P2S    | DDM2   |  P2S2  | OURS   |
> |----------|--------|--------|--------|--------|--------|--------|--------|--------|--------|--------|
> | **SNR**  | -0.7251 | 0.2372 | 0.1035 | -0.1598 | 0.2266 | -0.0099 | -0.1616 | *1.3040* | 0.1526 | **1.5735** |
> | **CNR**  | -1.0513 | 0.2191 | 0.0567 | -0.2004 | -0.0304 | -0.1161 | -0.3694 | *1.2725* | 0.1177 | **1.5687** |
>
> ----------
> **Q5:** What is the actual minimum number of B0 and DWI volumes required ? Can this work with data that have a single B0? Would that be denoised?
>
> **A5:** *What is the actual minimum number of B0 and DWI volumes required ?*
> For B0 volumes, our method does not require them to perform denoising. This is evident in our implementation (`config/hardi_150.json`), where `"valid_mask": [10,160]` filters out B0 volumes. Regarding the minimum number of DWI volumes required, this can be considered from two perspectives:
> 1. Total volumes: We conducted an experiment using only 30 DWI volumes (Stanford HARDI dataset), and Di-Fusion was still able to perform denoising effectively under this condition. Qualitative results are presented in the Appendix of our revised version.
> 2. Input volumes: This is fixed to 1 in our implementation.
>
> *Can this work with data that have a single B0? Would that be denoised?*
> Our method does not require B0 volumes for denoising. Without B0 volumes, Di-Fusion is still capable of performing denoising effectively.
>
> ----------
> **References:**
>
> [1] Veraart J, Novikov D S, Christiaens D, et al. Denoising of diffusion MRI using random matrix theory[J]. Neuroimage, 2016, 142: 394-406.
>
> [2] Fadnavis S, Chowdhury A, Batson J, et al. Patch2Self2: Self-supervised Denoising on Coresets via Matrix Sketching[C]//Proceedings of the IEEE/CVF Conference on Computer Vision and Pattern Recognition. 2024: 27641-27651.
>
> [3] Kim K, Ye J C. Noise2score: tweedie’s approach to self-supervised image denoising without clean images[J]. Advances in Neural Information Processing Systems, 2021, 34: 864-874.
>
> [4] Pang T, Zheng H, Quan Y, et al. Recorrupted-to-recorrupted: Unsupervised deep learning for image denoising[C]//Proceedings of the IEEE/CVF conference on computer vision and pattern recognition. 2021: 2043-2052.

---

### Official Review · Reviewer_Dn7R · 2024-11-03

**Soundness:** 2
**Presentation:** 2
**Contribution:** 3
**Rating:** 5
**Confidence:** 4

**Summary:**

This paper presents a new self-supervised learning-based denoising method for diffusion MRI (dMRI). The proposed method leveraged the diffusion modeling concept, but instead of training a diffusion model with “clean images” as x_0 and noise as x_T, it utilized two diffusion weighted images (DWIs) with different diffusion encodings at both ends of a “diffusion-like” process. A denoising network was trained by predicting one DWI using a linear combination of two DWIs and an added noise term. The linear combination coefficients are time-dependent and determined via a scheduling strategy similar to training a diffusion model. The network was then used for a conditional sampling step for generating the final denoised images. The idea to utilize images acquired with different diffusion encodings to denoise one of them is interesting and the training strategy is an interesting approach to leverage the diffusion modeling concept, especially with training only latter diffusion steps to reduce hallucinations. However, several key assumptions made are questionable and the overall methodology and presentation lacks clarity. Evaluation using only dMRI signal model goodness of fit is limited and can be biased. There are a few overstatements that can mislead the readers. Detailed comments can be found below.

**Strengths:**

A diffusion-like modeling that learns the relationship between two DWI volumes with different diffusion encodings to denoise one or each other.

Training only later step diffusion to avoid hallucination

A fusion strategy that exploits linear combination of two DWIs with different contrasts with time-dependent coefficients and iterative refinement.

Extensive evaluations using both simulations that exactly followed the assumptions for the proposed methodology and practical magnitude DWI data.

**Weaknesses:**

There are statements that can be misleading in the context of MR physics (aka domain knowledge). For example, "the noise predominantly originates from physical interferences (Fadnavis et al., 2020a)". This statement about physical interferences is  both vague and inaccurate. This work is dealing with thermal noise or noise resulting from thermal noise in the measurements, which is not really physical interferences depending on how ones interpret them. Another example, "Different clinical applications require varying numbers of diffusion vectors and acquisition strategies, which makes modeling the noise distribution and further implementing denoising techniques challenging". Acquiring DWIs with varying numbers of diffusion vectors had nothing to do with the difficulty of  modeling noise distribution.

Many key assumptions for the proposed method was built on do not hold which made the theoretical/mathematical foundations questionable, e.g.,
a) It seems that the authors assumed DWIs acquired with different diffusion encodings had the same underlying “clean” image and were corrupted by independent noise. This is inaccurate. In fact, two DWIs can have rather different contrasts due to the diffusion encoding effects, e.g., different diffusion encoding directions. More specifically, x and x’ cannot be simply modeled as the same y plus different noise. What are the implications of this assumption not met?

b) Line 111: The authors claimed that that the proposed method does not require an explicit noise model. This is an overstatement. The J-invariance assumption, which formed the basis of the training objective in Eq. (9) implicitly requires that the noise distribution be zero-means and conditionally independent. Furthermore, additive noise model was assumed, x = y + n1 (Line 200). In dMRI, the magnitude images with higher b-values (stronger diffusion weightings) can have lower SNR for which additive noise may not hold. These need to be clarified.

-  Overall, the presentation lacks clarity and there seem to be some concerning inaccuracies.
a) The linear combination relationship claimed in Section 3.1 does not seem accurate. I checked the derivation. Eq. 31 is correct which is known (so this is not a contribution of the authors), but I'm not sure about going from Eq. 31 to 32 as F_theta predicts x_0, but they are not equal, and there is also an additional term of sigma_t^2*z. Therefore, I don't think it's a correct statement to say x_(t-1) is a linear interpolation between x_out and x_t. But is this really needed for the proposed method? I really don’t see a connection between what’s argued theoretically and what’s actually being implemented.

b) There are a few other inaccurate mathematical statements and notations which are confusing. For example, Eq. 7, the left side has q(x1:T |xt*) which is a joint distribution for x1 to xT, and the right side is a Gaussian distribution for xt.
On Line 160: {xt}1:T was described as”obtained from the reverse process.” However, in
Figure 1 and on the right side of Equation (7) on Line 186, it appears that xt is a corrupted version of xt*.  This interpretation, along with the notation in the Fig. 1, implies that {xt}1:T would represent a forward process. It appears to this reviewer the authors had not been using a consistent definition of forward and reverse diffusion which made the overall description rather confusing. These are just examples of inconsistencies found.

c) According to the J-invariance property, the noise should ideally have zero mean
and be conditionally independent of the target output. This requirement is necessary to ensure that the expected loss for self-supervised training asymptotical approaching the supervised loss. However, the input to F(.) in Eq. (9) includes xt*, which is a linear combination of x and x’ (Eq. (6)). Given that x serves as the supervision signal for the loss, this implies a correlation between the input x∗t and the target x, which would violate the conditional independence requirement for J-invariance.

**Questions:**

In Eq. (5) on Line 155, the authors highlighted a specific term as the ”major difference” between xt−1 and x^bar_t−1. Could the authors clarify why this particular term is considered the primary source of difference? Furthermore, can the authors elaborate on the underlying reason(s) for the “drift” in the model and how it emerges during the reverse diffusion process?

According to the definition of the Fusion process in Eq. (6)  and the “forward process” in Eq. (7), it appears that the starting point for the forward process changes based on t, as x_t* is dependent on t. This dependence implies that the Fusion process dynamically adjusts the starting point of the forward process at each step, which is unconventional compared to typical diffusion models. Could the authors clarify the rationale behind this design?

Other more recent self-supervised denoising methods should be compared, if not for all, e.g., Noise2Score and Recorrupted2Recorrupted etc.

---

> ### Author Response · Authors · 2024-11-21
>
> ----------
> Thank you for the review and feedback.
>
> ----------
>
> Answering the questions:
>
> **Q1:** In Eq. (5) on Line 155, the authors highlighted a specific term as the ”major difference” between xt−1 and x^bar_t−1. Could the authors clarify why this particular term is considered the primary source of difference? Furthermore, can the authors elaborate on the underlying reason(s) for the “drift” in the model and how it emerges during the reverse diffusion process?
>
> **A1:** **Intuitively,** through the Fusion process, we gradually feed the target denoising slice to the model, guiding it to optimize towards a fixed direction during the training phase, thereby mitigating the drift.
>
> **In equation form,**
> The training objective without the Fusion process is:
>
> ${L _ {{\rm{simple}}}}(\theta ): = {{\mathbb{E}} _ {t,x',{\xi _ {x - x'}}}}\left[ {{{\left\| {x - {\cal{F} _ \theta }(\sqrt {{{\bar \alpha } _ t}} x' + \sqrt {1 - {{\bar \alpha } _ t}} {\xi _ {x - x'}},t)} \right\|}^2}} \right]$
>
> The sampling start from $x _ {T _ 1}=\sqrt {{{\bar \alpha } _ {T _ 1}}} x' + \sqrt {1 - {{\bar \alpha } _ {T _ 1}}} {\xi _ {x - x'}}$ (the reverse process serves as the mapping from $x'$ to $x$), the reverse process is:
>
> $x _ {{T _ 1}-1}=\frac{{\sqrt {{{\bar \alpha } _ {{T _ 1} - 1}}}{\beta  _ {T _ 1}}}}{{1 - {{\bar \alpha } _ {T _ 1}}}}{{{\cal F} _ \theta }\left( {x _ {T _ 1},{T _ 1}} \right)} + \frac{{\sqrt {{\alpha _ {T _ 1}}} \left( {1 - {{\bar \alpha } _ {{T _ 1} - 1}}} \right)}}{{1 - {{\bar \alpha } _ {T _ 1}}}}{x _ {T _ 1}} + {\sigma _ {T _ 1}}z$
>
> This contradicts the training objective. During training, the ${T _ 1}-1$ step is trained with
>
> ${\left\| {x - {\cal{F} _ \theta }(\sqrt {{{\bar \alpha } _ {{T _ 1}-1}}} x' + \sqrt {1 - {{\bar \alpha } _ {{T _ 1}-1}}} {\xi _ {x - x'}},{{T _ 1}-1})} \right\|}^2$
>
> If we still feed ${{\cal F} _ \theta }$ with $x _ {{T _ 1}-1}$ and ${T _ 1}-1$, which is different from the training objective, i.e.,
>
> ${x _ {{T _ 1} - 1}} = \underbrace {\frac{{\sqrt {{{\bar \alpha } _ {{T _ 1} - 1}}} {\beta _ t}}}{{1 - {{\bar \alpha } _ {T _ 1}}}}{{{\cal F} _ \theta }\left( {{x _ {T _ 1}},{T _ 1}} \right)}} _ {{\rm{major}}{\kern 1pt} {\rm{difference}}} + \frac{{\sqrt {{\alpha _ {T _ 1}}} \left( {1 - {{\bar \alpha } _ {{T _ 1} - 1}}} \right)}}{{1 - {{\bar \alpha } _ {T _ 1}}}}{x _ {T _ 1}} + {\sigma _ {T _ 1}}z \ne {{\bar x} _ {{T _ 1} - 1}} = \underbrace{\sqrt {{{\bar \alpha } _ {{T _ 1} - 1}}} x' + \sqrt {1 - {{\bar \alpha } _ {{T _ 1} - 1}}} z} _ {\rm training \ objective \ w/o \ Fusion}$
>
> We assume that ${x _ {T _ 1}}$ is a noisy version of $x'$ (close to $x'$), and the output of $\cal F _ \theta$ approximates $x$. The reverse diffusion process gradually maps $x'$ to $x$, so the training objective should involve a combination of the two endpoints and noise, similar to DDBM [8].
>
> **In summary,**
> The "drift" arises because, if we directly feed $x _ {{T _ 1}-1}$ and ${T _ 1}-1$ into ${{\cal F} _ \theta }$, the output would deviate slightly further from $x$. This occurs because during training, ${{\cal F} _ \theta }$ is optimized only with the objective:
> ${\left\| {x - {\cal{F} _ \theta }(\sqrt {{{\bar \alpha }_{{T _ 1}-1}}} x' + \sqrt {1 - {{\bar \alpha } _ {{T _ 1}-1}}} {\xi _ {x - x'}},{{T _ 1}-1})} \right\|}^2$. Importantly, $x _ {{T _ 1}-1}$ is one step closer to $x$. ($x _ {{T _ 1}-1}= \frac{{\sqrt {{{\bar \alpha } _ {{T _ 1} - 1}}}{\beta  _ {T _ 1}}}}{{1 - {{\bar \alpha } _ {T _ 1}}}}{{{\cal F} _ \theta }\left( {x _ {T _ 1},{T _ 1}} \right)}+ \frac{{\sqrt {{\alpha _ {T _ 1}}} \left( {1 - {{\bar \alpha } _ {{T _ 1} - 1}}} \right)}}{{1 - {{\bar \alpha } _ {T _ 1}}}}{x _ {T _ 1}} + {\sigma _ {T _ 1}}z$), rather than simply being a noisy version of $x'$. This drift accumulates over the sampling chain, ultimately leading the result to drift toward another slice.
>
> ----------
> **Q2:** According to the definition of the Fusion process in Eq. (6) and the “forward process” in Eq. (7), it appears that the starting point for the forward process changes based on t, as x_t* is dependent on t. This dependence implies that the Fusion process dynamically adjusts the starting point of the forward process at each step, which is unconventional compared to typical diffusion models. Could the authors clarify the rationale behind this design?
>
> **A2:** This idea is motivated by Noise2Void [11], Noisier2Noise [10], and Noise-as-Clean [9], which use data augmentation to construct training data. Specifically, Noise2Void achieves this through a blind-spot strategy, while Noise2Noise and Noise-as-Clean add additional noise to the original noisy image. In our approach, we leverage such a dynamic combination (the Fusion process) and continuously varying noise (the "Di-" process) to provide the model with more augmented training data, thereby enhancing its robustness. We will clarify this in revision.
>
> ----------

---

> > ### Author Response · Authors · 2024-11-21
> >
> > ----------
> > **Q3:** Other more recent self-supervised denoising methods should be compared, if not for all, e.g., Noise2Score and Recorrupted2Recorrupted etc.
> >
> > **A3:** Recorrupted2Recorrupted (R2R) [3] and Noise2Score (N2S) [2] are self-supervised denoising methods for natural images. We included these two methods, along with MPPCA [4] and P2S2 [5], for further comparisons on **tractography, microstructure model fitting, diffusion signal estimates, CNR and SNR metrics**. Due to page limitations, we will include the quantitative results and its visualizations in the appendix. Here is an abstract of the quantitative results:
> >
> > **Table 1**. ↑$R^2$ of microstructure model fitting on CSD & DTI. **Bold** and *italic* fonts denote the best and the second-best performance, respectively. As measured by $R^2$, Di-Fusion achieves the best results across all four different settings.
> >
> > | Method   | Noisy  | ASCM   | MPPCA  | DIP    | Nr2N   | N2S    | R2R    | P2S    | DDM2   | P2S2   | OURS   |
> > |----------|--------|--------|--------|--------|--------|--------|--------|--------|--------|--------|--------|
> > | **CSD-CC** | 0.797  | 0.934  | 0.884  | 0.868  | *0.959*  | 0.823  | 0.879  | 0.927  | 0.863  | 0.957  | **0.967** |
> > | **CSD-CSO** | 0.614  | 0.844  | 0.750  | 0.477  | 0.908  | 0.468  | 0.731  | 0.754  | 0.810  | *0.934*  | **0.939** |
> > | **DTI-CC**  | 0.789  | 0.942  | 0.881  | 0.875  | 0.961  | 0.831  | 0.872  | 0.725  | 0.845  | *0.973*  | **0.976** |
> > | **DTI-CSO** | 0.484  | 0.789  | 0.614  | 0.381  | *0.872*  | 0.348  | 0.677  | 0.675  | 0.790  | 0.867  | **0.876** |
> >
> > **Table 2**. Comparison of SNR and CNR. **Bold** and *italic* fonts denote the best and the second-best performance, respectively.
> >
> > | Method   | ASCM   | MPPCA  | DIP    | Nr2N    | N2S    | R2R   | P2S    | DDM2   |  P2S2  | OURS   |
> > |----------|--------|--------|--------|--------|--------|--------|--------|--------|--------|--------|
> > | **SNR**  | -0.7251 | 0.2372 | 0.1035 | -0.1598 | 0.2266 | -0.0099 | -0.1616 | *1.3040* | 0.1526 | **1.5735** |
> > | **CNR**  | -1.0513 | 0.2191 | 0.0567 | -0.2004 | -0.0304 | -0.1161 | -0.3694 | *1.2725* | 0.1177 | **1.5687** |
> >
> > ----------
> > **W1:** There are statements that can be misleading in the context of MR physics (aka domain knowledge). For example, "the noise predominantly originates from physical interferences (Fadnavis et al., 2020a)". This statement about physical interferences is both vague and inaccurate. This work is dealing with thermal noise or noise resulting from thermal noise in the measurements, which is not really physical interferences depending on how ones interpret them. Another example, "Different clinical applications require varying numbers of diffusion vectors and acquisition strategies, which makes modeling the noise distribution and further implementing denoising techniques challenging". Acquiring DWIs with varying numbers of diffusion vectors had nothing to do with the difficulty of modeling noise distribution.
> >
> > **W-A1:** "the noise predominantly originates from physical interferences (Fadnavis et al., 2020a)"  is changed to "the noise predominantly originates from numerous factors including thermal fluctuations (Fadnavis et al., 2020a)" for clarity. Physical interferences involve signal disturbances caused by external physical factors such as magnetic field fluctuations, mechanical vibrations, thermal fluctuations, or noise generated by other instruments.
> >
> > "Different clinical applications require varying numbers of diffusion vectors and acquisition strategies, which makes modeling the noise distribution and further implementing denoising techniques challenging." is changed to "Different clinical applications require varying numbers of diffusion vectors and acquisition strategies, leading to diverse noise sources and distributions, which complicates noise modeling and denoising implementation." for clarity.
> >
> > ----------

---

> > ### Author Response · Authors · 2024-11-21
> >
> > **W2:** Many key assumptions for the proposed method was built on do not hold which made the theoretical/mathematical foundations questionable, e.g., a) It seems that the authors assumed DWIs acquired with different diffusion encodings had the same underlying “clean” image and were corrupted by independent noise. This is inaccurate. In fact, two DWIs can have rather different contrasts due to the diffusion encoding effects, e.g., different diffusion encoding directions. More specifically, x and x’ cannot be simply modeled as the same y plus different noise. What are the implications of this assumption not met?
> >
> > **W-A2:** We follow the claim, "While each of these acquired volumes may be quite noisy, the fact that the **same structures** are represented in each offers the potential for significant denoising" from the "Introduction" section of [7], and "Noise2Noise (Lehtinen et al., 2018) constructs training pairs of two independent noisy measurements of the **same target** and trains a network to transform one measurement to the other. 4D MRI by their nature own independent noisy samples acquired at different gradient directions. In our experiments, we train Noise2Noise model by using the same slices at different volumes" as mentioned in Appendix B of [1].
> >
> > ----------
> > **W3:** b) Line 111: The authors claimed that that the proposed method does not require an explicit noise model. This is an overstatement. The J-invariance assumption, which formed the basis of the training objective in Eq. (9) implicitly requires that the noise distribution be zero-means and conditionally independent. Furthermore, additive noise model was assumed, x = y + n1 (Line 200). In dMRI, the magnitude images with higher b-values (stronger diffusion weightings) can have lower SNR for which additive noise may not hold. These need to be clarified.
> >
> > **W-A3:** We intended to convey that *the proposed method does not require an explicit noise model* in the sense that it does not necessitate training an additional noise model (like the case in DDM2 [1]) or making specific assumptions about the noise distribution (assume the noise distribution is Gaussian distribution). Instead, it only imposes a constraint that the noise has zero mean (we do this by performing a zero-mean operation in Eq. (8)) and assumes it to be additive. We will revise this statement accordingly in the updated version of the paper to change this statement (from "an explicit noise model" to "extra noise model training") to improve clarity.
> >
> > ----------

---

> > ### Author Response · Authors · 2024-11-21
> >
> > ----------
> > **W4:** Overall, the presentation lacks clarity and there seem to be some concerning inaccuracies. a) The linear combination relationship claimed in Section 3.1 does not seem accurate. I checked the derivation. Eq. 31 is correct which is known (so this is not a contribution of the authors), but I'm not sure about going from Eq. 31 to 32 as F_theta predicts x_0, but they are not equal, and there is also an additional term of sigma_t^2*z. Therefore, I don't think it's a correct statement to say x_(t-1) is a linear interpolation between x_out and x_t. But is this really needed for the proposed method? I really don’t see a connection between what’s argued theoretically and what’s actually being implemented.
> >
> > **W-A4:** *But is this really needed for the proposed method? I really don’t see a connection between what’s argued theoretically and what’s actually being implemented.*
> >
> > Please refer to **A1**.
> >
> > *The linear combination relationship*
> >
> > We will change this to "a linear interpolation between ${x_{out}}$ and ${x_t}$ plus a noise" to improve clarity.
> >
> > *but I’m not sure about going from Eq. 31 to 32 as F_theta predicts x_0*
> >
> > In DDPM framework [6], a "noise predictor" is used to estimate the noise in $x _ t$​. Specifically, the reverse process $p _ \theta \left( {{x _ {t-1}}{\rm{|}}x_t} \right)$ starts by obtaining $x _ {0|t}$​ from  $x _ t$ and ${\epsilon} _ \theta$ (the noise predictor) using the equation:
> >
> > $x _ {0|t}=\frac{1}{\sqrt{{\bar\alpha _ t}}}(x _ t + \sqrt{(1 - {\bar \alpha _ t})}{\epsilon} _ \theta(x _ {t},{t}))$
> >
> > Then, the reverse process uses this $x _ {0|t}$​ to compute the mean of the Gaussian distribution for the next step in the reverse diffusion process, ${\tilde {\boldsymbol{\mu }} _ t}({x _ t},{x _ 0})$, which is:
> >
> > ${\tilde {\boldsymbol{\mu }} _ t}({x _ t},{x _ 0})=\frac{\sqrt{\alpha _ t}(1 - \bar{\alpha} _ {t-1})}{1 - \bar{\alpha} _ t} {x} _ t + \frac{\sqrt{\bar{\alpha} _ {t-1}}\beta _ t}{1 - \bar{\alpha} _ t} {x} _ {0|t}$
> >
> > Thus, the reverse process $p _ \theta \left( {{x _ {t-1}}{\rm{|}}x _ t} \right)$ can be described as:
> >
> > $q({x} _ {t-1} \vert {x} _ t, {x} _ {0|t}) = {\cal N}({x} _ {t-1}; {\tilde{\boldsymbol{\mu}}}({x} _ t,  {x} _ {0|t}),{\sigma _ t^2{\mathbf{I}}})$
> >
> > where $\sigma _ t^2 := {\beta _ t}$ (In Section 2 Diffusion models of our paper).
> > In contrast, in our method, we use a "data predictor" that directly predicts $x _ {0|t}$​, i.e., $x _ {0|t} = \mathcal{F} _ \theta(x _ t,t)$. Following a similar reverse process, we can perform the reverse process.
> >
> > ----------
> >
> > **W5:** b) There are a few other inaccurate mathematical statements and notations which are confusing. For example, Eq. 7, the left side has q(x1:T |xt*) which is a joint distribution for x1 to xT, and the right side is a Gaussian distribution for xt. On Line 160: {xt}1:T was described as”obtained from the reverse process.” However, in Figure 1 and on the right side of Equation (7) on Line 186, it appears that xt is a corrupted version of xt*. This interpretation, along with the notation in the Fig. 1, implies that {xt}1:T would represent a forward process. It appears to this reviewer the authors had not been using a consistent definition of forward and reverse diffusion which made the overall description rather confusing. These are just examples of inconsistencies found.
> >
> > **W-A5:** Thank you for pointing this out. The expression $q(x _ {1:T} |x _ t^*)$ was a typo. The correct version should be:
> >
> > $q\left( {{x _ {t}}{\rm{|}}x _ t^*} \right): = {\cal{N}}\left( {{x _ t};\sqrt {{{\bar \alpha } _ t}} x _ t^*, ({1 - {{\bar \alpha } _ t}}) {\mathbf{I}}} \right)$
> >
> > We have carefully reviewed the entire manuscript and made the necessary corrections.
> >
> > We directly followed the notation used in DDPM [6] for the definition of $x_t$. In DDPM, the same notation $\mathbf{x}_t$ is used for both the forward and reverse diffusion processes. Similarly, in our paper, we use $x_t$ for both the forward and reverse diffusion.
> >
> > ----------

---

> > ### Author Response · Authors · 2024-11-21
> >
> > ----------
> >
> > **W6:** c) According to the J-invariance property, the noise should ideally have zero mean and be conditionally independent of the target output. This requirement is necessary to ensure that the expected loss for self-supervised training asymptotical approaching the supervised loss. However, the input to F(.) in Eq. (9) includes xt*, which is a linear combination of x and x’ (Eq. (6)). Given that x serves as the supervision signal for the loss, this implies a correlation between the input x∗t and the target x, which would violate the conditional independence requirement for J-invariance.
> >
> > **W-A6:**  We impose a constraint that the noise is zero mean (achieved by applying a zero-mean operation as shown in Eq. (8)). Due to the nature of the diffusion schedule, the parameter $\lambda _ 2^t = \frac{{\sqrt{{\alpha _ t}} \left( {1 - {{\overline \alpha } _ {t - 1}}} \right)}}{{1 - {{\overline \alpha } _ t}}}$ is much larger than $\lambda _ 1^t = \frac{{\sqrt{{{\overline \alpha } _ {t - 1}}} {\beta _ t}}}{{1 - {{\overline \alpha } _ t}}}$ most of the time. This implies that $x _ t^*$ is predominantly composed of $x'$. Additionally, we add noise to each $x^* _ t$. The added noise varies with the slice index and volume index. In this way, $x _ t$ can be approximately considered independent of $x$.
> >
> >
> > ----------
> > **References:**
> >
> > [1] Xiang T, Yurt M, Syed A B, et al. DDM $^ 2$: Self-Supervised Diffusion MRI Denoising with Generative Diffusion Models[C]//The Eleventh International Conference on Learning Representations.
> >
> > [2] Kim K, Ye J C. Noise2score: tweedie’s approach to self-supervised image denoising without clean images[J]. Advances in Neural Information Processing Systems, 2021, 34: 864-874.
> >
> > [3] Pang T, Zheng H, Quan Y, et al. Recorrupted-to-recorrupted: Unsupervised deep learning for image denoising[C]//Proceedings of the IEEE/CVF Conference on Computer Vision and Pattern Recognition. 2021: 2043-2052.
> >
> > [4] Veraart J, Novikov D S, Christiaens D, et al. Denoising of diffusion MRI using random matrix theory[J]. Neuroimage, 2016, 142: 394-406.
> >
> > [5] Fadnavis S, Chowdhury A, Batson J, et al. Patch2Self2: Self-supervised Denoising on Coresets via Matrix Sketching[C]//Proceedings of the IEEE/CVF Conference on Computer Vision and Pattern Recognition. 2024: 27641-27651.
> >
> > [6] Ho J, Jain A, Abbeel P. Denoising diffusion probabilistic models[J]. Advances in neural information processing systems, 2020, 33: 6840-6851.
> >
> > [7] Fadnavis S, Batson J, Garyfallidis E. Patch2Self: Denoising Diffusion MRI with Self-Supervised Learning​[J]. Advances in Neural Information Processing Systems, 2020, 33: 16293-16303.
> >
> > [8] Zhou L, Lou A, Khanna S, et al. Denoising Diffusion Bridge Models[C]//The Twelfth International Conference on Learning Representations.
> >
> > [9] Xu J, Huang Y, Cheng M M, et al. Noisy-as-clean: Learning self-supervised denoising from corrupted image[J]. IEEE Transactions on Image Processing, 2020, 29: 9316-9329.
> >
> > [10] Moran N, Schmidt D, Zhong Y, et al. Noisier2noise: Learning to denoise from unpaired noisy data[C]//Proceedings of the IEEE/CVF Conference on Computer Vision and Pattern Recognition. 2020: 12064-12072.
> >
> > [11] Krull A, Buchholz T O, Jug F. Noise2void-learning denoising from single noisy images[C]//Proceedings of the IEEE/CVF conference on computer vision and pattern recognition. 2019: 2129-2137.

---

> > ### Comment · Reviewer_Dn7R · 2024-11-27
> > **Reviewer response to A1 and A2**
> >
> > A1: This should be clarified in the main text.
> >
> > A2: Thanks for the clarification. But then in this case, can we still call it a diffusion model or diffusion process? The entire structure changed, though motivated from data augmentation perspective.

---

> > > ### Author Response · Authors · 2024-11-27
> > >
> > > ___________
> > > Thank you for your feedback.
> > > ___________
> > > >This should be clarified in the main text.
> > >
> > > We have included additional explanations (`Section 3.1 and Appendix C.1`) to improve clarity.
> > > ___________
> > > >But then in this case, can we still call it a diffusion model or diffusion process? The entire structure changed, though motivated from data augmentation perspective.
> > >
> > > This can be regarded as a subset of diffusion models mainly due to the following reasons:
> > >
> > > 1.  **Di-Fusion** retains the gradual noising process (forward process) and the reverse denoising process (reverse process) inherent to diffusion models. The modifications are implemented within the theoretical framework of diffusion models.
> > > 2.  Several studies adopt a family of processes that interpolate between two paired distributions given as endpoints [1-5]. In CDDB [3], they summarize I2SB [4] and InDI [5] in `Table 1`, where the sampling distribution is considered as a diffusion process. The Fusion process operates similarly, and such methods (diffusion bridge models) can be considered as a subset of diffusion models.
> > > ___________
> > > **References:**
> > >
> > > [1] Zhou L, Lou A, Khanna S, et al. Denoising Diffusion Bridge Models[C]//The Twelfth International Conference on Learning Representations.
> > >
> > > [2] He G, Zheng K, Chen J, et al. Consistency Diffusion Bridge Models[C]//The Thirty-eighth Annual Conference on Neural Information Processing Systems.
> > >
> > > [3] Chung, Hyungjin, Jeongsol Kim, and Jong Chul Ye. "Direct diffusion bridge using data consistency for inverse problems." Advances in Neural Information Processing Systems, 2023.
> > >
> > > [4] Liu G H, Vahdat A, Huang D A, et al. I2SB: image-to-image Schrödinger bridge[C]//Proceedings of the 40th International Conference on Machine Learning. 2023: 22042-22062.
> > >
> > > [5] Delbracio M, Milanfar P. Inversion by Direct Iteration: An Alternative to Denoising Diffusion for Image Restoration[J]. Transactions on Machine Learning Research.

---

> > > > ### Comment · Reviewer_Dn7R · 2024-11-27
> > > >
> > > > Sorry for my unfinished comments.
> > > >
> > > > I don't think the authors can write authoritatively on MRI related topics. I remain concerned about many of the statements made.
> > > >
> > > > First, one very key assumption the authors made about the nature of the dMRI image series is wrong (regardless of how the results look. For conferences like ICLR, I don't think we should judge a work's contribution mainly based on how good the results are as we all know these results depend on many different factors, one being hyperparameter tunning which is often biased).  W2, W-A2: "We follow the claim, 'While each of these acquired volumes may be quite noisy, the fact that the same structures are represented in each offers the potential for significant denoising' from the 'Introduction' section of [7]" Others' incorrect statement does not change the fact that this is fundamentally incorrect. Different dMRI image with different diffusion encoding, ignoring noise, are not the same image and can encode very different microstructural information. Furthermore, one also needs to consider the role this assumption actually plays in the problem formulation.
> > > >
> > > > Second example, the noise in magnitude diffusion-weighted MR images, especially at higher b-values, are not additive "noise", and they are signal-dependent. Additionally, Re W-A1: "varying numbers of diffusion vectors and acquisition strategies" do not lead to "diverse noise sources". The things you are referring to are artifacts or image distortion due to different kinds of diffusion encoding strategies, but those are not the noise you are removing, at least in the typical sense of "denoising".
> > > >
> > > > W-A6: The argument seems weak. There is certainly dependence in this case.
> > > >
> > > > With these issues, I stand behind my original recommendation, unless the chairs and other reviewers believe there are strong contributions and innovations from the machine learning perspectives.

---

> > > > > ### Author Response · Authors · 2024-12-03
> > > > >
> > > > > ----------
> > > > > Thank you again for your time and efforts in reviewing our paper.
> > > > >
> > > > > ----------
> > > > > >one very key assumption the authors made about the nature of the dMRI image series is wrong
> > > > >
> > > > > Thank you for your reconsideration. We understand that different dMRI images with different diffusion encoding, ignoring noise, are not the same image and can encode very different microstructural information. But this assumption is made for practical implementation:
> > > > >
> > > > > (1) "Leveraging the fact that each 3D volume of the 4D data can be assumed to be an independent measurement of the same underlying object, P2S proposes constructing a large Casorati matrix wherein each column can be assumed to be linearly independent of the other columns." in `Section 2 of [1]`.
> > > > >
> > > > > (2) "In dMRI, it is often assumed that the diffusion signal carries redundant information between the gradient directions (also referred to as q-space)." in `Section 2.1 of [2]`.
> > > > >
> > > > > (3) "While each of these acquired volumes may be quite noisy, the fact that the same structures are represented in each offers the potential for significant denoising" in `Section 1 of [4]`.
> > > > >
> > > > > (4) "Noise2Noise (Lehtinen et al., 2018) constructs training pairs of two independent noisy measurements of the same target and trains a network to transform one measurement to the other. 4D MRI by their nature own independent noisy samples acquired at different gradient directions. In our experiments, we train Noise2Noise model by using the same slices at different volumes" as mentioned in `Appendix B of [5]`.
> > > > >
> > > > > Based on the redundancy of dMRI signals in canonical spaces [2], the PCA denoising methods [9-11] and P2S-like [4, 5] methods achieve success in dMRI denoising. n.b., MPPCA [9] and Patch2Self [4] have been integrated into widely-used open-source packages like DIPY [13] and QSIPrep [14], etc. We will talk about the limitations of this assumption and point out future work in the limitations and future work.
> > > > >
> > > > > ----------
> > > > > >the noise in magnitude diffusion-weighted MR images, especially at higher b-values, are not additive "noise", and they are signal-dependent.
> > > > >
> > > > > (5) "For practical purposes, it is commonly assumed that noise in the real and imaginary parts of k -space is generated using a zero-mean stationary Gaussian process with equal variance [7,8]" in `Section 2.1.1 of [3]`
> > > > >
> > > > > (6) "The diffusion signal is always corrupted by noise, $w \in \mathbb{R}^M$, which is typically modeled as additive and with zero mean, that is, $y=x+w$. Denoising the signal $y$ can be formulated as an estimation problem, where the goal is to estimate the noise-free signal $x$ from the observed data $y$." in `Section 2.1 of [2]`
> > > > >
> > > > > (7) "DWI estimation in the presence of noise can be formulated as a set of matrix recovery subproblems by arranging the DWI data into a set of matrices $Y$ of size $M \times N$. Each of these matrices is assumed to be the result of the additive corruption of an underlying (approximately) low but unknown rank $R$ matrix $X$ with a random matrix with Gaussian noise entries $W$, so we can write $Y = X + W$." in `Section 2 of [11]`
> > > > >
> > > > > (8) "DWI measures are assumed to follow a circularly symmetric complex Gaussian stationary noise distribution of zero mean and a given variance $\sigma^2$, $\mathcal{JN}(\mathbf{0},\sigma^2)$ modeling the Johnson-Nyquist noise observed in the quadrature receiver" in `Section 2.2 Additive Gaussian noise in DWI Reconstruction of [11]`
> > > > >
> > > > > Based on the claims (5-8), it is a feasible approach to assume additive noise. We will clearly restrict the noise we address and note that the method may not be effective for spatially and temporally varying image distortions (often referred to as physiological noise [16-18]) in the limitations. Thank you again for your time and efforts in helping us improve our paper.
> > > > >
> > > > > ----------

---

> > > > > ### Author Response · Authors · 2024-12-03
> > > > >
> > > > > ----------
> > > > > >The things you are referring to are artifacts or image distortion due to different kinds of diffusion encoding strategies, but those are not the noise you are removing, at least in the typical sense of "denoising".
> > > > >
> > > > > **Noise versus artifacts**
> > > > >
> > > > > In the literature, the term “noise” has been used to refer to different sources of undesired signal fluctuations. The thermal noise [14, 15] is the random signal fluctuations induced by the motion of electrons or ions. The collection of spatially and temporally varying image distortions, such as cardiac pulsation and motion, often referred to as physiological noise [16-18], are imaging artifacts.
> > > > >
> > > > > ----------
> > > > > **References:**
> > > > >
> > > > > [1] Fadnavis S, Chowdhury A, Batson J, et al. Patch2Self2: Self-supervised Denoising on Coresets via Matrix Sketching[C]//Proceedings of the IEEE/CVF Conference on Computer Vision and Pattern Recognition. 2024: 27641-27651.
> > > > >
> > > > > [2] Ramos‐Llordén G, Vegas‐Sánchez‐Ferrero G, Liao C, et al. SNR‐enhanced diffusion MRI with structure‐preserving low‐rank denoising in reproducing kernel Hilbert spaces[J]. Magnetic resonance in medicine, 2021, 86(3): 1614-1632.
> > > > >
> > > > > [3] Chen G, Wu Y, Shen D, et al. Noise reduction in diffusion MRI using non-local self-similar information in joint x− q space[J]. Medical image analysis, 2019, 53: 79-94.
> > > > >
> > > > > [4] Fadnavis S, Batson J, Garyfallidis E. Patch2Self: Denoising Diffusion MRI with Self-Supervised Learning​[J]. Advances in Neural Information Processing Systems, 2020, 33: 16293-16303.
> > > > >
> > > > > [5] Xiang T, Yurt M, Syed A B, et al. DDM $^ 2$: Self-Supervised Diffusion MRI Denoising with Generative Diffusion Models[C]//The Eleventh International Conference on Learning Representations.
> > > > >
> > > > > [7] Gudbjartsson H, Patz S. The Rician distribution of noisy MRI data[J]. Magnetic resonance in medicine, 1995, 34(6): 910-914.
> > > > >
> > > > > [8] Aja-Fernández S, Vegas-Sánchez-Ferrero G. Statistical analysis of noise in MRI[J]. Switzerland: Springer International Publishing, 2016.
> > > > >
> > > > > [9] Veraart J, Novikov D S, Christiaens D, et al. Denoising of diffusion MRI using random matrix theory[J]. Neuroimage, 2016, 142: 394-406.
> > > > >
> > > > > [10] Manjón J V, Coupé P, Concha L, et al. Diffusion weighted image denoising using overcomplete local PCA[J]. PloS one, 2013, 8(9): e73021.
> > > > >
> > > > > [11] Cordero-Grande L, Christiaens D, Hutter J, et al. Complex diffusion-weighted image estimation via matrix recovery under general noise models[J]. Neuroimage, 2019, 200: 391-404.
> > > > >
> > > > > [12] Garyfallidis E, Brett M, Amirbekian B, et al. Dipy, a library for the analysis of diffusion MRI data[J]. Frontiers in neuroinformatics, 2014, 8: 8.
> > > > >
> > > > > [13] Cieslak M, Cook P A, He X, et al. QSIPrep: an integrative platform for preprocessing and reconstructing diffusion MRI data[J]. Nature methods, 2021, 18(7): 775-778.
> > > > >
> > > > > [14] Johnson J B. Thermal agitation of electricity in conductors[J]. Physical review, 1928, 32(1): 97.
> > > > >
> > > > > [15] Nyquist H. Thermal agitation of electric charge in conductors[J]. Physical review, 1928, 32(1): 110.
> > > > >
> > > > > [16] Chang L C, Jones D K, Pierpaoli C. RESTORE: robust estimation of tensors by outlier rejection[J]. Magnetic Resonance in Medicine: An Official Journal of the International Society for Magnetic Resonance in Medicine, 2005, 53(5): 1088-1095.
> > > > >
> > > > > [17] Chang L C, Walker L, Pierpaoli C. Informed RESTORE: a method for robust estimation of diffusion tensor from low redundancy datasets in the presence of physiological noise artifacts[J]. Magnetic resonance in medicine, 2012, 68(5): 1654-1663.
> > > > >
> > > > > [18] Walker L, Chang L C, Koay C G, et al. Effects of physiological noise in population analysis of diffusion tensor MRI data[J]. Neuroimage, 2011, 54(2): 1168-1177.

---

### Official Review · Reviewer_TzM3 · 2024-11-04

**Soundness:** 3
**Presentation:** 3
**Contribution:** 3
**Rating:** 8
**Confidence:** 3

**Summary:**

This paper introduces Di-Fusion, a fully self-supervised diffusion MRI (dMRI) denoising method designed to enhance the signal-to-noise ratio (SNR) of MRI data without requiring clean reference data. The authors leverage novel late diffusion steps and an adaptive sampling process to create a single-stage framework that operates without an explicit noise model. Di-Fusion demonstrates superior performance over state-of-the-art denoising methods in tasks such as microstructure modeling and tractography. The method’s efficacy is validated through extensive quantitative and qualitative evaluations on real and simulated data.

**Strengths:**

- **Flexibility with data and noise models**: Instead of relying on explicit noise models or clean training data, the method relies on an N2N training strategy and pixel shuffling to reorganize the noise, providing strong generalization potential across different noise distributions. This suggests that the method has the potential to be applied to a wider range of denoising scenarios, such as cryo-EM.
- Compared to the current state-of-the-art method, DDM^2, this approach demonstrates comprehensive improvements. Not only does it outperform in terms of performance, but it is also simpler to implement. Notably, this method does not require additional denoiser training, significantly enhancing its practical usability.
- As a study on dMRI denoising, this paper conducts thorough and comprehensive experiments, including extensive comparisons and analyses on downstream task performance. This renders the work methodologically and experimentally well-rounded.

**Weaknesses:**

Please refer to the **Questions** section for details.

**Questions:**

- To my understanding, the primary goal of dMRI denoising is to reduce the number of gradients required during acquisition, thus accelerating DWI scanning. In downstream tasks based on DTI, the authors compare DTI metrics computed from noisy images with those from denoised images. Why did the authors not use more DWI data to compute a clean DTI metric as a reference for comparison?

---

> ### Author Response · Authors · 2024-11-21
>
> ----------
> Thank you for your valuable feedback.
>
> ----------
>
> **Q1:** To my understanding, the primary goal of dMRI denoising is to reduce the number of gradients required during acquisition, thus accelerating DWI scanning. In downstream tasks based on DTI, the authors compare DTI metrics computed from noisy images with those from denoised images. Why did the authors not use more DWI data to compute a clean DTI metric as a reference for comparison?
>
> **A1:** We did not use more DWI data to compute a clean DTI metric as a reference for comparison in our paper because
> 1. This was not used in prior studies [1-3]. To ensure a fair comparison of the performance across methods, we directly followed these metrics without modification;
> 2. In DTI, the characteristics of noise are relatively complex, as the noise in images exhibits significantly different properties. Therefore, directly averaging these images may introduce additional errors. This is likely one of the reasons why previous studies did not adopt this metric.
>
> ----------
> **References:**
>
> [1] Veraart J, Novikov D S, Christiaens D, et al. Denoising of diffusion MRI using random matrix theory[J]. Neuroimage, 2016, 142: 394-406.
>
> [2] Fadnavis S, Chowdhury A, Batson J, et al. Patch2Self2: Self-supervised Denoising on Coresets via Matrix Sketching[C]//Proceedings of the IEEE/CVF Conference on Computer Vision and Pattern Recognition. 2024: 27641-27651.
>
> [3] Fadnavis S, Batson J, Garyfallidis E. Patch2Self: Denoising Diffusion MRI with Self-Supervised Learning​[J]. Advances in Neural Information Processing Systems, 2020, 33: 16293-16303.

---

> ### Comment · Reviewer_TzM3 · 2024-11-28
>
> In fact, using more DWIs data to fit DTI as a reference is a reasonable and widely adopted approach [1-3]. This is intuitive—if no denoising algorithm is available, acquiring a large number of scans can improve the accuracy of tensor signal fitting. I will withhold my score for now.
>
> ## Reference
> > [1] Li, Zihan, et al. "DIMOND: DIffusion Model OptimizatioN with Deep Learning." Advanced Science (2024): 2307965.
>
> > [2] Li, Hongyu, et al. "SuperDTI: Ultrafast DTI and fiber tractography with deep learning." Magnetic resonance in medicine 86.6 (2021): 3334-3347.
>
> > [3] McNab, Jennifer A., et al. "Surface based analysis of diffusion orientation for identifying architectonic domains in the in vivo human cortex." Neuroimage 69 (2013): 87-100.

---

> > ### Author Response · Authors · 2024-11-28
> >
> > ___________
> > Thank you for your valuable feedback. We carefully checked the referenced papers [1-3].
> >
> > We utilized 10 volumes with a b-value of 0 and 10 volumes with a b-value of 2000 (both denoised and original DWI data) to estimate fractional anisotropy (FA), axial diffusivity (AD), mean diffusivity (MD), and radial diffusivity (RD) following the same steps in `Appendix D.2`. Additionally, we computed the reference for FA, AD, MD, and RD using all available DWI data. Based on this, we performed the calculation of PSNR and SSIM for all slices. The results are summarized in the following table:
> >
> > **Table 1**. Comparisons of PSNR↑ and SSIM↑ on DTI diffusion signal estimates. We use **Bold** and *italic* fonts to denote the best and the second-best performance, respectively.
> >
> > | Method | Metric | FA      | MD      | RD      | AD      |
> > |--------|--------|---------|---------|---------|---------|
> > | noisy | PSNR    | 23.47  | 31.79  | 33.56  | 26.39  |
> > | noisy | SSIM    | 0.8760  | 0.9635  | 0.9637  | 0.9247  |
> > | ASCM | PSNR    | 22.63  | 21.81  | 24.70  | 20.89  |
> > | ASCM | SSIM    | 0.8897  | 0.8188  | 0.8921  | 0.8102  |
> > | MPPCA | PSNR    | 26.37  | 35.28  | 36.35  | 29.63  |
> > | MPPCA | SSIM    | 0.9048  | 0.9751  | 0.9780  | 0.9534  |
> > | DIP | PSNR    | 24.76  | 33.01  | 32.42  | 30.21  |
> > | DIP | SSIM    | 0.8894  | 0.9675  | 0.9630  | 0.9453  |
> > | N2S | PSNR    | 23.13  | 29.71  | 31.08  | 25.84  |
> > | N2S | SSIM    | 0.8691  | 0.9469  | 0.9491  | 0.8973  |
> > | R2R | PSNR    | 25.93  | 34.54  | 33.82  | 30.21  |
> > | R2R | SSIM    | 0.8809  | 0.9708  | 0.9678  | 0.9394  |
> > | P2S | PSNR    | 24.18  | 32.16  | 30.62  | 33.27  |
> > | P2S | SSIM    | 0.9090  | 0.9708  | 0.9632  | 0.9677  |
> > | Nr2N | PSNR    | 29.47  | 38.42  | 38.48  | *34.87*  |
> > | Nr2N | SSIM    | 0.9354  | 0.9851  | 0.9865  | *0.9712*  |
> > | DDM2 | PSNR    | 26.77  | 37.53  | 37.39  | 33.21  |
> > | DDM2 | SSIM    | 0.9041  | 0.9872  | 0.9848  | 0.9610  |
> > | P2S2 | PSNR    | *30.08*  | *39.28*  | *40.23*  | 34.06  |
> > | P2S2 | SSIM    | *0.9432*  | *0.9894*  | *0.9921*  | 0.9701  |
> > | OURS | PSNR    | **30.79**  | **40.26**  | **40.35**  | **35.30**  |
> > | OURS | SSIM    | **0.9450**  | **0.9931**  | **0.9923**  | **0.9763**  |
> >
> > Thank you for helping us evaluate the effectiveness of our method from a different perspective.
> > ___________
> > **References:**
> >
> > [1] Li, Zihan, et al. "DIMOND: DIffusion Model OptimizatioN with Deep Learning." Advanced Science (2024): 2307965.
> >
> > [2] Li, Hongyu, et al. "SuperDTI: Ultrafast DTI and fiber tractography with deep learning." Magnetic resonance in medicine 86.6 (2021): 3334-3347.
> >
> > [3] McNab, Jennifer A., et al. "Surface based analysis of diffusion orientation for identifying architectonic domains in the in vivo human cortex." Neuroimage 69 (2013): 87-100.

---

### Official Review · Reviewer_FDwC · 2024-11-04

**Soundness:** 3
**Presentation:** 3
**Contribution:** 3
**Rating:** 6
**Confidence:** 2

**Summary:**

The paper proposes a novel self-supervised denoising method Di-Fusion that leverages the latter diffusion steps and an adaptive sampling process.  Di-Fusion outperforms two slightly older methods and a state-of-the-art approach on   and on downstream processes like tractography.

**Strengths:**

Paper is easy to follow.
Results across multiple real and simulated datasets, suggesting generalizability of approach.
Baseline is a recent state-of-the-art.
The authors released their code to the reviewers, which is well-written, informative and aides reproducibility.

**Weaknesses:**

Did the authors consider using: https://arxiv.org/pdf/2305.00042  and    https://arxiv.org/pdf/2309.05794  as baselines?
Better signpost the extensive results in the supplementary materials.
Some parts of the paper read a bit odd and should be checked for oddities e.g. from the introduction 'The MRI, including ...', 'Consequently, the denoising technique plays a crucial role..'

**Questions:**

Please see weaknesses above.

---

> ### Author Response · Authors · 2024-11-21
>
> ----------
> Thank you for your valuable feedback.
>
> ----------
> **Q1:** Did the authors consider using:  [https://arxiv.org/pdf/2305.00042](https://arxiv.org/pdf/2305.00042)  and  [https://arxiv.org/pdf/2309.05794](https://arxiv.org/pdf/2309.05794)  as baselines?
>
> **A1:** [https://arxiv.org/pdf/2305.00042](https://arxiv.org/pdf/2305.00042)(Cycle-guided Denoising Diffusion Probability Model for 3D Cross-modality MRI Synthesis) was not considered as baselines because: *1.* They didn't release the code in their official repository[[EmoryDeepBiomedicalImagingLaboratory/Cycle-guided-diffusion-model](https://github.com/EmoryDeepBiomedicalImagingLaboratory/Cycle-guided-diffusion-model)]. Our attempt to replicate their approach might lead to discrepancies in experimental details. *2.* Their approach does not inherently support dMRI denoising [1], and adapting it to this context might lead to suboptimal performance or require significant modifications, which could misrepresent the true effectiveness of their method. We will add these discussion with related work in updated version.
>
> [https://arxiv.org/pdf/2309.05794](https://arxiv.org/pdf/2309.05794)(Robust Physics-based Deep MRI Reconstruction Via Diffusion Purification) proposes an interesting pipeline for MRI reconstruction (During the **Diffusion Stage**, noise is added to the deconstructed image to simulate the corruption process. In the subsequent **Purification Stage**, sampling is guided by data consistency constraints to iteratively refine the image. Finally, the refined output is passed through $MoDL_{\theta_{\rm FT}}$, which produces the final reconstruction result.). However, $MoDL_{\theta_{\rm FT}}$ was fine-tuned using 3000 purified images (*"We note that the DM model was trained on the knee training dataset. We conduct our experiments on the fast MRI [3] dataset, using 3000 purified images for fine-tuning the pre-trained MoDL network."* in 'Experimental Setup' section of [2]), "purified" images here are ground truth images in the fast MRI dataset. In dMRI, we don't have the "purified" images that could used to train $MoDL_{\theta_{\rm FT}}$. Directly adapting it to dMRI might lead to suboptimal performance or require significant modifications, which could misrepresent the true effectiveness of their method. We will add these discussion with related work in updated version.
>
> We will include MPPCA (Neuroimage2016) [6], Noise2Score (NeurIPS 2021) [4], Recorrupted2Recorrupted (CVPR 2021) [5], and Patch2Self2 (CVPR 2024) [7] for further comparisons on **tractography, microstructure model fitting, diffusion signal estimates, CNR and SNR metrics**. Due to page limitations, we will include the quantitative results and its visualizations in the appendix. Here is an abstract of the quantitative results:
>
> **Table 1**. ↑$R^2$ of microstructure model fitting on CSD & DTI. **Bold** and *italic* fonts denote the best and the second-best performance, respectively. As measured by $R^2$, Di-Fusion achieves the best results across all four different settings.
> | Method   | Noisy  | ASCM   | MPPCA  | DIP    | Nr2N   | N2S    | R2R    | P2S    | DDM2   | P2S2   | OURS   |
> |----------|--------|--------|--------|--------|--------|--------|--------|--------|--------|--------|--------|
> | **CSD-CC** | 0.797  | 0.934  | 0.884  | 0.868  | *0.959*  | 0.823  | 0.879  | 0.927  | 0.863  | 0.957  | **0.967** |
> | **CSD-CSO** | 0.614  | 0.844  | 0.750  | 0.477  | 0.908  | 0.468  | 0.731  | 0.754  | 0.810  | *0.934*  | **0.939** |
> | **DTI-CC**  | 0.789  | 0.942  | 0.881  | 0.875  | 0.961  | 0.831  | 0.872  | 0.725  | 0.845  | *0.973*  | **0.976** |
> | **DTI-CSO** | 0.484  | 0.789  | 0.614  | 0.381  | *0.872*  | 0.348  | 0.677  | 0.675  | 0.790  | 0.867  | **0.876** |
>
> **Table 2**. Comparison of SNR and CNR. **Bold** and *italic* fonts denote the best and the second-best performance, respectively.
>
> | Method   | ASCM   | MPPCA  | DIP    | Nr2N    | N2S    | R2R   | P2S    | DDM2   |  P2S2  | OURS   |
> |----------|--------|--------|--------|--------|--------|--------|--------|--------|--------|--------|
> | **SNR**  | -0.7251 | 0.2372 | 0.1035 | -0.1598 | 0.2266 | -0.0099 | -0.1616 | *1.3040* | 0.1526 | **1.5735** |
> | **CNR**  | -1.0513 | 0.2191 | 0.0567 | -0.2004 | -0.0304 | -0.1161 | -0.3694 | *1.2725* | 0.1177 | **1.5687** |
> ----------

---

> ### Author Response · Authors · 2024-11-21
>
> ----------
> **Q2:** Better signpost the extensive results in the supplementary materials.
>
> **A2:** In the main paper, we directly refer to the corresponding images within the main text and provide detailed descriptions in the figure captions. Readers can easily access the specified location by clicking on the hyperlinks. We will further check to ensure that these tables and figures are properly referenced via hyperlinks to assist readers in accessing them quickly.
>
> ----------
> **Q3:** Some parts of the paper read a bit odd and should be checked for oddities e.g. from the introduction 'The MRI, including ...', 'Consequently, the denoising technique plays a crucial role..'
>
> **A3:** Thank you for your valuable feedback. We have reviewed the phrasing in the introduction and refined the sentences to improve readability. Specifically, for these two parts 'The MRI, including ...' is changed to 'Magnetic Resonance Imaging (MRI), including ...'; 'Consequently, the denoising technique plays a crucial role..' is changed to 'Consequently,  the denoising technique is a vital processing step'. We will review the paper again to improve clarity.
>
> ----------
> **References:**
>
> [1] Pan S, Chang C W, Peng J, et al. Cycle-guided denoising diffusion probability model for 3d cross-modality mri synthesis[J]. arXiv preprint arXiv:2305.00042, 2023.
>
> [2] Alkhouri I, Liang S, Wang R, et al. Robust Physics-based Deep MRI Reconstruction Via Diffusion Purification[J]. arXiv e-prints, 2023: arXiv: 2309.05794.
>
> [3] Zbontar J, Knoll F, Sriram A, et al. fastMRI: An open dataset and benchmarks for accelerated MRI[J]. arXiv preprint arXiv:1811.08839, 2018.
>
> [4] Kim K, Ye J C. Noise2score: tweedie’s approach to self-supervised image denoising without clean images[J]. Advances in Neural Information Processing Systems, 2021, 34: 864-874.
>
> [5] Pang T, Zheng H, Quan Y, et al. Recorrupted-to-recorrupted: Unsupervised deep learning for image denoising[C]//Proceedings of the IEEE/CVF conference on computer vision and pattern recognition. 2021: 2043-2052.
>
> [6] Veraart J, Novikov D S, Christiaens D, et al. Denoising of diffusion MRI using random matrix theory[J]. Neuroimage, 2016, 142: 394-406.
>
> [7] Fadnavis S, Chowdhury A, Batson J, et al. Patch2Self2: Self-supervised Denoising on Coresets via Matrix Sketching[C]//Proceedings of the IEEE/CVF Conference on Computer Vision and Pattern Recognition. 2024: 27641-27651.

---

> > ### Comment · Reviewer_FDwC · 2024-11-21
> > **Response to Official Comment**
> >
> > Thank you for your response and the additional results.
> >
> > > Some parts of the paper read a bit odd ...
> >
> > > Thank you for your valuable feedback. We have reviewed the phrasing in the introduction and refined the sentences to improve readability
> >
> > To be clear, I think the authors just need to run the paper through an (e.g. online) typing assistant.  I want to make it clear to the AC that I found the paper easy to follow and it is clear enough for acceptance.
> >
> > I still recommend this paper for acceptance.  As the focus in only on denoising diffusion MRI, I will keep my score.  There are more than 100 million MRI scans performed per year, many of which are from noisy hardware, so this paper has real-world downstream applications.

---

> > > ### Author Response · Authors · 2024-11-27
> > >
> > > Thank you again for reconsidering and providing us with your feedback. The discussion phase will close on `December 2nd`, and we welcome any further questions or suggestions you may have.

---

### Author Response · Authors · 2024-11-26

Dear AC and Reviewers,

We sincerely appreciate the engaging discussion. Below, we provide a summary of the revisions made in response to your feedback:

1.  More comparisons with other denoising methods, including MPPCA, Noise2Score (N2S), Recorrupted2Recorrupted (R2R), and Patch2Self2 (P2S2) (`Appendix E`) [FDwC, Dn7R, K1kH, 9LpF].
2.  Patch2Self2 has been included as a related work (`Section 2.3`). We report the time and memory usage of Di-Fusion (`Appendix D.1`). Further experiments on limited dMRI volumes (`Appendix E.8`) [K1kH].
3.  Emphasized the quantitative evaluations of microstructure model fitting and diffusion signal estimates (`Section 4.2`) [9LpF].
4.  Clarified statements (`Sections 1, 3.1 and 5`) [FDwC, K1kH, Dn7R] and corrected all typos (`Section 3.1 and Appendix B.1`) [Dn7R].

With the discussion phase ending on December 2nd, we warmly welcome any further comments or suggestions.

Once again, we sincerely thank you for your invaluable feedback and support in improving our work.

Best regards,

The authors

---

### Meta-Review · Area_Chair_Bj1r · 2024-12-17

**Metareview:**

The paper received five reviews, and three are supportive or strongly supportive of publication.

The reviewers emphasize that the method performs well, and that a strength is that it adopts a clever noise2noise training strategy. Moreover, the method relies on relatively few assumptions and is simple to implement. The experiments are comprehensive and thorough overall.

The reviewers also pointed out several issues that were fixed for the most part. In particular, Reviewer Dn7R identified some overstatements and found several assumptions questionable:
- A central issue pointed out by the reviewer is about an assumption about the nature of the dMRI image series being wrong, i.e., in reality different dMRI images with different diffusion encodings are not equal and can encode different information. The method assumes they are the same for practical implementation. I agree with the reviewer on that being an assumption that is violated in practice, however for self-supervised methods assumptions that are at best approximately true are required, and this assumptions enables to build an algorithm that works relatively well as shown in the paper.
- A second assumption that the reviewer points too is that the noise is assumed additive and independent, while this is true, the authors point to the the literature justifying the Gaussian assumption.

Based on the reviews and my own reading, this is an interesting contribution to self-supervised imaging, and I therefore recommend to accept the paper.

**Additional Comments On Reviewer Discussion:**

See above.

---

### Decision · Program_Chairs · 2025-01-22

Accept (Poster)